# Drift-Resilient TabPFN: In-Context Learning Temporal Distribution Shifts on Tabular Data

**Kai Helli**[*,1,2]   **David Schnurr**[*,1,3]   **Noah Hollmann**[1,4]   **Samuel Müller**[1]   **Frank Hutter**[5,1]

[1] University of Freiburg, [2] Technical University of Munich, [3] ETH Zurich,
[4] Charité University Medicine Berlin, [5] ELLIS Institute Tübingen, [*] Equal contribution.
Correspondence to `kai.helli@tum.de`

## Abstract

While most ML models expect independent and identically distributed data, this assumption is often violated in real-world scenarios due to distribution shifts, resulting in the degradation of machine learning model performance. Until now, no tabular method has consistently outperformed classical supervised learning, which ignores these shifts. To address temporal distribution shifts, we present Drift-Resilient TabPFN, a fresh approach based on In-Context Learning with a Prior-Data Fitted Network that learns the learning algorithm itself: it accepts the entire training dataset as input and makes predictions on the test set in a single forward pass. Specifically, it learns to approximate Bayesian inference on synthetic datasets drawn from a prior that specifies the model's inductive bias. This prior is based on structural causal models (SCM), which gradually shift over time. To model shifts of these causal models, we use a secondary SCM, that specifies changes in the primary model parameters. The resulting Drift-Resilient TabPFN can be applied to unseen data, runs in seconds on small to moderately sized datasets and needs no hyperparameter tuning. Comprehensive evaluations across 18 synthetic and real-world datasets demonstrate large performance improvements over a wide range of baselines, such as XGB, CatBoost, TabPFN, and applicable methods featured in the Wild-Time benchmark. Compared to the strongest baselines, it improves accuracy from 0.688 to 0.744 and ROC AUC from 0.786 to 0.832 while maintaining stronger calibration. This approach could serve as significant groundwork for further research on out-of-distribution prediction.

## 1   Introduction

In traditional machine learning the train and test data are assumed to be sampled from the same distribution [1]. However, this assumption of independent and identically distributed (i.i.d.) data is commonly violated in real-world scenarios due to distribution shifts, resulting in performance degradation of standard machine learning (ML) models over time [1, 2]. Research in the area of temporal domain generalization (Temporal DG) tries to address these shifts by developing methods that perform consistently across temporal domains and generalize beyond the training regimen, i.e. into the future. In fields such as healthcare, climate science, or finance, data is most often organized in a tabular format [3, 4]. Here shifts are driven by hidden variables such as policy or climate changes, equipment updates, seasonal changes, or activity cycles, limiting real-world model deployment [2].

Young and Steele [5] describe declining mortality quantification in a hospital system while Pasterkamp et al. [6] show deterioration of cardiovascular risk models over time, leading to increased mortality; Ganesan et al. [7] show the COVID-19 pandemic exacerbated this issue, as ICU mortality prediction models failed to adapt to the unique characteristics of COVID-19 patients; environmental models need to continually adapt to climate changes [8]; fraud detection models need to continuously adapt

38th Conference on Neural Information Processing Systems (NeurIPS 2024).

**(a)** We generate synthetic datasets by sampling structural causal models whose edges shift over time.

A structural causal model (SCM) with sparse edge shifts indicated in red.

A 2nd-order SCM generates the strength of edge shifts depending on a time domain.

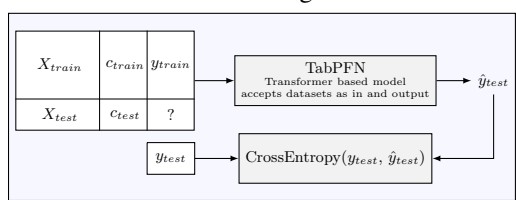
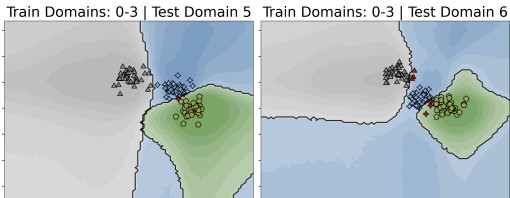

**(b)** TabPFN accepts entire datasets as inputs. Given millions of datasets from (a), it learns to make predictions on held-out test samples, learning the prediction algorithm itself with an inductive bias for addressing distribution shifts.

**(c)** Trained once, Drift-Resilient TabPFN can be applied to novel real-world data and makes predictions in a single forward pass, automatically detecting and extrapolating distribution shifts.

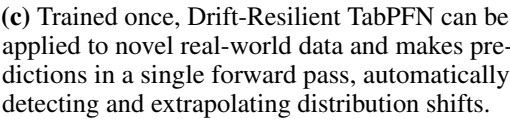

Figure 1: High-level overview of our method. We train a transformer that accepts entire datasets as input to learn the learning algorithm itself by training on millions of synthetic datasets once as part of algorithm development. The trained model can be applied to arbitrary real-world datasets. In (b), X, c, and y refer to features, time domain, and label respectively. In (c), we show predictions on test domains 4 (left) and 5 (right), where we see a distribution shift. Drift-Resilient TabPFN accurately updates decision boundaries in this example.

as the strategies of fraudsters adapt to the models themselves [9]; these feedback loops often arise in practice, where model deployment inherently causes a system change [10].

Robustness to such distribution shifts stands as a prominent challenge in current ML research [11]. So far multiple approaches have been proposed to address temporal distribution shifts using neural networks (NNs) [11–13]. However, modeling distribution shifts in tabular data presents a two-fold challenge: (i) NNs have struggled to model and extrapolate distribution shifts to date [11, 14] (ii) approaches for modeling distribution shifts have mostly employed NNs, while tree-based methods have consistently outperformed NNs in handling tabular data [3, 15–17] - leaving a wide methodological gap in addressing this common real-world scenario.

We provide a fresh perspective on predicting given distribution shifts by leveraging in-context-learning (ICL) to learn the prediction algorithm itself - bypassing many challenges encountered in this setting. Our approach builds on the foundation of Prior-Data Fitted Networks (PFNs; 18) and TabPFN (19; see Section 3.1). PFNs leverage large-scale ML and ICL techniques to approximate Bayesian inference accurately for any prior that can be sampled from. They are trained on millions of synthetic datasets sampled from this prior. For each such dataset, a supervised learning task is constructed, and the model is asked to predict on held out test samples. Then, the PFN is able to apply the principles learned on this synthetic data to real-world datasets, effectively having learned a prediction algorithm.

This paper introduces *Drift-Resilient TabPFN*, an adaptation of the TabPFN framework tailored for tabular datasets exhibiting temporal distribution shifts. Our idea is as follows: Data distribution shifts can be modeled as gradual changes to the structural causal model (SCM; 20, 21) underlying the data. By including the assumption that underlying models change over time into the approximated prior, our models learn to estimate, adapt to, and extrapolate these model changes. While likely no causal model exactly underlies real-world data, we find this approximation to be empirically and intuitively useful. Specifically, we suggest shifting edge weights between nodes that affect causal relationships while sampling instances in each dataset. To accurately model these edge shifts, we propose using a 2nd-order SCM, a secondary model that adjusts the primary SCM, that takes a time indicator as input and outputs the weight shifts. To represent our temporal domain indicators, we use Time2Vec [22].

To validate the robustness and adaptability of our approach, we conduct a comprehensive evaluation of 18 tabular datasets. Among these, 8 are synthetic toy problems, that allow us to analyze model behavior in detail. The remaining 10 are real-world datasets, each representative of different temporal distribution shifts. Our model is benchmarked against (i) state-of-the-art methods for handling temporal domain generalization (implemented in the Wild-Time benchmark [11]), as well as (ii) state-of-the-art methods for tabular data, that do not handle distribution shifts explicitly (the unmodified version of TabPFN, well-established classifiers such as XGBoost, Catboost, and LightGBM [23–25]). In addition, we qualitatively analyse the model's behavior and prediction characteristics.

Our model consistently outperforms all baselines on out-of-distribution data across both synthetic and real-world datasets while simultaneously exhibiting strong calibration. Qualitatively, we find that Drift-Resilient TabPFN dynamically adjusts its decision boundary over time, surpassing existing models in leveraging temporal domain information to extrapolate shifts far into the future and capture trends effectively.

Key contributions of this paper are:

1. We provide a novel perspective to learning under distribution shifts: Rather than specifying how exactly out-of-distribution data should be handled, we leverage in-context learning to learn an algorithm that handles such scenarios.

2. We propose a prior for temporal shifts, based on sparse, non-linear, and correlated mechanism shifts in structural causal models. This unifies shift types (covariate shift, prior probability shift, concept shift, and combinations thereof) and allows to extrapolate these changes.

3. We release Drift-Resilient TabPFN, a tabular data model for predicting under temporal distribution shifts, that automatically recognizes and adapts to shifts in the data and requires no hyperparameter tuning.

4. We extensively evaluate our models and state-of-the-art baselines on 18 synthetic and real-world datasets demonstrating improved out-of-distribution performance while requiring, on average, only 10.9s for training and prediction combined.

## 2 Background

### 2.1 Distribution Shift Settings

Consider $\mathcal{X}$, $\mathcal{Y}$, and $\mathcal{C}$, the sample spaces for features, labels, and domain indices, respectively. In these spaces, specific instances are represented by $\boldsymbol{x}$, $y$, and $\boldsymbol{c}$. The corresponding random variables for these instances are denoted as $X$, $Y$, and $C$.

A dataset is then a collection of $n$ tuples, denoted as $\mathcal{D} := \{(\boldsymbol{x}_i, y_i, \hat{\boldsymbol{c}}_k)\}_{i=1}^n$. Each tuple is thereby drawn from a conditional distribution $\mathbb{P}(X, Y \mid C = \boldsymbol{c}_k)$ over the spaces $\mathcal{X} \times \mathcal{Y}$. Here, $\boldsymbol{c}_k \in \mathcal{C}$ serves as a domain index that conditions the distribution from which the respective sample is drawn. Given that the true underlying temporal domain is mostly unknown in real-world data, it often is approximated and represented as $\hat{c}_k$ within the dataset. To isolate samples from a specific domain $\hat{\boldsymbol{c}}_k$, we introduce the sub-dataset $\mathcal{D}_{\hat{\boldsymbol{c}}_k}$, defined as $\mathcal{D}_{\hat{\boldsymbol{c}}_k} := \{(\boldsymbol{x}, y, \hat{\boldsymbol{c}}) \in \mathcal{D} | \hat{\boldsymbol{c}} = \hat{\boldsymbol{c}}_k\}$. This adapted notation allows for a more nuanced analysis of datasets with domain shifts, building upon the frameworks presented by Wang et al. [26] and Sheth et al. [27].

**Domain Generalization (DG).** DG aims to train a model that generalizes across a continuum of source domains $\mathcal{C}^{\text{train}}$ to the target domains $\mathcal{C}^{\text{test}}$ without access to samples of the target domains. The objective is to learn a mapping function $f : \mathcal{X} \times \mathcal{C} \to \mathcal{Y}$ that minimizes the expected loss when applied to new, previously unseen target domains.

**Temporal Domain Generalization (Temporal DG).** In temporal DG, a special case of DG, the domain index set $\mathcal{C}$ is one-dimensional and follows a total ordering, $c_1 \leq c_2 \leq \ldots$ [12]. In this framework, the training set is limited to source domains that precede target domains in this ordering, namely $\mathcal{C}^{\text{train}} = \{c_1, c_2, \ldots, c_t\}$. In this setting, the objective is to learn a predictive model $f$ that performs well on these source domains while also robustly generalizing to future, unseen domains $\mathcal{C}^{\text{test}} = \{c_{t+1}, c_{t+2}, \ldots, c_n\}$. The indices of all training domains $\mathcal{C}^{\text{train}}$ and the index of the current testing domain $c_k \in \mathcal{C}^{\text{test}}$ are provided to the model.

## 2.2 Related Work

While DG has drawn increasing attention in the research community [28–38], its temporal variant remains under-explored. We review existing DG benchmarks on tabular data, DG methods, the temporal DG benchmark Wild-Time [11], temporal DG methods, as well as relevant TabPFN studies.

**Tabular Distribution Shift Benchmarks.** Several benchmarks have been introduced to assess methods for tabular data under distribution shifts. (1) **Shifts** and **Shifts 2.0** [39, 40] are uncertainty focused benchmarks. **Shifts 2.0** includes five tasks, two of which involve tabular data subject to temporal and spatio-temporal shifts. (2) **WhyShift** [41], focusing on spatio-temporal shifts, offers five real-world tabular datasets, including the ACS dataset [42], which we also use in a subsampled form. Evaluating 22 methods like Gradient Boosted Decision Trees (GBDTs), MLPs, and robustness techniques, WhyShift finds that robustness methods do not consistently enhance out-of-distribution (OOD) performance. They also find that while GBDTs perform better, the gap between in-distribution (ID) and OOD performance persists, likely due to GBDTs being better fitted to the ID distribution. (3) **TableShift** [14] includes 15 tabular binary classification tasks, with 10 relevant to DG. Their work evaluates 19 model types, including several DG methods, finding that methods tailored for distribution shifts do not consistently outperform GBDTs. While generalization gaps can be slightly reduced, each robustness method comes at the cost of ID performance. (4) **Wild-Tab** [43] focuses on domain generalization within tabular regression using three large datasets. Their study compares 10 generalization techniques against standard Empirical Risk Minimization (ERM) applied to MLPs. Similar to previous work [44], they find that ERM was not consistently outperformed by specialized DG methods. Notably, no advantage of GBDTs over ERM on MLPs was observed in their datasets.

Unlike these benchmarks, which focus on large-scale datasets, our work addresses the overlooked challenges of small-scale temporal distribution shift datasets. However, their insights on generalization and robustness methods align with our findings, providing valuable context for our approach.

**DG Methods.** Several techniques have been proposed to improve robustness to distribution shifts in DG [45]. Key approaches include domain-invariant learning methods, such as **Deep CORAL** [35], **IRM** [31], and **DANN** [36], which aim to learn representations that generalize across domains. Data augmentation strategies, like **Mixup** [33] and **LISA** [34], contribute to generalization by generating synthetic data variations. Additionally, robust optimization techniques, including **VRex** [37], **GroupDRO** [32], **EQRM** [38], and **SWA** [46], aim to improve performance under distributional shifts by optimizing for worst-case scenarios or incorporating model uncertainty.

In relation to our work, data augmentation strategies are conceptually similar, as we also teach the model DG rather than adding invariances to the architecture. In contrast to augmentation, though, we learn to generalize using completely artificial data on the meta level rather than through manipulation of the target dataset. The other methods focus on designing models specifically for DG, while our approach completely relies on the model to learn to handle distribution shifts. While similar to continual meta-learning (CML) approaches, our method uses domain identifiers to generalize to unseen domains, whereas CML continuously adapts to evolving contexts without identifiers [47].

**Wild-Time Benchmark.** Wild-Time [11] is a benchmark of five datasets designed to study the real-world effects of temporal distribution shifts - an area largely overlooked by previous benchmarks. While Wild-Time primarily uses non-tabular data, it evaluates a wide array of techniques on its tabular dataset, including classical supervised learning (ERM), fine-tuning, and several previously mentioned general DG methods adapted to temporal distribution shifts. Despite the diversity of methods evaluated, Wild-Time reveals a significant performance gap between ID and OOD data, with none of the 13 tested methods consistently outperforming the standard ERM approach.

In our evaluation, we employ their evaluation strategy *Eval-Fix* and also benchmark our approach against the methods they considered. An in-depth overview of these methods is given in Appendix A.6.

**Temporal DG methods.** Recent specialized temporal DG methods include: (1) **DRAIN** [12], which employs a Bayesian framework alongside a recurrent neural network for predicting the dynamics of the model parameters across temporal domains. (2) **GI** [13], which explicitly incorporates the temporal domain as a feature, using a specialized Gradient Interpolation loss function, a time-sensitive activation, and enhanced domain reasoning via Time2Vec preprocessing [22]. However, both DRAIN and GI are limited in their ability to extrapolate beyond the near future, leading to their exclusion from our main evaluation. Notably, on the Rotated Two Moons Dataset - where DRAIN and GI have demonstrated their capabilities - our approach outperforms both. See Appendix A.4.3.4 for details.

**Recent TabPFN Studies.** To overcome the limitations of TabPFN in handling large samples and feature sets, typically found in DG benchmarks, several improvements have been proposed. Ma et al. [48] introduced a data distillation approach, where a distilled dataset serves as the model's context, optimized to maximize the training-data likelihood. Similarly, TuneTables [49], uses prompt-tuning to compress large datasets into a smaller context. Either of these methods could potentially be combined with Drift-Resilient TabPFN, as our modifications focus primarily on the pre-training phase, which remains unchanged in their approaches.

Another TabPFN variation, ForecastPFN [50], also introduces time dependence, but it does not consider any features. It only models simple time series data. Unlike our approach, which builds on TabPFN's SCM architecture for pre-training to handle a large set of features, ForecastPFN models synthetic time series using a single handcrafted function with sampled hyperparameters, simplifying the architecture while trading off diversity of the synthetic datasets.

## 3 Methodology

Our approach is built on PFNs, which use ICL to learn the learning algorithm itself. This approach also has a theoretical foundation as described by Müller et al. [18]: It can be viewed as approximating Bayesian prediction for a prior defined by the synthetic datasets. The trained PFN will approximate the posterior predictive distribution (PPD) and thus return a Bayesian prediction for the specified distribution over artificial datasets used during PFN training.

### 3.1 Structural Causal Model Prior

Hollmann et al. [19] introduce a prior based on Structural Causal Models (SCMs; 20; 21) to model complex feature dependencies and potential causal mechanisms underlying tabular data. To sample one dataset, this prior samples an SCM which is then used to sample the examples in the dataset. In this approach, each causal representation of a sampled SCM is converted into a functional representation to enable forward computation and dataset sampling.

Consider a sampled SCM graph $\mathcal{G} = (Z, R)$, where $Z = \{z_1, \ldots, z_n\}$ are the nodes (mechanisms) and $R = \{r_1, \ldots, r_m\}$ the edges (relationships). Each node is set using an individual assignment function $f_i$, $z_i = f_i(\{z_j \mid j \in \mathrm{PA}_i\}, \epsilon_i)$, with $\mathrm{PA}_i$ being the parent nodes of $z_i$ and $\epsilon_i$ random noise.

In the following, we will detail the underlying functional graph representation $\tilde{\mathcal{G}}$ of an SCM $\mathcal{G}$ sampled from the prior. We will later on apply distribution shifts on it. This representation is one level below the SCM and all nodes represent a single scalar value and all edges correspond to linear connections. The values of each node $v$ in $\tilde{\mathcal{G}}$ are set just like neurons in a neural network as a simple weighted sum and an activation function $v := h_j \left( \sum_{v' \in \mathrm{PA}_v} w_{i,j} v, \epsilon_j \right)$, where the scalar weights $w_{i,j}$ and the activation function $h_j$ are randomly chosen at graph creation and the noise term $\epsilon_j$ is sampled in each forward propagation to create a variety of samples within a dataset.

The functional graph representation $\tilde{\mathcal{G}}$ of an SCM $\mathcal{G}$ is defined as follows:

1. **Node Expansion.** Each node $z_i$ is expanded into a set of subnodes $Z_i := \{\tilde{z}_i^1, \ldots, \tilde{z}_i^k\}$, each representing a scalar value.

2. **Graph Expansion.** The following steps are performed for each node $z_j$ with a non-empty parent set $\mathrm{PA}_j$:

   (a) A new set of subnodes $F_j := \{\tilde{f}_j^1, \ldots, \tilde{f}_j^l\}$ is added.

   (b) We add the edges $E_j := \{Z_i \mid i \in PA_j\} \times F_j \cup F_j \times Z_j$.

Finally, the functional graph representation is defined as $\tilde{\mathcal{G}} := (\bigcup_{i \in \mathcal{I}} Z_i \cup \bigcup_{j \in \mathcal{I}} F_j, \bigcup_{i \in \mathcal{I}} E_i)$.

The features of our dataset are defined by a random subset $X \subseteq \bigcup_{i \in \mathcal{I}} Z_i$ and the target is defined by a randomly chosen $y \in \bigcup_{i \in \mathcal{I}} Z_i \backslash X$. Samples in a dataset are now generated by (i) sampling all noise terms $\epsilon_j$ in $\tilde{\mathcal{G}}$, (ii) propagating the values of all nodes through the graph, and finally (iii) retrieving values at the feature and target nodes. The resulting datasets, thus abide the computational flow of both $\mathcal{G}$ and $\tilde{\mathcal{G}}$. For a more intuitive understanding of this construction, refer to the simplified example provided in Figure 2.

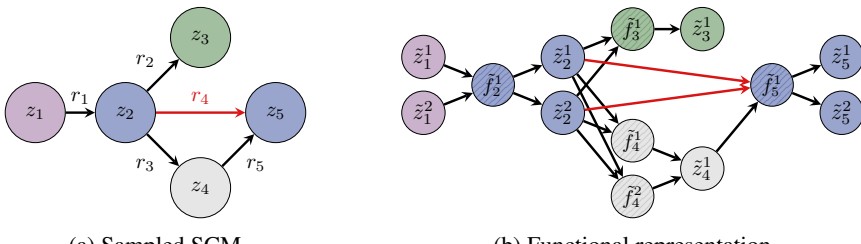

| (a) Sampled SCM | (b) Functional representation |

Figure 2: Illustrative transformation of an SCM to one exemplary functional representation. Shaded nodes indicate that their activations cannot be sampled. Feature nodes are blue, the target node is green, input/noise nodes are purple, and all others are gray. The figure also shows the mapping of shifted edges between a causal relationship and its functional form in red, ensuring that shifts specifically target the intended causal relationships without affecting others.

## 3.2 Inducing Temporal Robustness in SCMs

We extend TabPFN's prior to model distribution shifts, allowing the model to expand its posterior predictive distribution (PPD) calculations to incorporate temporal domain information. We propose modeling the dynamic of edge shifts using a $2^{nd}$-order SCM. This $2^{nd}$-order SCM is itself an SCM with feature nodes specifying the magnitude of edge shifts in the base SCM's functional graph.

Specifically, to extend the scope of the functional representation of the SCM $\tilde{\mathcal{G}}$, we introduce a version $\tilde{\mathcal{G}}_{c_k}$, depending on the temporal domain $c_k \in \mathcal{C}$. Our objective is to construct a time-dependent dataset $\mathcal{D}$ which is an aggregation of individual datasets $\mathcal{D}_{c_1}, \mathcal{D}_{c_2}, \ldots, \mathcal{D}_{c_n}$, where each dataset $\mathcal{D}_{c_k} := \{(\boldsymbol{x}_i, y_i, c_k)\}_{i=1}^{n_{c_k}}$ contains $n_{c_k}$ samples. To do this, we sample the temporal domains $\mathcal{C} = \{c_1, c_2, \ldots, c_n\}$ and for each domain $c_k$, we sample the number of samples $n_{c_k}$ it contains. An illustration of exemplary domain distributions across datasets is shown in Figure 14.

We then select a sparse subset of relationships in the causal representation of the SCM $\mathcal{G}$ to undergo temporal shifts based on the evidence that sparse shifts allow for causal reasoning [1]. For these selected edges, the $2^{nd}$-order SCM $\mathcal{H}$ is used to sample shift parameters that govern the corresponding edges in the functional representation $\tilde{\mathcal{G}}_{c_k}$. See Algorithm 1 for a high-level overview.

**Employing a $2^{nd}$-order SCM for Edge Shifting.** The $2^{nd}$-order SCM $\mathcal{H}$ takes temporal domains $\mathcal{C}$ as input and, through a single forward pass on the corresponding functional representation $\tilde{\mathcal{H}}$, produces dynamic edge shifts for each edge weight $w_{i,j}$ in $\tilde{\mathcal{G}}$ that corresponds to an edge in $\mathcal{G}$ that should be shifted. Note that although we perform a forward pass, there is no backward pass associated with it. Each $2^{nd}$-order SCM is randomly generated and used solely for sampling the weight shifts. This design allows for Bayesian reasoning over the edge shifts and enables $\tilde{\mathcal{H}}$ to generate complex, often correlated shifts over time. While we emphasize this construction does not reflect the true underlying dynamics of edge shifts in many real-world datasets, we use this heuristic prior construction for the following reasons: (a) Features in the SCM are correlated to each other in varying degrees, mimicking real-world processes of correlated changes in the underlying generating mechanisms of real-world data (b) An SCM with NN based causal mechanisms and nonlinear activation functions can extrapolate values outside of the data distribution, generalizing the underlying function to future domains (c) As demonstrated by TabPFN, a causal model prior is a sufficiently general approximation to many real-world datasets. We have visualized this approach in Figure 3 and a selection of the functions generated by a $2^{nd}$-order SCM in Figure 15.

**Shifting SCM edges can model various Types of Distribution Shifts.** A distribution shift is characterized by changing distributions between contexts ($\mathbb{P}(X, Y \mid C = \boldsymbol{c}_i) \neq \mathbb{P}(X, Y \mid C = \boldsymbol{c}_k)$). This definition can be further broken down into the following scenarios:

*Covariate Shift*: Changes in the feature distribution ($\mathbb{P}(X)$) while the conditional label distribution ($\mathbb{P}(Y|X)$) remains constant.

*Prior Probability Shift*: Changes in the label distribution ($\mathbb{P}(Y)$) while the feature distribution given the labels ($\mathbb{P}(X|Y)$) remains constant.

*Concept Shift*: Changes in the relationship between features and labels ($\mathbb{P}(Y|X)$ or $\mathbb{P}(X|Y)$) while the marginal distributions of features or labels remain constant.

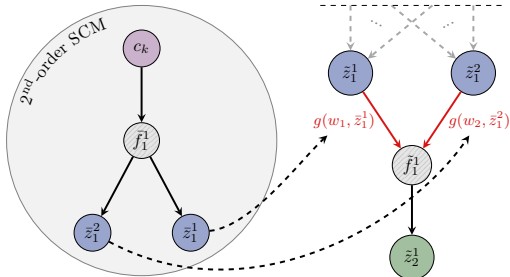

Figure 3: Diagram illustrating the integration of a 2nd-order SCM for adaptive edge shifting across evolving temporal domains. On the right, the primary network $\tilde{\mathcal{G}}$ generates data samples over multiple time domains, with red arrows indicating shifted edges. On the left, the 2nd-order SCM - an auxiliary network $\tilde{\mathcal{H}}$ - takes an input domain $c_k \in \mathcal{C}$ and outputs parameters to adaptively shift each edge weight $w_i$ in the base network.

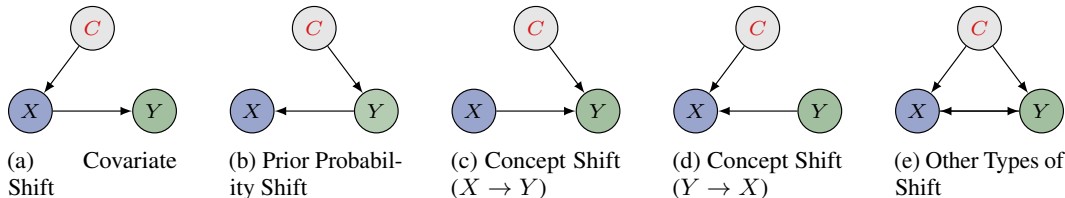

| (a) Covariate Shift | (b) Prior Probability Shift | (c) Concept Shift $(X \rightarrow Y)$ | (d) Concept Shift $(Y \rightarrow X)$ | (e) Other Types of Shift |

Figure 4: Types of distribution shifts based on the definitions by Moreno-Torres et al. [51] represented as Bayesian networks as defined by Kull and Flach [52]. Here $X$, $Y$, and $C$ denote the random variables of the features, label, and context, respectively. Note that all these types of shifts naturally arise in our prior, since we sample feature and target positions, as well as the locations of shifted edges, randomly at various positions in the synthetic datasets.

These shifts can also be viewed as Bayesian networks shown in Figure 4 - all arising in our prior by sampling features and targets at varying SCM positions. For further visualizations, refer to Figure 13.

**Encoding the Temporal Domain.** When encoded as inputs to our model, the temporally-dependent datasets $\mathcal{D} = \bigcup_{c_k \in \mathcal{C}} \mathcal{D}_{c_k}$ are partitioned at a particular instance into $\mathcal{D}^{\text{train}}$ and $\mathcal{D}^{\text{test}}$.

Temporal domains and features are normalized using only the training data, and so, to effectively encode temporal information and provide normalized inputs when projecting into the future, we use Time2Vec [22]. It converts each temporal domain index into an $m$-dimensional vector, using linear and sinusoidal functions characterized by learned parameters $\omega_i$ and $\varphi_i$. Specifically, the Time2Vec transformation for a temporal index $c_k \in \mathcal{C}$ is formulated as:

$$\mathbf{t2v}(c_k)[i] = \begin{cases} \omega_i c_k + \varphi_i, & \text{if } i = 0. \\ \sin(\omega_i c_k + \varphi_i), & \text{if } 1 \leq i < m. \end{cases} \tag{1}$$

## 4 Experiments

**Evaluation Strategy.** We evaluate analogous to the Eval-Fix setting outlined in Wild-Time [11], measuring both, ID and OOD performance. Here, each dataset $\mathcal{D}$ is split into three subsets: $\mathcal{D}^{\text{train}}$, $\mathcal{D}^{\text{ID}}$, and $\mathcal{D}^{\text{OOD}}$. Splits are based on a randomly sampled temporal domain $c_k$ that serves as the boundary between the train and test (OOD) portion. We only use such splits, where $\mathcal{D}^{\text{train}}$ comprises between 30% and 80% of the total domains and samples. To assess ID performance, we subsample 10% of the instances in each domain of $\mathcal{D}^{\text{train}}$ as the ID test set and the remainder as the train set. An illustration of the Eval-Fix strategy is provided in Figure 16.

Each class in the training set is required to be represented in both $\mathcal{D}^{\text{ID}}$ and $\mathcal{D}^{\text{OOD}}$, and vice versa. For all datasets, we generate three random splits and average metrics across these splits. We had to limit the number of splits to three due to the constrained number of available domains and the requirement for classes to be present in both the train and test splits. Each method is trained three times, and we report the average and 95%-confidence intervals calculated across model initializations.

**Metrics.**    We evaluate Accuracy, F1-Score (Harmonic mean of precision and recall, useful in imbalanced datasets), ROC AUC (Area under the receiver operating characteristic curve), and ECE (Expected Calibration Error; reflects the reliability of the model's probability outputs).

**Datasets.**    Our benchmark comprises 18 test datasets, 8 synthetic and 10 real-world. In addition, 12 validation datasets, 4 synthetic and the remaining 8 real-world, were used to optimize the hyper-parameters of our approach via random search. While some of these datasets have been analyzed in previous work, there has been no comprehensive benchmark focusing on small tabular datasets undergoing distribution shifts. To address this gap, we carefully selected and generated a diverse range of datasets that exhibit temporal distribution shifts. The selected datasets originate from open dataset platforms or previous work in DG. Ground truth domain indices $c_k \in \mathcal{C}$ are known for synthetic datasets. For real-world datasets, we approximated domain indices $\hat{c}_k$ based on features that encode temporal information, which we transformed into discrete intervals. Also, some real-world datasets required subsampling due to their large size, which was beyond the current architecture of TabPFN. We provide full details, including descriptions for each dataset and pre-processing steps in Section A.7 of the Appendix.

**Baseline Setup.**    Our baselines include state-of-the-art methods for tabular prediction. These include advanced GBDTs like CatBoost [24], XGBoost [23], and LightGBM [25], which have demonstrated superior performance to standard neural network approaches in handling tabular data [17]. We also include TabPFN in its unmodified form (TabPFN$_\text{base}$; 19). Methods from the Wild-Time benchmark are examined separately and detailed in Section A.4.4.3 of the Appendix. However, we have added the two best-performing Wild-Time methods, ERM and Stochastic Weight Averaging (SWA), to the main results table. All baseline methods besides TabPFN are subject to a time budget of 1,200 seconds on 8 CPUs and 1 GPU. For each method except TabPFN, which does not require tuning, a random hyperparameter search with 3-fold time series cross-validation was used. We chose the best-performing hyperparameters based on OOD ROC AUC within the allocated time.

Among our baselines, we considered three strategies:

1. Providing the full dataset $\mathcal{D}^\text{train}$ along with the corresponding domain indices $\mathcal{C}^\text{train}$ as a feature, aiming to allow for better reasoning of the shifts in the dataset (all dom. w. ind.).

2. Using the dataset without domain indices $\mathcal{D}^\text{train} = \{(\boldsymbol{x}_i^\text{train}, y_i^\text{train})\}_{i=1}^n$ (all dom. wo. ind.).

3. Limiting the training set to samples from the last training domain $c_t$. In this setting, we also omit the corresponding domain indices, resulting in the set $\mathcal{D}_{\hat{c}_t}^\text{train} = \{(\boldsymbol{x}_i^\text{train}, y_i^\text{train})\}_{i=1}^{n_{c_t}}$ (last dom. wo. ind.). The rationale behind the last scenario is to provide only training data closest to the subsequent test distribution. This strategy is not used for distribution shift baselines.

**TabPFN Setup.**    For the TabPFN variants, both the original and our modified method (TabPFN$_\text{dist}$) were pre-trained for 30 epochs across 8 GPUs. This results in a total of 30,720,000 synthetically generated datasets processed during pre-training. While this pre-training step is moderately expensive, it is done offline, in advance, and only once as part of our algorithm development. Furthermore, the preprocessing parameters of both methods were optimized once on the validation datasets by random search over 300 configurations. We chose the configurations that yielded the best OOD ROC AUC performance. The resulting model and hyperparameters are used for all datasets, resulting in, on average, 110 times faster training and prediction time on our benchmark.

**Quantitative Evaluation.**    Our method demonstrates superior predictive performance across all metrics for OOD data in 18 test datasets, for synthetic datasets and real-world datasets, as detailed in Table 1. Compared to the strongest baseline, our method improves accuracy from 0.665 to 0.754 on synthetic datasets and from 0.712 to 0.736 on real-world datasets. It also enhances the F1 score from 0.588 to 0.697 on synthetic datasets and from 0.668 to 0.682 on real-world datasets. Additionally, it increases the ROC AUC from 0.749 to 0.844 on synthetic datasets and from 0.82 to 0.822 on real-world datasets. Furthermore, our method shows much stronger calibration on OOD samples, reducing ECE from 0.164 to 0.126 on synthetic datasets and from 0.083 to 0.062 on real-world datasets. While baselines are often overconfident on OOD data, our method is able to predict uncertainty accurately. Compared to TabPFN and GBDTs, the NN-based methods in the Wild-Time Benchmark (Section A.4.4.3) show a substantial drop in performance, likely due to the limited training data. Since our method focuses on enhancing OOD robustness rather than optimizing ID tasks, we find lower predictive performance on ID tasks. While performance gains are observed on both, real-world and synthetic data, we observe stronger improvements on synthetic datasets. This can be partly attributed

to the, on average, stronger distribution shifts between ID and OOD data in our synthetic benchmark. Furthermore, real-world datasets often show multifaceted and complex shifts that are much more difficult to extrapolate into the future. Combined results across all datasets are provided in Table 5.

Table 1: Comparison of Drift-Resilient TabPFN with various baselines and settings across the subsets of **synthetic** and **real-world** datasets. Metrics include accuracy, F1, ROC, and ECE for both in-distribution (ID) and out-of-distribution (OOD) data, averaged over three initializations and reported with 95% confidence intervals. The best mean of each metric within a dataset subset is marked in bold. Metric arrows indicate optimization direction.

| Model | Variant | Acc. ↑ OOD | ID | F1 ↑ OOD | ID | ROC ↑ OOD | ID | ECE ↓ OOD | ID |
|---|---|---|---|---|---|---|---|---|---|
| **SYNTHETIC** | | | | | | | | | |
| $\text{TabPFN}_{\text{dist}}$ | all dom. w. ind. | $\mathbf{0.754}_{.032}$ | $0.959_{.011}$ | $\mathbf{0.697}_{.048}$ | $0.935_{.033}$ | $\mathbf{0.844}_{.03}$ | $0.987_{.002}$ | $\mathbf{0.126}_{.018}$ | $0.038_{.003}$ |
| $\text{TabPFN}_{\text{base}}$ | all dom. w. ind. | $0.658_{.018}$ | $\mathbf{0.963}_{.006}$ | $0.567_{.015}$ | $0.935_{.02}$ | $0.749_{.017}$ | $0.986_{.007}$ | $0.164_{.014}$ | $\mathbf{0.029}_{.003}$ |
| | all dom. wo. ind. | $0.571_{.014}$ | $0.901_{.006}$ | $0.467_{.016}$ | $0.848_{.009}$ | $0.631_{.01}$ | $0.955_{.003}$ | $0.322_{.016}$ | $0.053_{.005}$ |
| | last dom. wo. ind. | $0.651_{.002}$ | $0.939_{.015}$ | $0.574_{.002}$ | $0.918_{.031}$ | $0.727_{.006}$ | $0.975_{.001}$ | $0.27_{.011}$ | $0.066_{.001}$ |
| CatBoost | all dom. w. ind. | $0.665_{.008}$ | $0.958_{.003}$ | $0.588_{.008}$ | $\mathbf{0.94}_{.009}$ | $0.73_{.01}$ | $\mathbf{0.989}_{.002}$ | $0.297_{.006}$ | $0.037_{.006}$ |
| | all dom. wo. ind. | $0.575_{.006}$ | $0.885_{.003}$ | $0.476_{.008}$ | $0.831_{.002}$ | $0.613_{.013}$ | $0.942_{.007}$ | $0.325_{.006}$ | $0.063_{.014}$ |
| | last dom. wo. ind. | $0.639_{.004}$ | $0.932_{.017}$ | $0.564_{.005}$ | $0.916_{.021}$ | $0.684_{.005}$ | $0.962_{.012}$ | $0.301_{.019}$ | $0.065_{.006}$ |
| XGBoost | all dom. w. ind. | $0.645_{.018}$ | $0.936_{.012}$ | $0.57_{.019}$ | $0.931_{.005}$ | $0.705_{.011}$ | $0.968_{.02}$ | $0.253_{.032}$ | $0.075_{.013}$ |
| | all dom. wo. ind. | $0.582_{.07}$ | $0.872_{.031}$ | $0.48_{.074}$ | $0.818_{.039}$ | $0.621_{.06}$ | $0.926_{.035}$ | $0.245_{.088}$ | $0.097_{.034}$ |
| | last dom. wo. ind. | $0.645_{.008}$ | $0.916_{.009}$ | $0.565_{.009}$ | $0.88_{.019}$ | $0.688_{.017}$ | $0.956_{.012}$ | $0.256_{.018}$ | $0.111_{.003}$ |
| LightGBM | all dom. w. ind. | $0.646_{.016}$ | $0.943_{.007}$ | $0.57_{.015}$ | $0.927_{.015}$ | $0.687_{.013}$ | $0.982_{.002}$ | $0.281_{.008}$ | $0.056_{.011}$ |
| | all dom. wo. ind. | $0.581_{.01}$ | $0.884_{.005}$ | $0.482_{.007}$ | $0.829_{.005}$ | $0.617_{.016}$ | $0.935_{.001}$ | $0.273_{.01}$ | $0.069_{.017}$ |
| | last dom. wo. ind. | $0.629_{.004}$ | $0.917_{.003}$ | $0.553_{.006}$ | $0.892_{.018}$ | $0.662_{.005}$ | $0.958_{.007}$ | $0.288_{.012}$ | $0.077_{.007}$ |
| Wild-Time ERM | all dom. w. ind. | $0.648_{.046}$ | $0.945_{.017}$ | $0.489_{.092}$ | $0.906_{.026}$ | $0.65_{.042}$ | $0.973_{.021}$ | $0.304_{.038}$ | $0.041_{.006}$ |
| | all dom. wo. ind. | $0.576_{.021}$ | $0.885_{.04}$ | $0.487_{.035}$ | $0.837_{.049}$ | $0.621_{.03}$ | $0.943_{.015}$ | $0.282_{.012}$ | $0.058_{.024}$ |
| | last dom. wo. ind. | $0.632_{.006}$ | $0.921_{.017}$ | $0.566_{.007}$ | $0.9_{.035}$ | $0.688_{.01}$ | $0.962_{.005}$ | $0.282_{.018}$ | $0.094_{.016}$ |
| Wild-Time SWA | all dom. w. ind. | $0.636_{.05}$ | $0.923_{.034}$ | $0.489_{.088}$ | $0.877_{.06}$ | $0.651_{.035}$ | $0.958_{.03}$ | $0.313_{.046}$ | $0.054_{.015}$ |
| | all dom. wo. ind. | $0.573_{.032}$ | $0.887_{.019}$ | $0.48_{.042}$ | $0.837_{.017}$ | $0.631_{.043}$ | $0.943_{.012}$ | $0.3_{.055}$ | $0.059_{.002}$ |
| **REAL-WORLD** | | | | | | | | | |
| $\text{TabPFN}_{\text{dist}}$ | all dom. w. ind. | $\mathbf{0.736}_{.007}$ | $0.814_{.014}$ | $\mathbf{0.682}_{.012}$ | $0.759_{.015}$ | $\mathbf{0.822}_{.01}$ | $0.887_{.006}$ | $\mathbf{0.062}_{.004}$ | $0.103_{.026}$ |
| $\text{TabPFN}_{\text{base}}$ | all dom. w. ind. | $0.712_{.012}$ | $\mathbf{0.822}_{.013}$ | $0.661_{.022}$ | $\mathbf{0.777}_{.018}$ | $0.816_{.006}$ | $\mathbf{0.894}_{.013}$ | $0.083_{.001}$ | $0.097_{.007}$ |
| | all dom. wo. ind. | $0.704_{.01}$ | $0.813_{.023}$ | $0.668_{.014}$ | $0.764_{.03}$ | $0.82_{.006}$ | $0.882_{.011}$ | $0.106_{.019}$ | $\mathbf{0.095}_{.011}$ |
| | last dom. wo. ind. | $0.685_{.008}$ | $0.809_{.016}$ | $0.637_{.005}$ | $0.746_{.036}$ | $0.787_{.008}$ | $0.867_{.036}$ | $0.109_{.005}$ | $0.177_{.012}$ |
| CatBoost | all dom. w. ind. | $0.687_{.005}$ | $0.807_{.012}$ | $0.646_{.003}$ | $0.753_{.017}$ | $0.796_{.004}$ | $0.862_{.019}$ | $0.161_{.016}$ | $0.122_{.019}$ |
| | all dom. wo. ind. | $0.677_{.008}$ | $0.797_{.025}$ | $0.642_{.005}$ | $0.741_{.015}$ | $0.794_{.011}$ | $0.856_{.022}$ | $0.172_{.032}$ | $0.125_{.037}$ |
| | last dom. wo. ind. | $0.671_{.002}$ | $0.788_{.023}$ | $0.627_{.004}$ | $0.728_{.05}$ | $0.752_{.007}$ | $0.863_{.022}$ | $0.221_{.017}$ | $0.188_{.023}$ |
| XGBoost | all dom. w. ind. | $0.68_{.008}$ | $0.797_{.011}$ | $0.642_{.01}$ | $0.745_{.004}$ | $0.793_{.01}$ | $0.845_{.02}$ | $0.147_{.027}$ | $0.141_{.027}$ |
| | all dom. wo. ind. | $0.674_{.025}$ | $0.798_{.019}$ | $0.639_{.004}$ | $0.746_{.026}$ | $0.795_{.011}$ | $0.845_{.024}$ | $0.153_{.028}$ | $0.138_{.049}$ |
| | last dom. wo. ind. | $0.679_{.014}$ | $0.75_{.041}$ | $0.626_{.034}$ | $0.66_{.109}$ | $0.769_{.008}$ | $0.833_{.022}$ | $0.154_{.028}$ | $0.212_{.021}$ |
| LightGBM | all dom. w. ind. | $0.654_{.015}$ | $0.761_{.042}$ | $0.614_{.01}$ | $0.698_{.031}$ | $0.778_{.008}$ | $0.848_{.013}$ | $0.133_{.017}$ | $0.123_{.02}$ |
| | all dom. wo. ind. | $0.66_{.029}$ | $0.79_{.031}$ | $0.624_{.028}$ | $0.728_{.029}$ | $0.778_{.003}$ | $0.849_{.018}$ | $0.14_{.009}$ | $0.12_{.018}$ |
| | last dom. wo. ind. | $0.629_{.036}$ | $0.701_{.019}$ | $0.533_{.055}$ | $0.582_{.046}$ | $0.706_{.008}$ | $0.768_{.03}$ | $0.172_{.014}$ | $0.206_{.025}$ |
| Wild-Time ERM | all dom. w. ind. | $0.61_{.037}$ | $0.721_{.056}$ | $0.553_{.046}$ | $0.664_{.059}$ | $0.719_{.017}$ | $0.8_{.042}$ | $0.23_{.07}$ | $0.161_{.035}$ |
| | all dom. wo. ind. | $0.587_{.033}$ | $0.737_{.014}$ | $0.545_{.028}$ | $0.673_{.013}$ | $0.714_{.012}$ | $0.798_{.024}$ | $0.233_{.043}$ | $0.14_{.008}$ |
| | last dom. wo. ind. | $0.551_{.041}$ | $0.674_{.045}$ | $0.498_{.067}$ | $0.612_{.03}$ | $0.648_{.039}$ | $0.749_{.037}$ | $0.355_{.061}$ | $0.267_{.034}$ |
| Wild-Time SWA | all dom. w. ind. | $0.62_{.049}$ | $0.745_{.028}$ | $0.57_{.055}$ | $0.697_{.046}$ | $0.712_{.039}$ | $0.805_{.007}$ | $0.242_{.059}$ | $0.155_{.021}$ |
| | all dom. wo. ind. | $0.6_{.019}$ | $0.733_{.017}$ | $0.57_{.019}$ | $0.677_{.004}$ | $0.721_{.029}$ | $0.798_{.008}$ | $0.232_{.021}$ | $0.15_{.017}$ |

**Qualitative Analysis.** Next, we take an in-depth look at the predictions made by our method. Figure 5 illustrates the decision boundaries of our method and TabPFN$_{\text{base}}$ on the synthetic Intersecting Blobs dataset. In this evaluation, we restrict the training domains to $\mathcal{C}^{\text{train}} = \{0, 1, 2, 3\}$ and aim to predict samples in test domains $\mathcal{C}^{\text{test}} = \{4, 5, 6\}$ without adding additional data to the training set. This setup requires the model to extrapolate the temporal shifts into the future based solely on existing training data. In this setting, our model accurately extrapolates decision boundaries to future domains, while TabPFN$_{\text{base}}$ tends to retain its initial boundary. Our analysis reveals two key attributes of our model: (i) The model decreases prediction certainty over time, improving calibration. (ii) Our model adjusts the decision boundary dynamically, boosting accuracy. Further visualizations, including decision boundaries for the Rotated Two Moons dataset, are available in Figure 7. Likewise, plots illustrating the overall shifts in these datasets are provided in Figure 6.

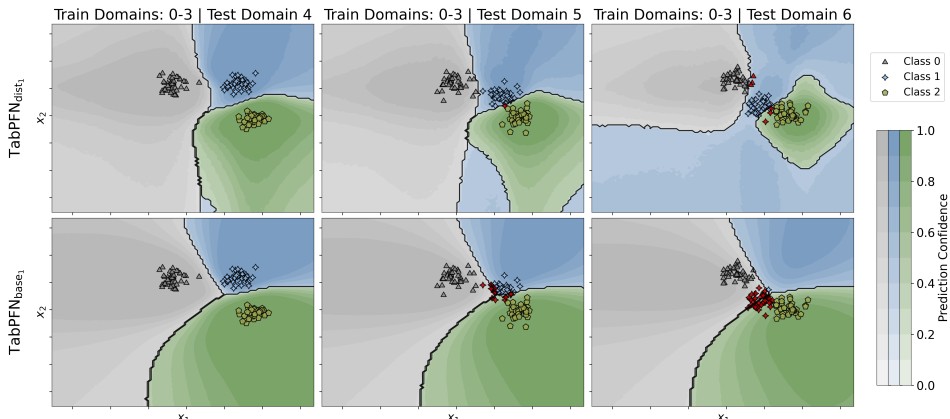

Figure 5: This figure displays the predictive behavior of TabPFN$_{dist}$ in the top row and TabPFN$_{base}$ in the bottom row on the Intersecting Blobs dataset. It illustrates how each model adapts to unseen test domains when trained on domains $\mathcal{C}^{train} = \{0, 1, 2, 3\}$. The baseline is given the domain indices as a feature in train and test. The coloring indicates the probability of the most likely class at each point. Incorrectly classified samples are highlighted in red.

**Impact of Time2Vec Preprocessing on Model Performance.** To examine Time2Vec's contributions to improved OOD performance, we conduct an ablation study in Appendix A.4.1. The ablation reveals that while Time2Vec provides at most slight improvements, the substantial performance gains are to be attributed to the prior construction used during the model's pre-training phase.

# 5 Conclusions & Limitations

In this work, we presented a Bayesian approach to address the issue of temporal domain generalization in tabular data. Specifically, we focused on enhancing TabPFN to improve its robustness to temporal distribution shifts. Within this framework, we introduced a novel approach that changes the causal relationships in the SCM prior over time, thereby enabling TabPFN to inherently adapt to these shifts. Our method outperforms all baselines on the evaluated datasets and demonstrates notable improvements both qualitatively and quantitatively, particularly on synthetic OOD datasets. Furthermore, it requires no hyperparameter tuning, is not limited to particular types of distribution shifts and takes only 10.9s for training and prediction combined.

Despite these advancements, our methodology inherits certain limitations from the underlying TabPFN model. (1) Due to the quadratic scaling of the attention mechanism with respect to the number of samples, our method does not scale to large datasets. Here, our research will benefit from the continued improvements of TabPFN, ICL, and sequence-based models in general. (2) The TabPFN, like many transformer-based models, acts as a "black box", making it challenging to interpret the model's predictions and understand the recognized distribution shifts. (3) The underlying prior for structural causal models with sparse mechanism shifts may not accurately describe all real-world datasets. While we find it to be empirically and intuitively useful, real-world shifts might have underlying complexities that our prior currently does not capture.

For future work, next to addressing the existing limitations, we have identified in initial experiments promise in extending our model to (1) transductive and (2) online continual learning settings. Also, our prior could be adapted to support (3) spatial or spatio-temporal distribution shifts. Employing a prior that (4) models shifts in the underlying causal model could improve the robustness of the baseline TabPFN in standard classification tasks where temporal shifts are often implicit.

We provide code, pre-trained models, and a Colab notebook at `https://github.com/automl/Drift-Resilient_TabPFN`. We further describe reproducibility, the release of code and models, the broader impact of our models, and the computational resources used for method development in Appendix A.1, A.2 and A.3. We add a discussion of baselines in A.6, an in-detail discussion of the evaluated datasets in A.7, and additional quantitative and qualitative evaluations in A.4.

# 6 Acknowledgments

Frank Hutter acknowledges the financial support of the Hector Foundation. This research was funded by the Deutsche Forschungsgemeinschaft (DFG, German Research Foundation) under grant number 417962828.

We acknowledge funding by the European Union (via ERC Consolidator Grant DeepLearning 2.0, grant no. 101045765). Views and opinions expressed are however those of the author(s) only and do not necessarily reflect those of the European Union or the European Research Council. Neither the European Union nor the granting authority can be held responsible for them.

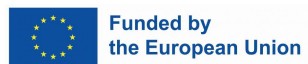

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

# Table of Contents

# A   Appendix

## A.1   Reproducibility

**Code Availability.** In an effort to ensure reproducibility, we release code, version specification of our baselines, our pre-trained Drift-Resilient TabPFN and an interactive Colab notebook, that lets you interact with our scikit-learn interface, at `https://github.com/automl/Drift-Resilient_TabPFN`.

**Data Availability.** All real-world datasets used in our experiments are freely available at `OpenML.org` [53] or via the original sources referenced in Section A.7, with downloading scripts or instructions

included in the submission code. Code to generate our synthetic datasets can be found in our code repository, see above.

**Details of Hyperparameters for Drift-Resilient TabPFN and Baselines.** An overview of the hyperparameters used for running Drift-Resilient TabPFN and our baselines can be found in Tables 8, 9 and 10 respectively.

**Evaluation Reproducibility.** Consistent dataset splits were pre-determined to ensure comparability across all evaluations, reinforcing the reliability of our findings.

## A.2 Broader Impacts

As a general method for handling distribution shift in tabular data, Drift-Resilient TabPFN does not have immediate direct societal implications in the same way as AI systems designed to automate specific tasks or replace human jobs. However, Drift-Resilient TabPFN could offer several potential positive societal impact:

1. **Increased Model Longevity**: Our approach extends the usable lifespan of deployed ML models by adapting to distribution shifts, reducing the need for retraining and leading to cost savings and more stable performance. In healthcare, this can ensure diagnostic and prognostic models remain reliable as data shifts over time.

2. **Improved Decision Making and Long-Term Predictions**: Drift-Resilient TabPFN enhances decision-making in critical fields like finance and climate science by enabling more robust, longer-term predictions. Our Bayesian approach for tackling distribution shift provides a new perspective that can spur further methodological innovations.

However, we also recognize potential negative impacts:

1. **Potential Misuse**: Like any ML advance, more robust models could be misused for harmful purposes if not developed and deployed responsibly.

2. **Environmental Cost**: While the trained model is now efficiently applicable to various datasets, the considerable computational resources used for initial training should be noted. However, we note that these costs are one-time, while the resulting model can be applied with minimal energy usage.

## A.3 Computational Resources

In the course of our research on Drift-Resilient TabPFN, we employed substantial computational resources across various stages of model development, training, and evaluation. The computation specifics are as follows:

1. **Infrastructure:** The experiments were conducted on an internal SLURM cluster equipped with RTX 2080 TI GPUs and CPUs of type AMD EPYC 7502, 32C/64T, @ 2.50-3.35GHz.

2. **Baseline Experiments:** Each baseline experiment utilized 8 CPUs, 1 GPU, and 62.5 GB RAM, with a hyperparameter optimization (HPO) runtime of 1200 seconds per dataset per split, repeated three times to ensure reliability.

3. **Pre-training for Drift-Resilient TabPFN and TabPFN-base:** These models were pre-trained three times each, requiring 64 CPUs, 8 GPUs, and 500 GB RAM, requiring approximately 7 and 8 days respectively.

4. **Hyperparameter Optimization for Drift-Resilient TabPFN and TabPFN-base:** We used 32 CPUs, 4 GPUs, and 250 GB RAM, running approximately 40 configurations and taking about one day per pre-training session for optimizing the hyperparameters of the novel prior-data generating mechanism of Drift-Resilient TabPFN. Both TabPFN-base and Drift-Resilient TabPFN underwent preprocessing optimization that utilized 8 CPUs, 1 GPU, and 62.5 GB RAM across 300 runs, each lasting between 0.5 to 1 hour.

Additional computational resources were allocated for method development tests and other experimental setups not detailed in the final publication. Thus the full scope of the research required more computational resources than those detailed above due to these preliminary and unreported experiments.

### A.4 Additional Experiments

### A.4.1 Ablation Studies

*Is our model's performance mostly based on Time2Vec preprocessing?* To address this question, we conducted an ablation study where we trained a model with temporal domain indices normalized but not subjected to Time2Vec preprocessing (No T2V).

Table 2 presents the performance metrics of Drift-Resilient TabPFN, TabPFN-base, and our ablation model. The results indicate that while Time2Vec preprocessing may slightly improve model performance, it is statistically insignificant. Rather, the substantial performance improvements are largely due to our prior construction, used during the pre-training phase of the model. The decision to keep Time2Vec in the final model was guided by our HPO, which indicated a positive impact on average performance.

Table 2: Comparison of Drift-Resilient TabPFN with respect to the stated ablations. Metrics include ROC AUC and accuracy for both in-distribution (ID) and out-of-distribution (OOD) data.

| Model | Variant | Acc. ↑ | | F1 ↑ | | ROC ↑ | | ECE ↓ | |
|---|---|---|---|---|---|---|---|---|---|
| | | OOD | ID | OOD | ID | OOD | ID | OOD | ID |
| **TabPFN$_{dist}$** | all dom. w. ind. | **0.744** $_{.018}$ | 0.879 $_{.012}$ | **0.689** $_{.028}$ | 0.837 $_{.022}$ | **0.832** $_{.018}$ | 0.932 $_{.002}$ | **0.091** $_{.006}$ | 0.074 $_{.014}$ |
| No T2V | all dom. w. ind. | 0.742 $_{.004}$ | 0.877 $_{.007}$ | 0.685 $_{.002}$ | 0.834 $_{.014}$ | **0.832** $_{.004}$ | 0.931 $_{.009}$ | 0.093 $_{.009}$ | 0.071 $_{.005}$ |
| | all dom. w. ind. | 0.688 $_{.01}$ | **0.885** $_{.01}$ | 0.62 $_{.012}$ | **0.847** $_{.017}$ | 0.786 $_{.007}$ | **0.935** $_{.01}$ | 0.119 $_{.006}$ | **0.067** $_{.005}$ |
| **TabPFN$_{base}$** | all dom. wo. ind. | 0.645 $_{.011}$ | 0.852 $_{.016}$ | 0.579 $_{.014}$ | 0.801 $_{.02}$ | 0.736 $_{.001}$ | 0.914 $_{.007}$ | 0.202 $_{.011}$ | 0.076 $_{.007}$ |
| | last dom. wo. ind. | 0.67 $_{.005}$ | 0.867 $_{.004}$ | 0.609 $_{.004}$ | 0.823 $_{.011}$ | 0.76 $_{.003}$ | 0.915 $_{.019}$ | 0.181 $_{.003}$ | 0.128 $_{.007}$ |

### A.4.2 Perturbation of Temporal Domain Indices within the Prior

This additional experiment analyzes the impact of perturbing temporal domain indices on the performance of our Drift-Resilient TabPFN model. In real-world datasets, ground truth temporal domain information is often unknown and must be approximated, creating a crucial difference between these datasets and those generated by the prior in our methodology. To assess this, we conducted an experiment during model development wherein we intentionally modified the ground truth domain indices $\mathcal{C}$ during the pre-training phase prior to feeding them into the transformer.

In this context, we explored three primary techniques:

**Shifting Domain Boundaries Probabilistically.** We shifted the boundaries of domains and reported instances near those boundaries as belonging to adjacent domains.

**Merging Domains.** Multiple domains were probabilistically merged into a single domain, thereby introducing ambiguity in domain information.

**Noise Injection.** Random noise was added to each domain indicator, further complicating reasoning about temporal distribution shifts based on these indicators.

Our experiments on our validation datasets, listed in Table 3, show that overall these perturbations adversely affect model performance. The findings suggest that the model requires ground-truth domain indices for effective training, emphasizing the importance of accurate domain information in real-world applications.

Table 3: Performance evaluation of Drift-Resilient TabPFN against models with perturbed domain information. Three techniques were examined: (1) Shifting domain boundaries probabilistically, reporting instances near those boundaries as belonging to the other domain; (2) Probabilistically merging multiple domains into one; (3) Adding noise to each domain indicator. Our results indicate that perturbed domain indices overall led to a decline in model performance, emphasizing the model's requirement for ground-truth domain indices during training.

| Model | Variant | Acc. ↑ | | F1 ↑ | | ROC ↑ | | ECE ↓ | |
|---|---|---|---|---|---|---|---|---|---|
| | | OOD | ID | OOD | ID | OOD | ID | OOD | ID |
| **TabPFN$_{dist}$** | no changes | 0.74 | **0.867** | **0.692** | **0.832** | **0.837** | **0.908** | **0.085** | **0.076** |
| | alt dom. bound. | 0.734 | 0.86 | 0.682 | 0.822 | 0.829 | 0.905 | 0.092 | 0.078 |
| **TabPFN$_{pert. dom.}$** | alt dom. bound. / merge dom. | **0.742** | 0.857 | 0.69 | 0.815 | 0.833 | 0.906 | 0.09 | 0.079 |
| | alt dom. bound. / merge dom. / noise | 0.704 | 0.835 | 0.64 | 0.78 | 0.792 | 0.888 | 0.101 | 0.108 |

### A.4.3 Qualitative Analysis

**A.4.3.1 Overview of the Shifts in the Datasets Analyzed** This section offers plots of the Intersecting Blobs and Rotated Two Moons datasets across specific temporal domains. The Intersecting Blobs dataset is visualized in Figure 6a for domains $\mathcal{C} = \{0, 4, 8, 13\}$. The Rotated Two Moons dataset is presented in Figure 6b for domains $\mathcal{C} = \{0, 3, 6, 9\}$.

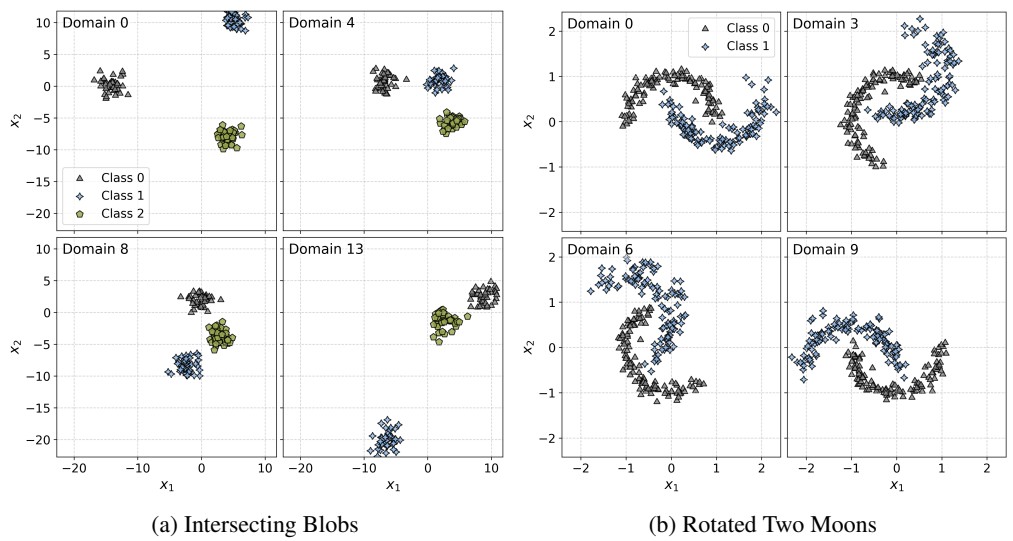

(a) Intersecting Blobs                    (b) Rotated Two Moons

Figure 6: This figure shows the temporal shifts of two synthetic datasets across selected domains.

**A.4.3.2 Decision Boundaries on Rotated Two Moons Dataset** In addition to the qualitative analysis conducted for the Intersecting Blobs dataset in the main text of this work, this subsection provides additional illustrations of the Rotated Two Moons dataset. The corresponding visualizations for our approach as well as the TabPFN baseline are provided in Figure 7.

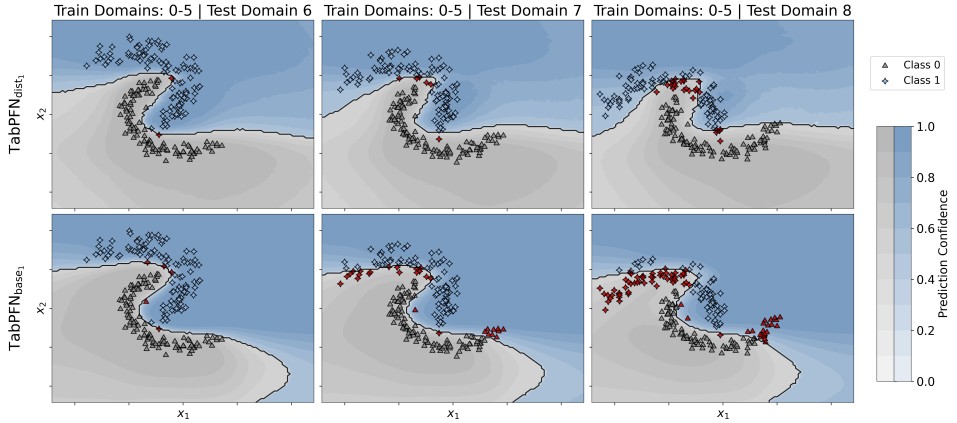

Figure 7: This figure contrasts the predictive behavior of $\text{TabPFN}_{\text{dist}}$ and $\text{TabPFN}_{\text{base}}$ on the Rotated Two Moons dataset. It illustrates how each model adapts to different testing domains when trained on domains $\mathcal{C}^{\text{train}} = \{0, 1, 2, 3, 4, 5\}$. The color shading indicates the maximum class probability at each point, with decision boundaries shown when this probability exceeds 50%. Incorrectly classified samples are highlighted in red.

**A.4.3.3 Decision Boundaries by Type of Shift** To analyze our models behavior under distinct shift types, we separate decision boundaries by shift category to provide insights into how the model adapts to each.

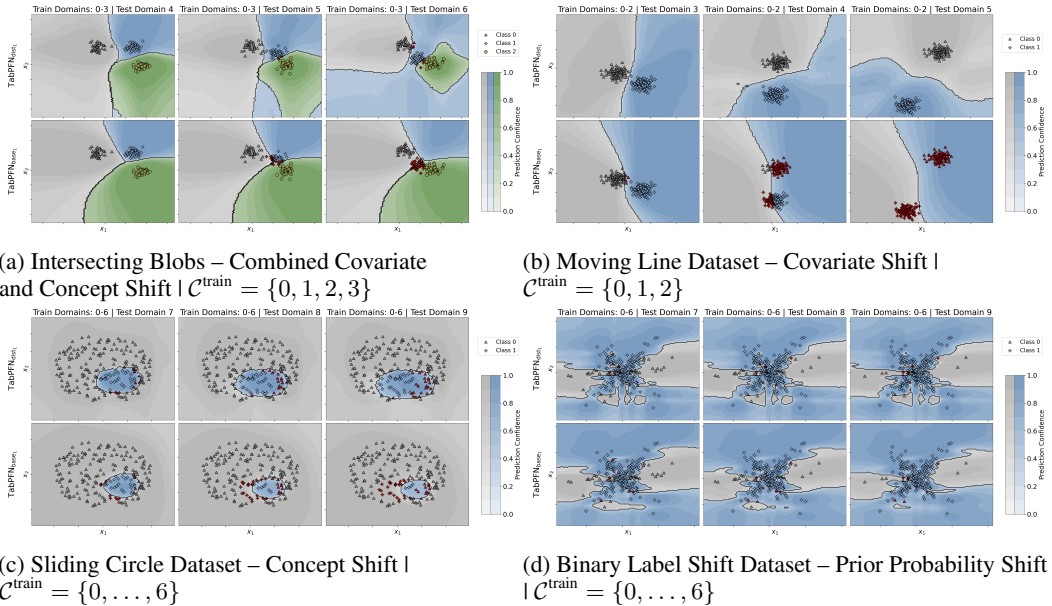

(a) Intersecting Blobs – Combined Covariate and Concept Shift | $\mathcal{C}^{\text{train}} = \{0, 1, 2, 3\}$

(b) Moving Line Dataset – Covariate Shift | $\mathcal{C}^{\text{train}} = \{0, 1, 2\}$

(c) Sliding Circle Dataset – Concept Shift | $\mathcal{C}^{\text{train}} = \{0, \ldots, 6\}$

(d) Binary Label Shift Dataset – Prior Probability Shift | $\mathcal{C}^{\text{train}} = \{0, \ldots, 6\}$

Figure 8: Each figure displays the predictive behavior of TabPFN$_{\text{dist}}$ in the top row and TabPFN$_{\text{base}}$ in the bottom row. It illustrates how each model adapts to unseen test domains when trained on domains $\mathcal{C}^{\text{train}}$. The baseline is given the domain indices as a feature in train and test. The coloring indicates the probability of the most likely class at each point. Incorrectly classified samples are highlighted in red.

**A.4.3.4 Comparison Against DRAIN and GI** To show our improved performance compared to the state-of-the-art methods DRAIN [12] and GI [13], we compare our method with the qualitative analysis performed by the authors of DRAIN on the Rotated Two Moons dataset. In this setting, all domains except the last are used for training, with the final domain reserved for testing. We illustrate our method's decision boundary compared to those provided by DRAIN in Figure 9. The accuracy results are listed in Table 4. As the contour levels are unknown, we display only the pure decision boundary for clearer comparison. Our analysis shows that our model forecasts the rotation of the two moons more accurately compared to the baselines and adapts its decision boundary more precisely.

Train Domains: 0-8 | Test Domain 9

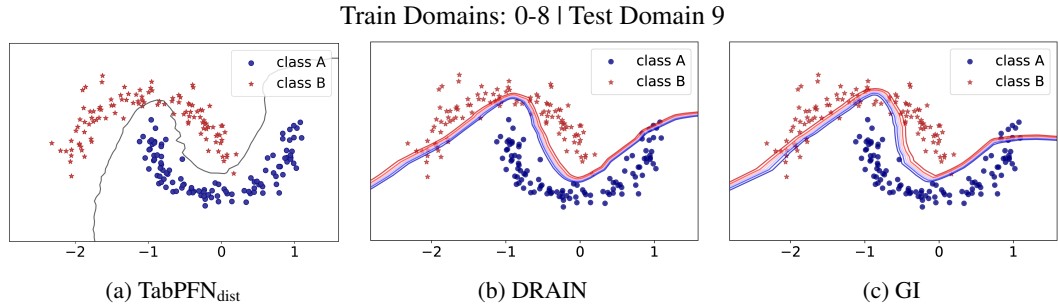

(a) TabPFN$_{\text{dist}}$        (b) DRAIN        (c) GI

Figure 9: Comparison of our method against DRAIN [12] and GI [13] on the Rotated Two Moons dataset. The models were trained on domains $\mathcal{C} = \{0, 1, \ldots, 8\}$ and tested on domain 9. While the authors of DRAIN present different, unknown levels of the decision boundary, we present the decision boundary with 50% probability. The plots for DRAIN and GI were taken from Bai et al. [12].

Table 4: Comparison of Drift-Resilient TabPFN against DRAIN and GI on the Rotated Two Moons dataset. The metric reported is the mean out-of-distribution (OOD) accuracy along with the standard deviation. Results for DRAIN and GI are taken from Bai et al. [12].

| Model | OOD Acc. ↑ |
|---|---|
| **TabPFN_dist** | **0.98** .002 |
| **DRAIN** | 0.968 .012 |
| **GI** | 0.965 .014 |

### A.4.4 Quantitative Analysis

### A.4.4.1 Combined Results Across All Datasets

Table 5: Comparison of Drift-Resilient TabPFN with various baselines and settings across the combined **real-world** and **synthetic** datasets. Metrics include accuracy, F1, ROC, and ECE for both in-distribution (ID) and out-of-distribution (OOD) data, averaged over three initializations and reported with 95% confidence intervals. The best mean of each metric is marked in bold. Metric arrows indicate optimization direction.

| Model | Variant | Acc. ↑ | | F1 ↑ | | ROC ↑ | | ECE ↓ | |
|---|---|---|---|---|---|---|---|---|---|
| | | OOD | ID | OOD | ID | OOD | ID | OOD | ID |
| **TabPFN_dist** | all dom. w. ind. | **0.744** .018 | 0.879 .012 | **0.689** .028 | 0.837 .022 | **0.832** .018 | 0.932 .002 | **0.091** .006 | 0.074 .014 |
| **TabPFN_base** | all dom. w. ind. | 0.688 .01 | **0.885** .01 | 0.62 .012 | **0.847** .017 | 0.786 .007 | **0.935** .01 | 0.119 .006 | **0.067** .005 |
| | all dom. wo. ind. | 0.645 .011 | 0.852 .016 | 0.579 .014 | 0.801 .02 | 0.736 .001 | 0.914 .007 | 0.202 .011 | 0.076 .007 |
| | last dom. wo. ind. | 0.67 .005 | 0.867 .004 | 0.609 .004 | 0.823 .011 | 0.76 .003 | 0.915 .019 | 0.181 .003 | 0.128 .007 |
| **CatBoost** | all dom. w. ind. | 0.677 .006 | 0.874 .007 | 0.62 .005 | 0.836 .01 | 0.766 .003 | 0.919 .011 | 0.222 .007 | 0.084 .009 |
| | all dom. wo. ind. | 0.632 .003 | 0.836 .013 | 0.568 .005 | 0.781 .009 | 0.714 .012 | 0.894 .014 | 0.24 .02 | 0.097 .015 |
| | last dom. wo. ind. | 0.657 .002 | 0.852 .014 | 0.599 .004 | 0.811 .024 | 0.722 .005 | 0.907 .01 | 0.256 .006 | 0.133 .012 |
| **XGBoost** | all dom. w. ind. | 0.664 .005 | 0.859 .004 | 0.61 .013 | 0.828 .003 | 0.754 .006 | 0.9 .019 | 0.194 .018 | 0.111 .02 |
| | all dom. wo. ind. | 0.633 .035 | 0.831 .024 | 0.568 .033 | 0.778 .031 | 0.718 .033 | 0.881 .028 | 0.194 .054 | 0.12 .042 |
| | last dom. wo. ind. | 0.664 .01 | 0.824 .023 | 0.599 .016 | 0.758 .054 | 0.733 .009 | 0.887 .016 | 0.199 .024 | 0.167 .011 |
| **LightGBM** | all dom. w. ind. | 0.65 .009 | 0.842 .024 | 0.594 .008 | 0.8 .024 | 0.738 .008 | 0.908 .008 | 0.198 .009 | 0.093 .016 |
| | all dom. wo. ind. | 0.625 .02 | 0.832 .019 | 0.561 .018 | 0.773 .018 | 0.706 .009 | 0.888 .01 | 0.199 .009 | 0.097 .005 |
| | last dom. wo. ind. | 0.629 .02 | 0.797 .009 | 0.542 .031 | 0.72 .028 | 0.686 .006 | 0.852 .018 | 0.224 .011 | 0.149 .016 |
| **Wild-Time ERM** | all dom. w. ind. | 0.627 .036 | 0.821 .032 | 0.525 .052 | 0.771 .044 | 0.688 .028 | 0.877 .025 | 0.263 .054 | 0.108 .017 |
| | all dom. wo. ind. | 0.582 .028 | 0.803 .01 | 0.519 .028 | 0.746 .018 | 0.673 .02 | 0.862 .008 | 0.255 .019 | 0.104 .015 |
| | last dom. wo. ind. | 0.587 .026 | 0.784 .03 | 0.528 .034 | 0.74 .011 | 0.666 .018 | 0.843 .02 | 0.323 .038 | 0.19 .025 |
| **Wild-Time SWA** | all dom. w. ind. | 0.627 .047 | 0.824 .003 | 0.534 .069 | 0.777 .014 | 0.685 .036 | 0.873 .015 | 0.274 .053 | 0.11 .007 |
| | all dom. wo. ind. | 0.588 .025 | 0.802 .007 | 0.53 .029 | 0.748 .008 | 0.681 .03 | 0.863 .008 | 0.262 .036 | 0.109 .01 |

**A.4.4.2 Critical Difference Diagrams**

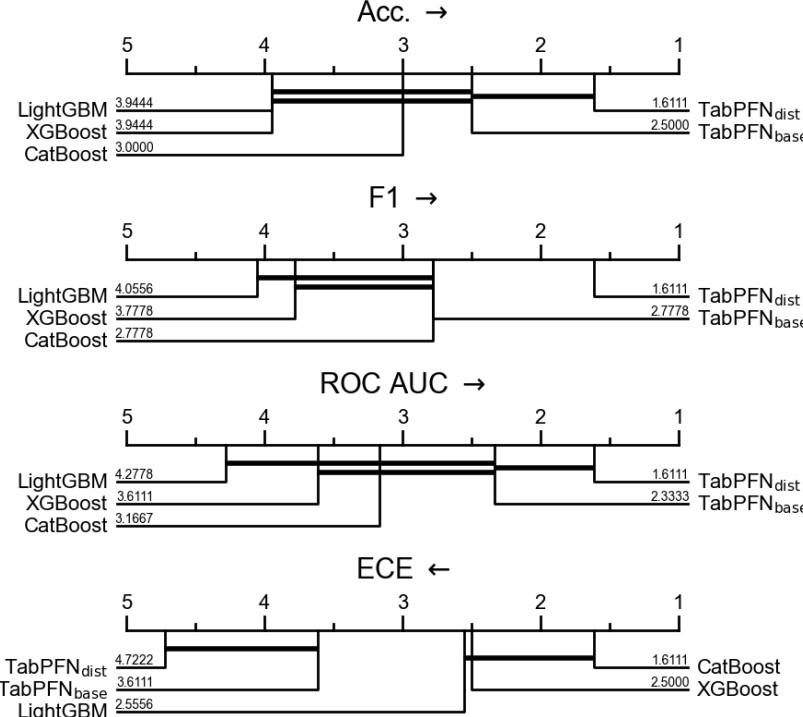

Figure 10: This figure presents critical difference diagrams of our evaluated metrics on OOD data, analyzed using the Wilcoxon-Holm method [54–57] across the best performing OOD models evaluated in this work. The diagrams indicate significant differences of Drift-Resilient TabPFN against the tree-based methods. For the F1 metric our method shows significant differences against all top performing baselines. Arrows indicate optimization direction.

**A.4.4.3  Comparison Against Wild-Time Methods**   This section offers a quantitative evaluation of our model against methods found in the Wild-Time benchmark [11]. We focus exclusively on methods applicable to tabular data, omitting SimCLR and SwAV which are specifically designed for image datasets. Detailed explanations of these methods are available in Section A.6.

As a base model, we used a Multilayer Perceptron (MLP) optimized through Hyperparameter Optimization (HPO).

The findings of this quantitative comparison are presented in Table 6. Notably, none of the evaluated Wild-Time methods demonstrated performance equal with our approach or the other baseline methods on OOD data. This discrepancy is likely due to the small number of instances in the datasets used in our evaluations, which affects the generalization of these deep learning techniques. Among the Wild-Time methods, SWA emerged as the most effective in handling OOD data.

Table 6: Comparison of Drift-Resilient TabPFN with the applicable baselines of the Wild-Time benchmark [11] across the combined **real-world** and **synthetic** datasets. Metrics include accuracy, F1, ROC, and ECE for both in-distribution (ID) and out-of-distribution (OOD) data, averaged over three initializations and reported with 95% confidence intervals. The best mean of each metric is marked in bold. Metric arrows indicate optimization direction.

| Model | Variant | Acc. ↑ | | F1 ↑ | | ROC ↑ | | ECE ↓ | |
|---|---|---|---|---|---|---|---|---|---|
| | | OOD | ID | OOD | ID | OOD | ID | OOD | ID |
| **TabPFN**$_{dist}$ | all dom. w. ind. | **0.744** $_{.018}$ | **0.879** $_{.012}$ | **0.689** $_{.028}$ | **0.837** $_{.022}$ | **0.832** $_{.018}$ | **0.932** $_{.002}$ | **0.091** $_{.006}$ | **0.074** $_{.014}$ |
| **ERM** | all dom. w. ind. | 0.627 $_{.036}$ | 0.821 $_{.032}$ | 0.525 $_{.052}$ | 0.771 $_{.044}$ | 0.688 $_{.028}$ | 0.877 $_{.025}$ | 0.263 $_{.054}$ | 0.108 $_{.017}$ |
| | all dom. wo. ind. | 0.582 $_{.028}$ | 0.803 $_{.01}$ | 0.519 $_{.028}$ | 0.746 $_{.018}$ | 0.673 $_{.02}$ | 0.862 $_{.008}$ | 0.255 $_{.019}$ | 0.104 $_{.015}$ |
| | last dom. wo. ind. | 0.587 $_{.026}$ | 0.784 $_{.03}$ | 0.528 $_{.034}$ | 0.74 $_{.011}$ | 0.666 $_{.018}$ | 0.843 $_{.02}$ | 0.323 $_{.038}$ | 0.19 $_{.025}$ |
| **FT** | all dom. w. ind. | 0.51 $_{.031}$ | 0.645 $_{.014}$ | 0.422 $_{.046}$ | 0.559 $_{.033}$ | 0.594 $_{.027}$ | 0.693 $_{.039}$ | 0.357 $_{.013}$ | 0.256 $_{.007}$ |
| | all dom. wo. ind. | 0.544 $_{.038}$ | 0.677 $_{.034}$ | 0.481 $_{.046}$ | 0.604 $_{.03}$ | 0.65 $_{.025}$ | 0.756 $_{.031}$ | 0.33 $_{.03}$ | 0.231 $_{.061}$ |
| **EWC** | all dom. w. ind. | 0.498 $_{.026}$ | 0.621 $_{.034}$ | 0.405 $_{.038}$ | 0.522 $_{.042}$ | 0.569 $_{.027}$ | 0.668 $_{.006}$ | 0.342 $_{.006}$ | 0.239 $_{.034}$ |
| | all dom. wo. ind. | 0.518 $_{.023}$ | 0.686 $_{.045}$ | 0.456 $_{.032}$ | 0.617 $_{.051}$ | 0.603 $_{.012}$ | 0.74 $_{.026}$ | 0.309 $_{.048}$ | 0.195 $_{.005}$ |
| **SI** | all dom. w. ind. | 0.478 $_{.057}$ | 0.63 $_{.006}$ | 0.383 $_{.089}$ | 0.526 $_{.015}$ | 0.566 $_{.061}$ | 0.673 $_{.023}$ | 0.37 $_{.046}$ | 0.238 $_{.012}$ |
| | all dom. wo. ind. | 0.495 $_{.019}$ | 0.674 $_{.035}$ | 0.425 $_{.036}$ | 0.588 $_{.035}$ | 0.597 $_{.022}$ | 0.744 $_{.03}$ | 0.346 $_{.002}$ | 0.214 $_{.049}$ |
| **A-GEM** | all dom. w. ind. | 0.513 $_{.031}$ | 0.677 $_{.038}$ | 0.405 $_{.04}$ | 0.576 $_{.045}$ | 0.594 $_{.061}$ | 0.741 $_{.038}$ | 0.367 $_{.024}$ | 0.223 $_{.021}$ |
| | all dom. wo. ind. | 0.486 $_{.035}$ | 0.684 $_{.023}$ | 0.403 $_{.061}$ | 0.583 $_{.05}$ | 0.58 $_{.035}$ | 0.747 $_{.019}$ | 0.364 $_{.073}$ | 0.221 $_{.054}$ |
| **CORAL-T** | all dom. w. ind. | 0.579 $_{.051}$ | 0.777 $_{.02}$ | 0.481 $_{.085}$ | 0.714 $_{.041}$ | 0.637 $_{.036}$ | 0.837 $_{.016}$ | 0.252 $_{.058}$ | 0.143 $_{.025}$ |
| | all dom. wo. ind. | 0.569 $_{.032}$ | 0.771 $_{.035}$ | 0.495 $_{.029}$ | 0.702 $_{.039}$ | 0.643 $_{.016}$ | 0.837 $_{.021}$ | 0.232 $_{.027}$ | 0.15 $_{.003}$ |
| **GroupDRO-T** | all dom. w. ind. | 0.58 $_{.025}$ | 0.779 $_{.014}$ | 0.503 $_{.038}$ | 0.723 $_{.009}$ | 0.642 $_{.036}$ | 0.84 $_{.007}$ | 0.285 $_{.019}$ | 0.125 $_{.01}$ |
| | all dom. wo. ind. | 0.576 $_{.025}$ | 0.775 $_{.023}$ | 0.514 $_{.023}$ | 0.725 $_{.036}$ | 0.658 $_{.008}$ | 0.843 $_{.006}$ | 0.264 $_{.076}$ | 0.135 $_{.04}$ |
| **IRM-T** | all dom. w. ind. | 0.577 $_{.021}$ | 0.746 $_{.014}$ | 0.49 $_{.051}$ | 0.689 $_{.013}$ | 0.633 $_{.021}$ | 0.819 $_{.016}$ | 0.259 $_{.03}$ | 0.12 $_{.013}$ |
| | all dom. wo. ind. | 0.559 $_{.018}$ | 0.742 $_{.023}$ | 0.498 $_{.031}$ | 0.678 $_{.047}$ | 0.644 $_{.005}$ | 0.822 $_{.013}$ | 0.252 $_{.031}$ | 0.134 $_{.011}$ |
| **Mixup** | all dom. w. ind. | 0.617 $_{.028}$ | 0.8 $_{.023}$ | 0.521 $_{.016}$ | 0.702 $_{.008}$ | 0.681 $_{.029}$ | 0.859 $_{.021}$ | 0.239 $_{.014}$ | 0.14 $_{.007}$ |
| | all dom. wo. ind. | 0.574 $_{.034}$ | 0.782 $_{.027}$ | 0.492 $_{.025}$ | 0.688 $_{.013}$ | 0.68 $_{.025}$ | 0.85 $_{.022}$ | 0.212 $_{.021}$ | 0.142 $_{.017}$ |
| **LISA** | all dom. w. ind. | 0.621 $_{.041}$ | 0.822 $_{.033}$ | 0.517 $_{.079}$ | 0.76 $_{.063}$ | 0.679 $_{.005}$ | 0.881 $_{.012}$ | 0.272 $_{.046}$ | 0.116 $_{.02}$ |
| | all dom. wo. ind. | 0.583 $_{.013}$ | 0.804 $_{.015}$ | 0.512 $_{.009}$ | 0.739 $_{.024}$ | 0.672 $_{.025}$ | 0.859 $_{.024}$ | 0.258 $_{.025}$ | 0.108 $_{.023}$ |
| **SWA** | all dom. w. ind. | 0.627 $_{.047}$ | 0.824 $_{.003}$ | 0.534 $_{.069}$ | 0.777 $_{.014}$ | 0.685 $_{.036}$ | 0.873 $_{.015}$ | 0.274 $_{.053}$ | 0.11 $_{.007}$ |
| | all dom. wo. ind. | 0.588 $_{.025}$ | 0.802 $_{.007}$ | 0.53 $_{.029}$ | 0.748 $_{.008}$ | 0.681 $_{.03}$ | 0.863 $_{.008}$ | 0.262 $_{.036}$ | 0.109 $_{.01}$ |

### A.4.5 Investigating Performance Saturation of Main Baseline Methods

In this subsection, we assess the computational budget allocated for HPO on our main baseline methods. By comparing model performance at 1200 seconds versus 3600 seconds per method and dataset split, we show that the increased budget for HPO does not yield significant gains overall, indicating sufficient saturation within the initial budget.

Table 7: Comparison of GBDT baselines across combined **real-world** and **synthetic** datasets for two different hyperparameter optimization (HPO) time budgets: 1200s and 3600s. Metrics include accuracy, F1, ROC, and ECE for both in-distribution (ID) and out-of-distribution (OOD) data, averaged over three initializations and reported with 95% confidence intervals. Metric arrows indicate optimization direction.

| Model | Variant | Acc. ↑ | | F1 ↑ | | ROC ↑ | | ECE ↓ | |
|---|---|---|---|---|---|---|---|---|---|
| | | OOD | ID | OOD | ID | OOD | ID | OOD | ID |
| $\text{CatBoost}_{1h}$ | all dom. w. ind. | $0.678_{.007}$ | $0.872_{.006}$ | $0.621_{.01}$ | $0.833_{.004}$ | $0.764_{.007}$ | $0.914_{.011}$ | $0.205_{.017}$ | $0.084_{.025}$ |
| | all dom. wo. ind. | $0.632_{.005}$ | $0.838_{.008}$ | $0.568_{.007}$ | $0.782_{.01}$ | $0.713_{.004}$ | $0.893_{.026}$ | $0.226_{.002}$ | $0.099_{.018}$ |
| | last dom. wo. ind. | $0.656_{.007}$ | $0.85_{.024}$ | $0.598_{.01}$ | $0.813_{.032}$ | $0.721_{.005}$ | $0.903_{.018}$ | $0.264_{.019}$ | $0.136_{.028}$ |
| $\text{CatBoost}_{20min}$ | all dom. w. ind. | $0.677_{.006}$ | $0.874_{.007}$ | $0.62_{.005}$ | $0.836_{.01}$ | $0.766_{.003}$ | $0.919_{.011}$ | $0.222_{.007}$ | $0.084_{.009}$ |
| | all dom. wo. ind. | $0.632_{.003}$ | $0.836_{.013}$ | $0.568_{.005}$ | $0.781_{.009}$ | $0.714_{.012}$ | $0.894_{.014}$ | $0.24_{.02}$ | $0.097_{.015}$ |
| | last dom. wo. ind. | $0.657_{.002}$ | $0.852_{.014}$ | $0.599_{.004}$ | $0.811_{.024}$ | $0.722_{.005}$ | $0.907_{.01}$ | $0.256_{.006}$ | $0.133_{.012}$ |
| $\text{XGBoost}_{1h}$ | all dom. w. ind. | $0.662_{.013}$ | $0.864_{.013}$ | $0.61_{.01}$ | $0.834_{.017}$ | $0.754_{.009}$ | $0.911_{.012}$ | $0.187_{.017}$ | $0.101_{.017}$ |
| | all dom. wo. ind. | $0.639_{.012}$ | $0.833_{.002}$ | $0.578_{.01}$ | $0.781_{.007}$ | $0.724_{.006}$ | $0.886_{.014}$ | $0.181_{.019}$ | $0.124_{.022}$ |
| | last dom. wo. ind. | $0.664_{.011}$ | $0.833_{.021}$ | $0.598_{.022}$ | $0.765_{.04}$ | $0.736_{.012}$ | $0.895_{.014}$ | $0.195_{.016}$ | $0.164_{.005}$ |
| $\text{XGBoost}_{20min}$ | all dom. w. ind. | $0.664_{.005}$ | $0.859_{.004}$ | $0.61_{.013}$ | $0.828_{.003}$ | $0.754_{.006}$ | $0.9_{.019}$ | $0.194_{.018}$ | $0.111_{.02}$ |
| | all dom. wo. ind. | $0.633_{.035}$ | $0.831_{.024}$ | $0.568_{.033}$ | $0.778_{.031}$ | $0.718_{.033}$ | $0.881_{.028}$ | $0.194_{.054}$ | $0.12_{.042}$ |
| | last dom. wo. ind. | $0.664_{.01}$ | $0.824_{.023}$ | $0.599_{.016}$ | $0.758_{.054}$ | $0.733_{.009}$ | $0.887_{.016}$ | $0.199_{.024}$ | $0.167_{.011}$ |
| $\text{LightGBM}_{1h}$ | all dom. w. ind. | $0.647_{.012}$ | $0.846_{.005}$ | $0.593_{.013}$ | $0.805_{.003}$ | $0.736_{.006}$ | $0.909_{.005}$ | $0.211_{.004}$ | $0.096_{.007}$ |
| | all dom. wo. ind. | $0.618_{.005}$ | $0.822_{.003}$ | $0.555_{.007}$ | $0.767_{.006}$ | $0.708_{.007}$ | $0.888_{.018}$ | $0.2_{.014}$ | $0.102_{.008}$ |
| | last dom. wo. ind. | $0.624_{.015}$ | $0.795_{.021}$ | $0.546_{.029}$ | $0.724_{.032}$ | $0.687_{.005}$ | $0.845_{.01}$ | $0.231_{.014}$ | $0.148_{.017}$ |
| $\text{LightGBM}_{20min}$ | all dom. w. ind. | $0.65_{.009}$ | $0.842_{.024}$ | $0.594_{.008}$ | $0.8_{.024}$ | $0.738_{.008}$ | $0.908_{.008}$ | $0.198_{.009}$ | $0.093_{.016}$ |
| | all dom. wo. ind. | $0.625_{.02}$ | $0.832_{.019}$ | $0.561_{.018}$ | $0.773_{.018}$ | $0.706_{.009}$ | $0.888_{.01}$ | $0.199_{.009}$ | $0.097_{.005}$ |
| | last dom. wo. ind. | $0.629_{.02}$ | $0.797_{.009}$ | $0.542_{.031}$ | $0.72_{.028}$ | $0.686_{.006}$ | $0.852_{.018}$ | $0.224_{.011}$ | $0.149_{.016}$ |

### A.4.6 Analyzing the Relationship Between Out-of-Distribution Performance and Difficulty Across Datasets

In this supplementary evaluation, we investigate the correlation between the inherent difficulty of a dataset concerning distribution shifts and the performance of our approach in comparison to the baselines.

**Defining Difficulty Metrics.** Our first step involves assigning a difficulty score to individual dataset splits, using a basic XGBoost model that doesn't include domain indices as a feature of the dataset. This score is determined using the formula $ID - OOD$, where both ID and OOD performances are measured using ROC AUC. This difficulty score quantifies the decline in model performance when transitioning from ID to OOD data. After calculating the difficulty scores for each dataset split, we proceed to train each of the evaluated methods across all domains, including the domain indices. We then compute the performance gap, which is also calculated as $ID - OOD$, for each method.

These metrics are then visualized by plotting the dataset's difficulty score against its corresponding performance gap for each dataset split. We also fit linear regression lines to the scatter plots representing each method. The final plot is illustrated in Figure 11.

Although the data points cannot be interpreted directly, the linear regression suggests that our method experiences the least decline in ROC AUC performance as dataset difficulty increases, affirming its robustness. The TabPFN baseline is the second-best performer, while the other models show more significant drops in performance.

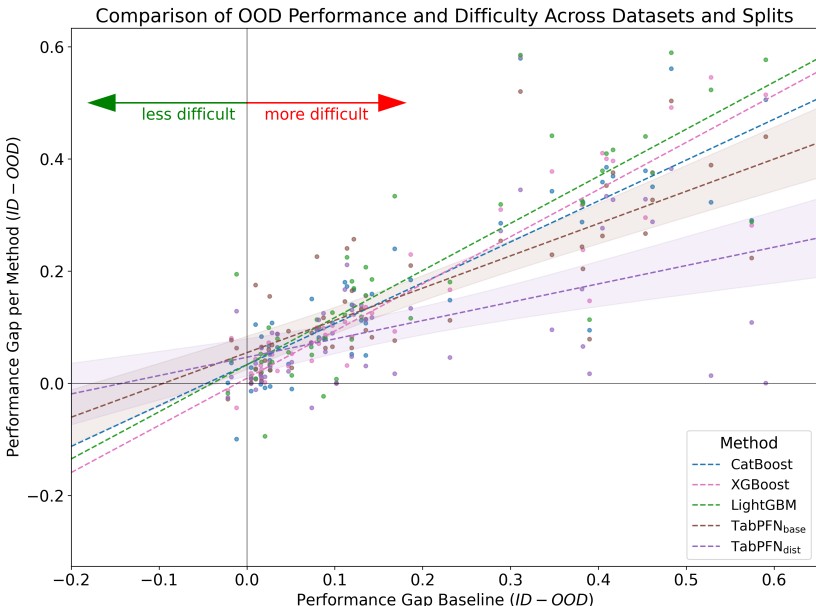

Figure 11: This figure illustrates a comparative analysis of the resilience of the listed methods to out-of-distribution (OOD) difficulty across multiple datasets and splits. The $x$-axis captures the difficulty of each dataset split, while the $y$-axis measures the performance drop of a method compared to in-distribution (ID) performance. Individual methods are represented by scatter points and their corresponding linear regression lines, with shaded regions indicating the 95% confidence intervals for TabPFN methods. Directional arrows signify increasing or decreasing dataset difficulty. Flatter regression slopes indicate models that are more resilient to increases in dataset difficulty due to distribution shifts.

### A.4.7 Number of Shift Observations Required for Effective Extrapolation

For this analysis, we evaluated the performance of both our method and TabPFN-base by fixing the prediction to the last domain of the Intersecting Blobs dataset and gradually increasing the number of domains available during training. Our results confirm observations made during method development: Our method requires significantly fewer training domains to accurately extrapolate shifts into the distant future, while the TabPFN-base only has acceptable decision boundaries when there is data whose distribution is close to the test domain. Below in Figure 12 are the results for this experiment with the Intersecting Blobs dataset discussed in Section A.4.3.1. The figure clearly shows that our model greatly improves its predictions with the first domains. When provided with the domains $\mathcal{C}^{\text{train}} = \{0, \ldots, 3\}$, the shift is largely understood, with newer domains having little impact on performance. On the other hand, TabPFN-base requires much more training domains to achieve similar performance to our method.

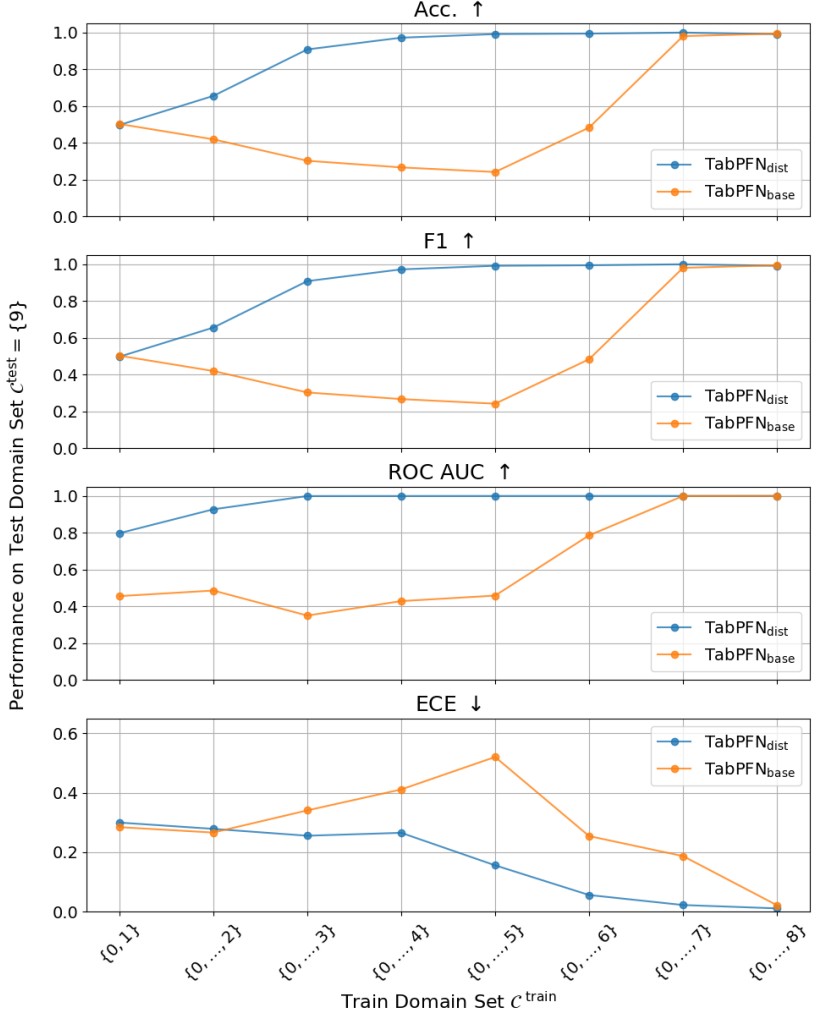

Figure 12: This figure shows the performance of Drift-Resilient TabPFN and the baseline TabPFN on the Intersecting Blobs dataset. Thereby, we always test on domain $\mathcal{C}^{\text{test}} = \{9\}$ and gradually increase on the $x$-axis the number of training domains starting with $\mathcal{C}^{\text{train}} = \{0, 1\}$ up to $\mathcal{C}^{\text{train}} = \{0, 1, \ldots, 8\}$. The results show that Drift-Resilient TabPFN achieves effective extrapolation with as few as four training domains, while TabPFN-base needs significantly more to reach similar performance.

## A.5 Supplementary Illustrations and Methodological Overviews

### A.5.1 Plots Illustrating Types of Distribution Shifts

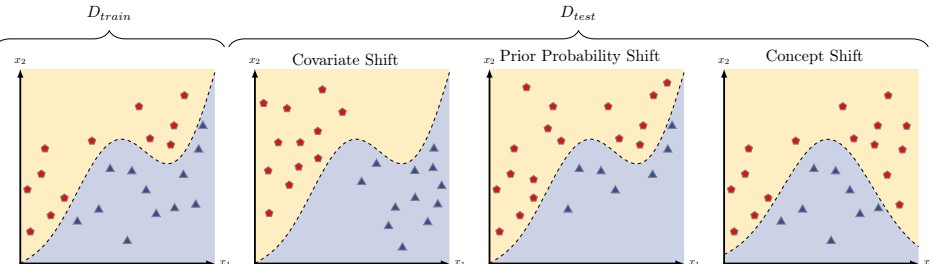

Figure 13: Illustration of initial and shifted data distributions alongside their optimal decision boundaries. The left panel depicts the initial classification dataset with two features and its true-data-optimal decision boundary. The right panel presents the dataset subjected to the three primary types of distribution shifts observed during test time.

### A.5.2 Plots Illustrating Sampled Parameters of the Adjusted Prior

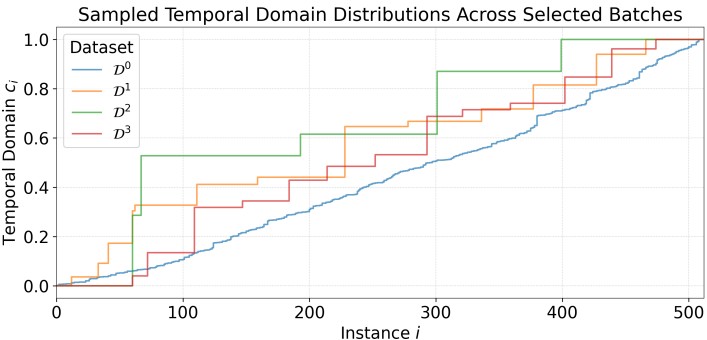

Figure 14: Share of temporal domains in exemplary datasets prior seen up to any instance $i$. The figure illustrates the range and structure of the sampled temporal domains $c_k \in \mathcal{C}$ across four representative datasets. It highlights variations in domain size and demonstrates the presence of arbitrary gaps, simulating irregularities in data sampling.

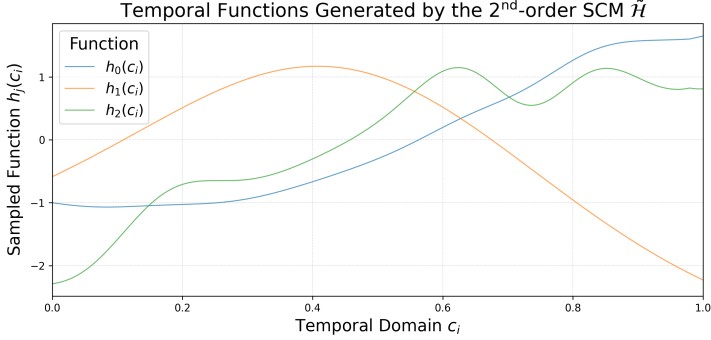

Figure 15: This figure presents three exemplary functions sampled from nodes within the network of a 2nd-order SCM $\tilde{\mathcal{H}}$. In the plot, the $x$-axis represents the input temporal domain $c_k \in \mathcal{C}$, while the $y$-axis displays the corresponding node activation.

### A.5.3  Algorithmic Overview of Our Approach

---

**Algorithm 1** This algorithm provides a high-level overview for generating a synthetic dataset in our prior. Although steps are depicted sequentially for clarity, many can be parallelized in actual implementation.

---

1: **procedure** SAMPLEDATASET
2:   $\mathcal{G} \leftarrow$ SAMPLESCM()           ▷ Sample data-generating SCM
3:   $\tilde{\mathcal{G}} \leftarrow \mathcal{G}$.EXPAND()          ▷ Expand to functional representation

4:   $\mathcal{H} \leftarrow$ SAMPLESCM()          ▷ Sample 2$^{\text{nd}}$-order SCM
5:   $\tilde{\mathcal{H}} \leftarrow \mathcal{H}$.EXPAND()          ▷ Expand to functional representation

6:   $\mathcal{C} \leftarrow \{c_1, c_2, \ldots, c_t\}$         ▷ Sample temporal domains
7:   $\mathcal{D} \leftarrow \emptyset$             ▷ Initialize dataset

8:   **for all** $c_k \in \mathcal{C}$ **do**
9:    $\boldsymbol{\omega}_{c_k} \leftarrow \tilde{\mathcal{H}}$.FORWARD$(c_k)$      ▷ Sample edge shifts
10:    $\tilde{\mathcal{G}}_{c_k} \leftarrow \tilde{\mathcal{G}}$.UPDATE$(\boldsymbol{\omega}_{c_k})$      ▷ Update edge weights

11:    $\mathcal{D}_{c_k} \leftarrow \emptyset$          ▷ Initialize sub-dataset
12:    **for all** $i \in \{1, ..., n_{c_k}\}$ **do**     ▷ Sample sub-dataset
13:     $(\boldsymbol{x}_i, y_i, c_k) \leftarrow \tilde{\mathcal{G}}_{c_k}$.FORWARD$(\epsilon_i)$
14:     $\mathcal{D}_{c_k} \leftarrow \mathcal{D}_{c_k} \cup \{(\boldsymbol{x}_i, y_i, c_k)\}$
15:    **end for**

16:    $\mathcal{D} \leftarrow \mathcal{D} \cup \mathcal{D}_{c_k}$         ▷ Extend dataset
17:   **end for**

18:   **return** $\mathcal{D}$           ▷ Return dataset
19: **end procedure**

---

### A.5.4  Illustration of Adopted Evaluation Strategy

In Figure 16, we provide an illustration of the Eval-Fix evaluation strategy, originally proposed by Yao et al. [11] in the context of the Wild-Time benchmark. It should be noted that the formalism has been adapted to align with the notation used in this paper.

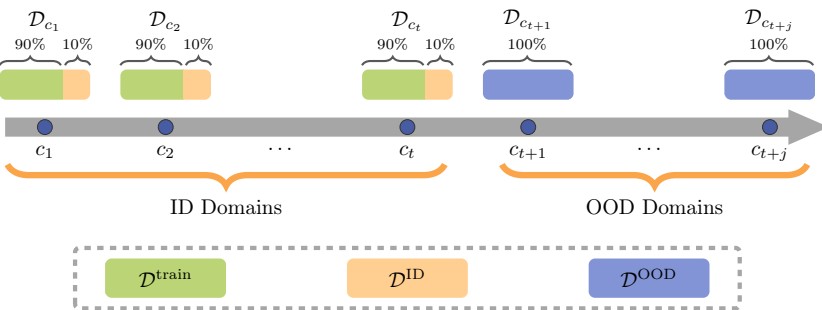

Figure 16: Adapted from the Wild-Time benchmark [11], this illustration portrays the Eval-Fix evaluation strategy employed in our study. The domain boundary is indicated by $c_t$, beyond which datasets are considered part of the out-of-distribution (OOD) test set $\mathcal{D}^{\text{OOD}}$. To evaluate in-distribution (ID) performance, we subsample 10% of the samples from each dataset prior to this boundary, forming the datasets $\mathcal{D}^{\text{train}}$ and $\mathcal{D}^{\text{ID}}$.

### A.6 Detailed Overview of Wild-Time Methods

#### A.6.1 Classical Supervised Learning

**Empirical Risk Minimization (ERM).** ERM - a fundamental approach in supervised learning - focuses on minimizing the average loss over the training dataset. In Wild-Time ERM is defined as typical supervised learning without making use of any temporal information.

#### A.6.2 Continual Learning

**Fine-Tuning (FT).** FT extends the ERM approach by training on the data of each successive temporal domain separately, allowing the model to adapt to new distributions but risking catastrophic forgetting of past tasks.

**Elastic Weight Consolidation (EWC).** EWC [58] counters catastrophic forgetting by adding a regularization term that constrains the changes to important model parameters, thus preserving knowledge from previous tasks.

**Synaptic Intelligence (SI).** SI [59] captures a synaptic strength metric over time for each model parameter. This metric is used as a regularizer to limit changes to important parameters during learning.

**Averaged Gradient Episodic Memory (A-GEM).** A-GEM [60] maintains a small episodic memory and computes gradients not just for the current task but also the average of the gradients over several past tasks stored in the episodic memory.

#### A.6.3 Temporally Invariant Learning

**Deep Correlation Alignment (Deep CORAL).** Initially developed for domain adaptation, Deep CORAL [35] aims to align the second-order statistics of features between the source and target domains to minimize distribution shift. In the Wild-Time benchmark, this original purpose is modified to align features across different temporal domains within the training set, thus converting it into a DG method. For handling temporal shifts, it is extended into CORAL-T, which employs sliding windows to segment the data stream into temporal substreams, treating each as a separate domain for alignment. [11]

**Group Distributionally Robust Optimization (GroupDRO).** Originally designed at optimizing on the worst-performing group within the training data, GroupDRO [32] aims to learn a model that is robust across varying group distributions. In the Wild-Time benchmark, this method is adapted to the temporal context as GroupDRO-T. It utilizes sliding window-based segmentation to create temporal substreams, treating each as a separate group for distributionally robust optimization. [11]

**Invariant Risk Minimization (IRM).** IRM [31] aims to identify a data representation that is consistently predictive across different domains. In the context of Wild-Time, the method is adapted to temporal shifts and named IRM-T. It employs sliding window-based segmentation to create temporal substreams, which are then treated as individual domains for invariant risk minimization. [11]

**Mixup.** Mixup [33] is an interpolation-based data augmentation technique that creates new training examples by blending the features and labels of existing samples. This technique aims to enhance the model's ability to generalize across domains by diversifying the training data. It replaces the original training samples with these newly generated interpolations for more robust training.

**Learning with Selective Augmentation (LISA).** LISA [34], motivated by Mixup, employs selective interpolation to neutralize domain-specific information in the training data. It comes in two variants: intra-label LISA, which interpolates examples from different domains but having the same label, and intra-domain LISA, which interpolates examples within the same domain but having different labels. In Wild-Time, only intra-label LISA is used [11].

#### A.6.4 Self-Supervised Learning

**Simple Framework for Contrastive Learning of Visual Representations (SimCLR).** SimCLR [61] employs contrastive learning to maximize the agreement between different augmentations of the same image, thereby enhancing the quality of learned visual representations. The approach benefits from learnable nonlinear transformations and optimized contrastive loss parameters.

**Swapping Assignments between multiple Views of the same image (SwAV).** SwAV [62] employs a clustering approach within the contrastive learning framework. It enforces consistency between cluster assignments across different augmentations of the same image. This obviates the need for pairwise feature comparisons, offering computational efficiency.

### A.6.5 Bayesian Learning

**Stochastic Weight Averaging (SWA).** SWA [46] averages multiple parameter values along the stochastic gradient descent (SGD) trajectory to improve in-distribution generalization. It operates with minimal computational overhead and aims to approximate the posterior distribution over model parameters, reflected in the flatness of the learned optima. [11]

## A.7 Datasets

### A.7.1 Validation Datasets

#### A.7.1.1 Synthetic Datasets

**Dataset 1 (Shifting Sin Classification).** The Shifting Sin Classification dataset is a synthetic, binary classification dataset of 1,500 instances evenly distributed across 10 domains. Each domain is differentiated by a unique shift in the offset of the sinusoid wave function, creating distinct decision boundaries for classification. The dataset contains two features corresponding to the $x$ and $y$ coordinates of each instance. Instances are labeled as 1 if they lie above the sine curve and 0 otherwise in their respective domains.

**Dataset 2 (Rotated Five Blobs).** The Rotated Five Blobs dataset is a synthetically generated dataset consisting of five blobs rotated sequentially counterclockwise $-20°$ around a central point in each domain. It comprises two numerical features representing the $x$ and $y$ coordinates of each data point. Each blob consists of 40 samples, resulting in 200 samples per domain, for a total of 2,000 samples across the 10 domains represented.

**Dataset 3 (Moving Square).** The Moving Square is a synthetically generated dataset designed for a multi-class classification task. It encompasses two features and is divided into six domains, each containing 200 instances, thereby leading to a total of 1,200 samples. In the construction of this dataset, each of the four clusters—representing distinct classes—is initially located on one corner of a square. As we transition through the six domains, each cluster progressively moves along the edge of the square to the next corner.

**Dataset 4 (Moving Diagonal Line).** The Moving Diagonal Line dataset is a synthetic dataset, generated using the sklearn blobs function. It comprises 1,200 instances, divided across 6 domains, with each domain holding 200 instances. There are two features, corresponding to the $x$ and $y$ coordinate of each instance. In this dataset, there are two clusters, each representing a class, following a diagonal line that moves with each domain. Thereby, both clusters move in opposite directions along parallel diagonal next to each other. Each domain in this context represents different stages of the diagonal movement.

#### A.7.1.2 Real World Datasets

**Dataset 5 (Indian Liver Patient Dataset).** The Indian Liver Patient Dataset (ILPD), referenced from Ramana and Venkateswarlu [63], is tailored for the binary classification task of identifying liver disease. It contains 583 records featuring 10 attributes, including age, gender, and diverse biochemical measurements. Originating from Andhra Pradesh, India, it comprises 416 liver patient records and 167 non-liver patient records. In our settings, every 5-year age interval is considered as an individual domain. The dataset, sourced from the UCI Machine Learning Repository, is geared towards supporting the diagnosis of liver disease.

**Dataset 6 (Istanbul Stock Exchange Returns).** The Istanbul Stock Exchange Returns dataset sourced from the UCI Machine Learning Repository provided by Akbilgic [64] includes 536 instances of returns from the Istanbul Stock Exchange and seven international indices from June 2009 to February 2011. The eight attributes represent various market return indices. The dataset is thereby used to predict the changes in the Istanbul stock exchange given all the other indices. The target was thereby discretized into 9 categories. The data was processed by dropping the USD column of the ISE and converting dates into a monthly domain feature, introducing a time-based distribution shift.

**Dataset 7 (Diabetes 130-US Hospitals).** The Diabetes 130-US Hospitals dataset provided by Strack et al. [65] encapsulates a decade (1999-2008) of diabetes care across 130 US hospitals, detailing 50 features related to patient demographics and hospitalization details. Criteria for inclusion are inpatient and diabetic encounters, with stays ranging from 1 to 14 days, where both lab tests and medications were administered. Features include patient identifiers, race, gender, age, admission type, duration of stay, attending physician's specialty, lab test counts, HbA1c results, diagnoses, medication details, and counts of healthcare visits prior to admission. The target of the prediction is to determine whether and in what time frame a patient will be readmitted. It is categorized into <30 days, >30 days, or No for no readmission. The original dataset, with 101,766 instances and 50 features, has been subsampled to 964 instances.

**Dataset 8 (Airlines Delay).** The Airlines Delay dataset, sourced from OpenML and provided by Bifet and Ikonomovska [66], contains 539,383 instances, each with 7 features. The task is to predict flight delays based on the scheduled departure information. Features include Airline, Flight, AirportFrom, AirportTo, DayOfWeek, and Time of departure. Notably, departure time, discretized to an interval of full hours, will be our distribution shift domain. The dataset was subsampled, thereby we sampled at most 60 samples per discrete time step. This resulted in 1,380 instances.

**Dataset 9 (Pima Indians Diabetes).** The Pima Indians Diabetes dataset from the National Institute of Diabetes and Digestive and Kidney Diseases, referenced by Smith et al. [67], contains 768 instances and 8 medical diagnostic features. These data represent female Pima Indian patients aged 21 or older. The task involves binary classification for predicting diabetes onset. We categorize each successive 2-year age interval as a separate domain, highlighting shifts in the dataset across age groups.

**Dataset 10 (Diabetes Prediction through Questionaire).** This dataset, collected from Sylhet Diabetes Hospital in Bangladesh and provided by Islam et al. [68], aims to predict early-stage diabetes. It comprises 520 instances and 16 features, representing symptoms and demographic information of patients. The task is binary classification, predicting whether a patient has diabetes or not. Age groups of every successive 5-year interval are considered as different domains, providing 14 age-based domains.

**Dataset 11 (Room Occupancy Detection).** The dataset is sourced from the UCI Machine Learning Repository and provided by Candanedo [69]. The preprocessed dataset, reduced to 1800 instances from the original 20,560, is used for binary classification of room occupancy based on Temperature, Humidity, Light, and CO2 levels. After removing 'date' and 'Id' features, 'day' and 'hour' were added. Thereby the 'day' was used as the temporal domain.

**Dataset 12 (Sao Paulo Urban Traffic Behavior).** The Sao Paulo Urban Traffic Behavior dataset, sourced from the UCI Machine Learning Repository provided by Ferreira et al. [70], captures records of urban traffic behavior in Sao Paulo, Brazil, from December 14 to 18, 2009, and tries to predict the slowness in traffic. The dataset contains 135 instances each with 18 attributes.

Attributes are various traffic indicators such as Hour, Immobilized bus, Broken Truck, Vehicle excess, Accident victim, and more. We have discretized the target "Slowness in traffic (%)" into intervals of 7.5 percent. Each day represents a different domain, thus introducing a time-based shift in the dataset.

### A.7.2  Test Datasets

#### A.7.2.1  Synthetic Datasets

**Dataset 13 (Rotated Two Moons).** The Rotated Two Moons dataset as stated by Nasery et al. [13] and Bai et al. [12], is a derivative of the 2-entangled moons dataset and includes 220 instances in each of the 10 domains. Each domain is differentiated by counter-clockwise rotations of $18°$, resulting in a rotation of $18 \cdot i°$ in domain $i$. Also, the distribution of a subset of the instances varies across domains.

**Dataset 14 (Intersecting Blobs).** The Intersecting Blobs Dataset is a synthetically created set tailored for complex, binary classification tasks. The dataset contains two features per sample and 120 samples per domain in a total of 14 domains. Each domain comprises three classes, each represented by 40 samples appearing as blobs. These blobs move and vary, getting quite close to one another, almost intersecting. This dynamic creates sudden shifts in decision boundaries, increasing the difficulty and complexity of the classification tasks. The continuous shifts in blobs' positions across domains are implemented by adjusting their centers and standard deviations.

**Dataset 15 (Binary Label Shift).** The Binary Label Shift Dataset is a synthetic dataset tailored for binary classification in an environment with prior probability shifts. The dataset consists of

10 unique domains, with each containing 200 samples and each sample consisting of two features. The key feature of this dataset is the systematic manipulation of class probabilities across domains. This is realized by starting with a high probability of 0.95 for one class in the first domain, which progressively diminishes to 0.05 in the final domain. Simultaneously, the probability for the other class increases from 0.05 to 0.95, following the opposite direction. This dynamic essentially portrays a "fade out and fade in" pattern of the classes across the domains, representing the prior probability shift.

**Dataset 16 (Rotating Hyperplane).** The Rotating Hyperplane binary classification dataset is artificially generated based on the package `scikit-multiflow` provided by Montiel et al. [71]. It consists of five features, of which three shift over time. The dataset is divided into 15 domains of 100 instances each, providing a total of 1500 samples. As the name implies, the key aspect of this dataset is about a rotating hyperplane. In other words, the decision boundary - or hyperplane - shifts as we navigate from one domain to the next, making the classification task increasingly difficult.

**Dataset 17 (RandomRBF Drift).** The RandomRBF Drift binary classification dataset is synthetically generated based on the package `scikit-multiflow` provided by Montiel et al. [71]. This dataset has been constructed by introducing drifts in the data using Radial Basis Functions. It is characterized by the motion of cluster centroids, which are responsible for creating data drift. This movement can be visualized as clusters that change their positions, altering the distribution of data over time. The dataset consists of 8 features, 15 domains with 100 samples in each domain, for a total of 1,500 samples.

**Dataset 18 (Rotating Segments).** The Rotating Segments dataset is a synthetically generated dataset tailored for a binary classification task. The dataset visualizes a circle partitioned into four segments, similar to the slices of a cake. The data points are thereby labeled alternately. As we traverse through the ten domains, these segments undergo a rotation. Each domain contains 150 samples, accumulating to 1500 samples in total.

**Dataset 19 (Sliding Circle).** The Sliding Circle dataset is a synthetically generated dataset and represents a binary classification task. It comprises two features, the dataset is partitioned into ten domains, each possessing 200 samples, summing up to a total of 2,000 samples. The unique aspect of this dataset is its visual representation: a smaller circle slides around the inner perimeter of a larger circle. Within the larger circle, points are classified based on whether they lie inside the smaller sliding circle or outside of it. As we traverse through the ten domains, the position of the smaller circle changes, causing a shift in the classification of the points.

**Dataset 20 (Shifting Two Spirals).** The Shifting Two Spirals dataset is designed for binary classification tasks. The dataset visually represents two intertwined spirals. As we move from one domain to the next, the classification boundary in the spirals evolves. Specifically, one spiral gradually transitions its labels from the center towards the outer end, while the other spiral does the exact opposite, transitioning its labels from the outer end towards the center. This dynamic showcases a fascinating interplay of domain adaptation across the ten domains. Each domain has 200 samples, with 100 samples from each spiral, summing up to a total of 2,000 samples.

### A.7.2.2 Real World Datasets

**Dataset 21 (Free Light Chain Mortality).** The free light chain dataset comprises data from a study investigating the link between serum free light chain (FLC) and mortality. It includes 1125 stratified samples per domain and target from an original pool of 7,874, featuring residents of Olmsted County, Minnesota aged 50 or more.

The task involves predicting mortality based on 9 features, which include age, sex, year of blood sample, FLC portions (kappa and lambda), the FLC group, serum creatinine, MGUS diagnosis, and days from enrollment to death or last follow-up. The feature 'chapter' was omitted because it is direct information on whether someone died or not. We treat "sample.yr", the year in which the sample was taken, as a domain, resulting in a total number of 9 domains. We suspect that both the measurements themselves and the selection of participants have changed over time. The dataset is sourced from studies by Dispenzieri et al. [72] and Kyle et al. [73].

**Dataset 22 (Electricity).** This dataset as used by Harries [74] and Gama et al. [75] contains electricity demand in the Australian New South Wales Electricity Market. Since the prices are not fixed, they fluctuate depending on supply and demand. It has 45,312 instances and 5 features. For reproducibility, the dataset includes additional features such as the New South Wales electricity price, which was

used to form the target class according to the original paper, and the Victoria electricity price, which was not used in the original paper. The dataset features the demand of electricity in to provinces as well as the transfer between those for periods of 30 minutes. The task is a binary classification, which requires predicting whether the price in the current period is higher or lower than the average of the last 24 hours. The dataset contains seasonal data due to varying demand for electricity. The effects of long-term price trends on the class label are removed by the 24-hour moving average. We consider one-week periods as a single domain. To comply with TabPFN sequence length limits, we keep only two hourly intervals for each day and subsample 15 weeks of the whole time period.

**Dataset 23 (Absenteeism at Work).** The Absenteeism at Work dataset, sourced from the UCI Machine Learning Repository provided by Martiniano and Ferreira [76], comprises 740 instances across 21 features. It captures various attributes of employees and their working conditions, such as the reason for absence, day of the week, seasons, distance to work, and more, with the target feature being the absenteeism time in hours which was discredited into 4-quantiles.

The primary shift in this dataset is supposed to be seasonal. Thereby each consecutive season is treated as a different domain. Furthermore, no significant preprocessing or subsampling was required due to the manageable size of the dataset.

**Dataset 24 (Heart Disease Dataset).** The Heart Disease dataset, sourced from the UCI Machine Learning Repository and provided by Janosi et al. [77], targets the classification task of identifying the presence (values 1,2,3,4) or absence (value 0) of heart disease in patients. We treat the task as a binary classification, predicting only whether a patient has heart disease or not. It includes 303 instances, each with 13 health-related features such as age, sex, resting blood pressure, cholesterol levels, etc. In this context, each consecutive 4-year age interval is viewed as a single domain. Rows with missing values have been omitted.

**Dataset 25 (Parking Birmingham).** The Parking Birmingham dataset, provided by Stolfi [78] via the UCI Machine Learning Repository, initially comprises the capacity and occupancy rates(target) of multiple car parks. We have processed it to only include the car park labeled 'Others-CCCPS133', thereby reducing the number of instances to 1,294 from 35,717. The original 'LastUpdated' attribute has been transformed into 'day', 'week', and 'month' features, with the 'week' serving as the temporal domain. The 'Occupancy' target, originally an absolute figure, is now presented as a percentage of the parking space utilization, discretized into 25% intervals.

**Dataset 26 (Ames Housing Prices).** The Ames Housing Dataset, curated by De Cock [79], consists of 1460 instances each with 79 features detailing various aspects of residential homes in Ames, Iowa. We use only the training portion of this dataset due to the absence of ground truth targets in the test data. We have discretized the house price, which was originally a continuous variable, into a categorical variable. This transformation was achieved by partitioning the price data into intervals. Specifically, the intervals are defined as: $[0, 125k], (125k, 300k], (300k, \infty)$. The task is to predict the price range of a home based on its features. We treat the 'YearBuilt' attribute, divided into 15-year periods, as our domain to capture changes in housing trends over time. It is to be expected that the data set shows temporal shifts, as the price distribution between older and more modern houses differs.

**Dataset 27 (Folktables US Census).** The folktables datasets, derived from the US Census Public Use Microdata Sample (PUMS) data and published by Ding et al. [42] consists of demographic and socioeconomic data between 2015 and 2021. Each year within this timeframe represents a distinct domain for a series of tasks: ACSIncome, ACSPublicCoverage, and ACSEmployment. We purposefully limited our focus to the state Maryland, to limit the shifts to the temporal domain. The size of the datasets necessitated a stratified subsample for the target per year to reach a total of approximately 1300 instances per dataset and meet the TabPFN model requirements. This subsampling ensured a representative yet computationally manageable sample.

The tasks are as follows:

- **ACSIncome:** The task predicts whether an individual's income surpasses $50,000, narrowing the ACS PUMS data to individuals over 16 who reported at least one working hour per week in the past year and a minimum income of $100. The task consists of 10 features accross the 7 domains.

- **ACSPublicCoverage:** The objective is to predict if an individual is covered by public health insurance. The dataset is filtered to include only individuals under 65 with an income below

$30,000, focusing the prediction on low-income individuals ineligible for Medicare. The task consists of 19 features across the 7 domains.

- **ACSEmployment:** The task is to predict whether an individual is employed. The dataset is filtered to include only individuals between 16 and 90 years of age. The task consists of 16 features accross the 7 domains.

**Dataset 28** (**Chess**). The Chess dataset published by Žliobaitė [80] is derived from recorded chess games, aiming to predict game outcomes (draw, lost, won) through a multi-class classification task. It consists of nine features which provide insights into the game details and player attributes, including move sequences, player side (white or black), current rating, opponent's rating, type of game, speed, and the date of the game (broken down into year, month, and day). The dataset is segmented into 27 domains, where each domain represents 20 consecutive games. This segmentation helps capture the evolution of a player's progress over time. The dataset, in its entirety, holds 533 instances. It has been constructed based on games played between 7 December 2007 and 26 March 2010.

## A.8 Hyperparameter Search Spaces

Table 8: Hyperparameter search spaces we used for our baselines.

| **XGBoost** | |
| --- | --- |
| Parameter | Values |
| learning_rate | $e^{\mathcal{U}(-7,0)}$ |
| max_depth | $\mathcal{U}\{1, 2, \ldots, 10\}$ |
| subsample | $\mathcal{U}(0.2, 1)$ |
| colsample_bytree | $\mathcal{U}(0.2, 1)$ |
| colsample_bylevel | $\mathcal{U}(0.2, 1)$ |
| min_child_weight | $e^{\mathcal{U}(-16,5)}$ |
| alpha | $e^{\mathcal{U}(-16,2)}$ |
| lambda | $e^{\mathcal{U}(-16,2)}$ |
| gamma | $e^{\mathcal{U}(-16,2)}$ |
| n_estimators | $\mathcal{U}\{100, 101, \ldots, 4000\}$ |

| **LightGBM** | |
| --- | --- |
| Parameter | Values |
| num_leaves | $\mathcal{U}\{5, 6, \ldots, 50\}$ |
| max_depth | $\mathcal{U}\{3, 4, \ldots, 20\}$ |
| learning_rate | $e^{\mathcal{U}(-3,0)}$ |
| n_estimators | $\mathcal{U}\{50, 51, \ldots, 2000\}$ |
| min_child_weight | 1e-5, 1e-3, 1e-2, 1e-1, 1, 1e1, 1e2, 1e3, 1e4 |
| subsample | $\mathcal{U}(0.2, 0.8)$ |
| colsample_bytree | $\mathcal{U}(0.2, 0.8)$ |
| reg_alpha | 0, 1e-1, 1, 2, 5, 7, 10, 50, 100 |
| reg_lambda | 0, 1e-1, 1, 5, 10, 20, 50, 100 |

| **CatBoost** | |
| --- | --- |
| Parameter | Values |
| learning_rate | $e^{\mathcal{U}(-5,0)}$ |
| random_strength | $\mathcal{U}\{1, 2, \ldots, 20\}$ |
| l2_leaf_reg | $e^{\mathcal{U}(0, log(10))}$ |
| bagging_temperature | $\mathcal{U}(0.0, 1.0)$ |
| leaf_estimation_iterations | $\mathcal{U}\{1, 2, \ldots, 20\}$ |
| iterations | $\mathcal{U}\{100, 101, \ldots, 4000\}$ |

Table 9: Hyperparameter search spaces we used for Wild-time baselines.

**Underlying MLP (+ ERM, A-GEM, FT)**

| Parameter | Values |
|---|---|
| train_update_iter | 500, 1000, 2000, 3000, 4000, 5000, 6000, 7000 |
| lr | $e^{\mathcal{U}(-14,-4)}$ |
| use_scheduler | True, False |
| ft_scheduler_gamma | $\mathcal{U}(0.9, 1.0)$ |
| weight_decay | $0.0, 1e-5, 1e-2$ |
| early_stopping | True, False |
| early_stop_holdout | $0.1, 0.15, 0.2$ |
| early_stop_patience | $\mathcal{U}\{10, 11, \ldots, 30\}$ |

**LISA**

| Parameter | Values |
|---|---|
| mix_alpha | $e^{\mathcal{U}(0.5,4.0)}$ |
| cut_mix | True, False |

**Mixup**

| Parameter | Values |
|---|---|
| mix_alpha | $e^{\mathcal{U}(-5,0)}$ |

**EWC**

| Parameter | Values |
|---|---|
| gamma | $e^{\mathcal{U}(1.0,2.0)}$ |
| ewc_lambda | $e^{\mathcal{U}(0.5,2.0)}$ |

**GroupDRO-T**

| Parameter | Values |
|---|---|
| group_size | $\mathcal{U}\{1, 2, \ldots, 6\}$ |
| non_overlapping | True, False |
| group_loss_adjustments | None, 0.1, 0.5, 1.0 |
| group_loss_btl | True, False |

**SWA**

| Parameter | Values |
|---|---|
| swa_portion | $\mathcal{U}(0.5, 0.9)$ |
| swa_lr_factor | $\mathcal{U}\{1, 2, \ldots, 6\}$ |

**IRM-T**

| Parameter | Values |
|---|---|
| group_size | $\mathcal{U}\{1, 2, \ldots, 6\}$ |
| non_overlapping | True, False |
| irm_lambda | $\mathcal{U}\{1, 2, \ldots, 100\}$ |
| irm_penalty_anneal_iters | 0, 250, 500, 750, 100 |

**SI**

| Parameter | Values |
|---|---|
| si_c | $\mathcal{U}(0.05, 0.2)$ |
| epsilon | $\mathcal{U}(0.0005, 0.002)$ |

**CORAL-T**

| Parameter | Values |
|---|---|
| group_size | $\mathcal{U}\{1, 2, \ldots, 6\}$ |
| non_overlapping | True, False |
| coral_lambda | $\mathcal{U}(0.1, 1.0)$ |

Table 10: Preprocessing search spaces for Drift-Resilient TabPFN and TabPFN-base.

| Parameter | Search Space |
|---|---|
| model_type | single |
| N_ensemble_configurations | 16, None |
| preprocess_transforms | See Table 11 |
| softmax_temperature | log(0.75), log(0.8), log(0.9), log(0.95) |
| use_poly_features | True, False |
| max_poly_features | 50 |
| remove_outliers | -1, 7.0, 9.0, 12.0 |
| add_fingerprint_features | True, False |
| subsample_samples | 0.9, 0.99, -1 |

Table 11: Parameters and values for enumerate_preprocess_transforms function.

| Parameter | Values |
|---|---|
| names | ["safepower"], ["quantile_uni_coarse"], ["quantile_norm_coarse"], ["adaptive"], ["norm_and_kdi"], ["quantile_uni"], ["none"], ["robust"], ["kdi_uni"], ["kdi_alpha_0.3"], ["kdi_alpha_3.0"], ["safepower", "quantile_uni"], ["kdi", "quantile_uni"], ["none", "power"] |
| categorical_name | ["numeric", "ordinal_very_common_categories_shuffled", "onehot", "none"] |
| append_original | [True, False] |
| subsample_features | [-1, 0.99, 0.95, 0.9] |
| global_transformer | [None, "svd"] |

