# OpenReview forum: "Drift-Resilient TabPFN: In-Context Learning Temporal Distribution Shifts on Tabular Data"
_NeurIPS.cc/2024/Conference — NeurIPS 2024 poster_

### Official Review · Reviewer_m2Lp · 2024-07-07

**Soundness:** 2
**Presentation:** 2
**Contribution:** 3
**Rating:** 6
**Confidence:** 4

**Summary:**

This paper aims to address the temporal distribution shifts in Tabular data based on TabPFN. Concretely, the proposed method use two structural causal models (SCM) to model the prior for TabPFN, one for the gradual shift of inductive bias over time and the other is to model the shift of the first SCM. Empirical studies on various datasets illustrate the effectiveness of the proposed approach.

**Strengths:**

1. The paper aims to address temporal shifts in tabular data, which could be a useful setting in practice.
2. The proposed method intuitively makes sense.
3. Experiments on synthetic and real datasets have been tested. The code is released.

**Weaknesses:**

1. As far as I am concerned, CatBoost, XGboost, and LightGBM do not explicitly consider the temporal shift. The proposed method introduced additional sampled data from a temporal shift prior to PFN. I would doubt such a comparison is not that fair. Does any of the baselines in [1]  outperform the proposed method on the datasets used in this paper?
2. The experimental setting could be more detailed in the main paper.
3. It seems the proposed method is a simple modification to TabPFN that changes the SGM prior with additional temporal shift add-ons. So it's a little bit combinatorial to me, but not a huge problem.
4. Modelling the temporal shifts of inductive bias shares similarity with some continual learning approaches using meta-learning techniques, which lack a brief discussion, such as [2, 3, 4].

[1] Benchmarking distribution shift in tabular data with tableshift. Advances in Neural Information Processing Systems,36 (2024).

[2] Online fast adaptation and knowledge accumulation (osaka): a new approach to continual learning. Advances in Neural Information Processing Systems, 33:16532–16545, 2020.

[3] Reconciling meta-learning and continual learning with online mixtures of tasks. Advances in Neural Information Processing Systems, 32, 2019.

[4] On the stability-plasticity dilemma in continual meta-learning: theory and algorithm. Advances in Neural Information Processing Systems, 36 (2024).

**Questions:**

1. Why do you need the functional representation graph $\tilde{\mathcal{G}}$? What are the benefits of the original sampled graph?
2. Why do sparse shifts allow for causal reasoning (Line 195)?
3. How are the edge weights decided?
4. Why can SCM with NN-based causal mechanisms and nonlinear activations extrapolate values out of data distribution (Line 208)? Is this due to a causal mechanism or NN?

**Limitations:**

Yes.

---

> ### Author Rebuttal · Authors · 2024-08-07
>
> Dear Reviewer m2Lp,
>
> Thank you for dedicating your time to review our work and providing valuable feedback. We really appreciate your recognition that our intuitive approach could address a useful setting in practice.
>
> > As far as I am concerned, CatBoost, XGboost, and LightGBM do not explicitly consider the temporal shift. The proposed method introduced additional sampled data from a temporal shift prior to PFN. I would doubt such a comparison is not that fair. Does any of the baselines in [1] outperform the proposed method on the datasets used in this paper?
>
> We did compare to all applicable baselines from the Wild-Time benchmark in Table 6 in Section A.10.7 of the appendix. The Wild-Time benchmark is similar to the TableShift benchmark [1], but focused on temporal domain generalization (DG). We will more prominently highlight these methods in our camera ready.
>
> We compared to other DG methods in the appendix only, as traditional methods are the strongest baselines for out-of-distribution prediction on our Benchmark as can be seen in Table 6. This aligns with findings from both TableShift, you cite, [1] and Wild-Time benchmarks.
>
> Our comparison with DG methods in Table 6 also shows that the performance of DG methods significantly lags behind GBDTs and TabPFN. This discrepancy can be attributed to the nature of our studies, which involve small tabular datasets affected by temporal distribution shifts. Modeling distribution shifts on smaller datasets might be a very different problem from modeling on larger datasets. We'll add further justification for our baseline choices in the camera-ready version.
>
> > The experimental setting could be more detailed in the main paper.
>
> Thank you for this feedback. We have further improved the clarity of our experiments.
>
> 1. We have separated the quantitative results for synthetic and real-world datasets in the main paper, like previously done in the appendix.
> 2. We have included a brief discussion of the nature of our datasets in Section 4 of the main paper.
> 3. We have improved the plots in our qualitative analysis by coloring the probability of the most probable class at each point in the plot. These enhanced plots can already be seen in the provided demo code Colab notebook, and we have included one of these plots in the PDF in the global response to all reviewers.
>
> If you have any specific suggestions to further improve the comprehensiveness of our paper, we would greatly appreciate your input.
>
> > It seems the proposed method is a simple modification to TabPFN that changes the SGM prior with additional temporal shift add-ons. So it's a little bit combinatorial to me, but not a huge problem.
>
> Due to our novel approach to tackling the task of temporal domain generalization being very intuitive, we understand that our work might initially appear simple. However, the problem setting we aim to address and the detailed implementation of our approach are certainly non-trivial, extending the scope of TabPFN in a substantial way and addressing the gap in studying temporal distribution shifts on tabular data. Furthermore, we would argue that having an intuitive approach that significantly outperforms the baselines is a strength rather than a weakness.
>
> > Modelling the temporal shifts of inductive bias shares similarity with some continual learning approaches using meta-learning techniques, which lack a brief discussion, such as [2, 3, 4].
>
> Thank you for bringing this very relevant and interesting line of research to our attention. We will add a discussion of it to our camera-ready version.
> The difference between this line of work, continual meta-learning, and ours is that they assume there is a distribution shift in the datasets they train on (at the meta-level). Whereas we train (at the meta level) on a static distribution across datasets, but deal with a distribution shift within datasets. That is, they assume that the distribution over tasks changes over time, while we assume that the distribution over data points within each task changes over time.
>
> > Why do you need the functional representation graph $\tilde{\mathcal{G}}$ What are the benefits of the original sampled graph?
>
> In our paper, we introduce two distinct representations within structural causal models (SCMs) to fulfill specific functions. The traditional causal representation, denoted as $\mathcal{G} = (Z, R)$, follows the framework established by Pearl [a], where $Z$ represents causal mechanisms, and $R$ the causal relationships. In this model, each node value $z_i$ is determined through a function $f_i$ that integrates the values of its parent nodes $PA_i$ and an independent noise component $\epsilon_i$, mathematically expressed as $z_i = f_i({z_j\ | j \in PA_i}, \epsilon_i)$.
>
> The functional representation graph $\tilde{\mathcal{G}}$, however, then utilizes neural networks to model these assignment functions $f_i$ of our causal representation $\mathcal{G}$. By randomly sampling elements such as linear layers, activation functions, and Gaussian noises, this approach allows the dynamic generation of diverse functional relationships. The functional representation then allows for the propagation of noise through the network, with certain nodes designated as features while others serve as targets in the generated dataset. A key advantage of this method over handcrafted functions is its ability to model a broader range of functional relationships. For a detailed mathematical definition and potential areas for clarification, please refer to Section 3.1 of our paper. We would welcome further feedback to refine our explanations.
>
> *Due to the extensive rebuttal and feedback we wish to provide, we have split our response into two parts.*

---

> ### Author Response · Authors · 2024-08-07
> **Rebuttal by Authors (contd.)**
>
> > Why do sparse shifts allow for causal reasoning (Line 195)?
>
> For details, we refer to the foundational work of Perry et al. [b], which addresses the challenge of causal discovery from observed data. In traditional settings with i.i.d. data from non-shifted SCMs, the recovery of causal graphs is limited to the Markov equivalence class due to the symmetrical factorization of joint distributions.
>
> Perry et al. demonstrate that data generated from SCMs experiencing sparse shifts can distinctly break this symmetry, enabling the identification of causal structures beyond the Markov equivalence class. They provably show that sparse shifts serve as a learning signal for inferring causal relationships, whereas unrestricted shifts cannot be traced in the data.
> These results are important for our approach. They indicate that our transformer can learn to extract and utilize causal relationships from the sparse shifts in our prior. During pre-training, the model encounters these sparse shifts, allowing it to infer causal relationships. When presented with sparsely shifted data during training, the transformer can apply this learned causal reasoning during inference. This theoretical foundation underpins the effectiveness of our approach and supports the validity of our method as outlined in our paper.
>
> > How are the edge weights decided?
>
> The edge weights of a sampled functional representation graph are initially randomized using a well-known weight initialization technique (e.g., Xavier or Kaiming). We then determine randomly whether each causal relationship will shift over time. For the edges that are shifted, we randomly decide whether the shift will be multiplicative or additive. The parameters for scaling or shifting the weights are then sampled from our "hypernetwork", a secondary SCM, using the corresponding time index $t$ as input. This results in the functional representation graph for time $t$, from which instances can be sampled by inputting random noise. We hope this clarifies how the shifts are applied to the functional representation $\tilde{\mathcal{G}}$. For additional details, please refer to Section 3.2 and Algorithm 1 in A.7 of the appendix. If you have any suggestions for improving clarity, please feel free to share them with us. We are willing to revise this section to facilitate understanding while maintaining the right balance between intuition and mathematical rigor.
>
> > Why can SCM with NN-based causal mechanisms and nonlinear activations extrapolate values out of data distribution (Line 208)? Is this due to a causal mechanism or NN?
>
> This ability to extrapolate values out of the observed data distribution stems from our use of a prior that assumes data can be explained by a simple SCM with NN-based functions, rather than employing a specific trained neural network.
> In our framework, the first-order SCM models the data generation process, while the second-order SCM (which we initially termed a "hypernetwork") captures trends in how the first-order SCM's parameters change over time. This second-order SCM doesn't directly predict data points, but rather models the evolution of the data-generating process itself.
> The extrapolation occurs because the second-order SCM picks up on trends in weight changes that could explain the observed changes in the first-order SCM's target mapping. These trends are then extrapolated according to our prior, which favors simple trends (i.e., those that can be described by NN functions with few parameters).
> This approach allows for extrapolation that is both flexible (due to the use of neural network-based functions) and constrained (by the simplicity prior and the causal structure imposed by the SCMs). It's this combination that enables meaningful extrapolation beyond the observed data distribution, while maintaining the underlying causal relationships and favoring simpler explanations for distributional changes.
>
> We hope we have addressed all your questions and concerns. If we have addressed your concerns, we would very much appreciate it if you could consider increasing your score. Thank you again for your valuable feedback.
>
> ### References:
> - [a] Judea Pearl. Causality. Cambridge University Press, 2 edition, 2009.
> - [b] Ronan Perry, Julius Von Kügelgen, and Bernhard Schölkopf. Causal discovery in heterogeneous environments under the sparse mechanism shift hypothesis. In Proceedings of the 36th International Conference on Advances in Neural Information Processing Systems (NeurIPS’22), 2022.

---

> > ### Author Response · Authors · 2024-08-13
> >
> > Dear Reviewer m2Lp,
> >
> > We thank you again for the time and effort you invested in reviewing our submission, and hope our rebuttal has addressed your questions and concerns regarding the paper. However, if there is anything you still like to see addressed or clarified, we are more than happy to provide additional details until the end of the discussion period.
> >
> > Best regards, The Authors

---

> > > ### Comment · Reviewer_m2Lp · 2024-08-14
> > > **Score 4 to 6**
> > >
> > > Thanks for the authors' detailed reply.
> > >
> > > Most of my concerns are addressed. I raised the score accordingly.
> > >
> > > Based on your response, I recommend the following changes to improve the paper's clarity:
> > > 1. Add some examples for how to span the nodes and graph, that is, how you create $\tilde{z}_1^1$ and $\tilde{z}_1^2$ from $z_1$ and how you $\tilde{f}_2^1$. The current presentation in Section 3.1 is very obscure.
> > >
> > > 2. Move some important experimental results and details to the main paper.
> > >
> > > 3. Add discussion w.r.t related works and other explanations or verifications to your claims, as presented in the rebuttal.

---

> ### Author Response · Authors · 2024-08-14
> **Follow-up on Reviewer's Response**
>
> Dear Reviewer m2Lp,
>
> Thank you for your feedback and for raising your score in light of our revisions. We appreciate your recognition of the improvements we've made and your suggestions for further enhancing the clarity of our paper.
>
> > 1. Add some examples for how to span the nodes and graph, that is, how you create $\tilde{z}_1^1$ and $\tilde{z}_1^2$ from $z_1$ and how you $\tilde{f}_2^1$. The current presentation in Section 3.1 is very obscure.
>
> We thank you for this suggestion and will include these details in the camera-ready version to make the methodology more accessible to the reader.
>
> > 2. Move some important experimental results and details to the main paper.
>
> We are glad that you see the value of the additional experimental results that we have included in the appendix, and acknowledge that some of these results need to be highlighted more prominently in the main paper.
>
> As mentioned in our latest response to Reviewer LELq, we move a short discussion on the key findings related to the Wild-Time baselines as well as the results of the best performing DG method in Table 6, into the main paper. We also add a table of contents to the beginning of the appendix to provide an easier overview of the additional experiments included. In addition, we summarize the main results of the Time2Vec ablation in Section 4 - Impact of Time2Vec Preprocessing on our Model Performance (L305 - L307). Furthermore, we split Table 1 to present synthetic and real-world results separately, as previously done in the appendix, with corresponding discussions in Section 4 - Quantitative Results.
>
> We believe these changes provide more clarity to the reader and offer a better picture of the robustness of our approach. However, if there is anything beyond these changes that you believe should be included in the main text, we would be glad to consider those as well.
>
> > 3. Add discussion w.r.t related works and other explanations or verifications to your claims, as presented in the rebuttal.
>
> We appreciate that you found the revision of the related works section, as well as the other proposed clarifications and additional experiments beneficial. We will include these in the camera-ready version.
>
> Thank you once again for your feedback and for acknowledging our efforts to refine the paper.
>
> Best regards, The Authors

---

### Official Review · Reviewer_84bT · 2024-07-10

**Soundness:** 3
**Presentation:** 4
**Contribution:** 3
**Rating:** 7
**Confidence:** 4

**Summary:**

The authors propose an extension of the TabPFN framework such that the model can be resilient to distribution shifts during the inference phase by observing the shifted samples in-context. Akin to TapPFN, the pipeline involves pretraining based on an SCM, and the authors introduce a temporal aspect to the SCM construction such that the model will be aware that several common types of distribution shifts can be possible at inference time.

**Strengths:**

The paper was well structured and well written. The motivation of the work was described in a compelling way. The objective of the work was clearly described. The description of the SCM was sufficiently clear and intuitively it makes sense that the described approach should lead to an improvement in handling cases where a distribution shift is noted. All the figures were helpful in understanding the details of the method, and I found the visualization of the decision boundaries especially convincing. It seems there could be a real world impact in areas where TabPFN-like methods are a good fit based on this work.

**Weaknesses:**

I did not feel the work had any major structural weaknesses within the scope of what the authors wished to achieve.

A small suggestion that the title should imply that the work is about temporal domain shifts.

The “runs in seconds” clearly would not be true for all kinds of target scales, although it rightfully highlights the potential of the approach in some scenarios. I recommend adding a short qualifier on the scale where this is true.

Additional clarity on what exactly is the proposed input and output to the final model would have been appreciated in the methods section. For instance, does the model need to observe all previous intervals eg. [0 1 2] and the future intervals [3 4 5]? Only the previous intervals? Or just a trailing list of recent intervals? What is the justification that future intervals [4 5] will be available to the user if they are still in the present (domain 3)?

Due to some of the ambiguity, it was also not clear at a high level if a user needs to directly indicate to the model that a domain shift is generally expected at certain domain indices, or the model makes such a decision for every domain index internally.

“We approximated domain indices … based on features that encode temporal information, which we transformed into discrete intervals.” It seems that the accuracy attained by the models could vary a bit based on the way that the domains are approximated, unless the discretization was completely random uniform and guaranteed to be fair. Some additional assurance seems deserved on how this approximation was done.

I think the authors motivate the problem enough such that readers will be convinced that existing methods would not do well under distribution shifts. Given this, I think a more competitive benchmark could be to see how well baselines trained on small parts of the OOD do well on the OOD domains, and how this compares with the various ways of applying DSTabPFN. I would be curious to know if this was considered.

**Questions:**

Do the authors have any more considerations on the extent to which in-domain (ID) capabilities are affected when adopting the Distribution Shift TabPFN approach as opposed to the original TabPFN? This is hinted in line 287 but it seemed like an important point that gets at some quantitative and motivation-related questions. For instance, what is the false positive rate of the DSTabPFN guessing a distribution shift when there is none? What if there are some outliers in the inference phase despite the underlying distribution will continue longer term? Should users rely on the model to distinguish between one shift to another or only turn to DSTabPFN when a distribution shift is noticed? Some more intuition on this matter may convince readers further.

What is the rationale for the compute budget imposed in line 262? Generally eg. XGB would be parallelized with dozens of cores and requires a long time to cross validate & fit, but perhaps this was not necessary for the scale of the datasets explored.

It would be great to understand how the approach scales with the amount of samples from the SCM. For instance, are x5 more samples (taking x5 times longer to train) needed for the model to be robust to 5 different types of distribution shifts? Or is a similar accuracy to TabPFN possible with an equal number of samples while also being robust to shifts? Is the reason why DSTabPFN does not match the accuracy of TabPFN because it crosses into an overfit regime with too many samples?

A related analysis that could have been nice to have is to see how many “shots” in the shifted distribution are needed for DSTabPFN to adjust appropriately.

**Limitations:**

The work discusses limitations to a fair extent in the final section. I did not assess that the work could have a major negative societal impact.

---

> ### Author Rebuttal · Authors · 2024-08-07
>
> Dear Reviewer 84bT,
>
> Thank you very much for your thorough review and in-depth questions, which have sparked ideas for further analyses and follow-up work. We highly appreciate your positive feedback on the structure and clarity of our paper. We are glad that our approach was communicated comprehensively and that the visualizations were helpful for understanding. Additionally, we are pleased that we were able to compellingly motivate this important problem setting and demonstrate the potential real-world impact of our approach.
>
> > A small suggestion that the title should imply that the work is about temporal domain shifts.
>
> This is a great suggestion. We have adjusted the title to explicitly highlight this focus. The new title is "Drift-Resilient TabPFN: In-Context Learning *Temporal Distribution Shifts* on Tabular Data".
>
> > The “runs in seconds” clearly would not be true for all kinds of target scales, although it rightfully highlights the potential of the approach in some scenarios. I recommend adding a short qualifier on the scale where this is true.
>
> We completely agree that the "runs in seconds" claim is only applicable to certain dataset sizes. We have added a qualifier to clarify that these speeds are achieved within the dataset sizes evaluated in this work (e.g., approximately < 100 features, < 2500 instances).
>
> > Additional clarity on what exactly is the proposed input and output [...]
>
> This is an excellent question, as other approaches are often more restricted in this respect. Our approach only requires that a temporal domain is reported for all training instances and that the temporal domain is known for the sample we want to predict. Beyond this, all kinds of combinations are possible, including gaps and variable time differences within our training and testing domains (e.g., train on [1, 4, 5, 8] and test on [5, 14, 15]). Overall, the temporal domain is just a special feature of each dataset instance.
>
> Regarding the concern about future domains being available during test time, there is a misunderstanding. Future domains are never provided, as this would violate the principles of Domain Generalization and result in data leakage. If this question arises from Figure 5, where the decision boundary is plotted over time, we clarify that the model is trained only on the domains [0, 1, 2, 3] and then tested individually on [4], [5], and [6]. When predicting domain 5, the model does not have access to data from domains 4 or 6.
>
> > Does the user need to indicate to the model that a domain shift is generally expected at certain domain indices?
>
> No, the user does not need to indicate to the model that a domain shift is expected at certain indices. Our model internally reasons about the types of shifts present in the training data and determines when they occur, handling everything implicitly without needing external guidance. One insight here is that the weight shifts produced by the hypernetwork can remain constant across temporal domains, resulting in i.i.d. data generated by the data-generating SCM even as the temporal domain index in the dataset changes.
>
> > It seems that the accuracy attained by the models could vary a bit based on the way that the domains are approximated [...]
>
> This is a valid point. While our model can theoretically handle continuous intervals, making discretization unnecessary, computational limitations require discretization. Pre-training for continuous intervals is currently challenging as each shifted SCM involves a separate set of shifted weight matrices, complicating parallelization of the matrix multiplications.
>
> The discretized domain approximation should be approximately right as those can guide or misguide the model. However, our model does not fully rely on the domain indices. During pre-training, constant shifts can also be sampled, which helps the model learn to handle potential inaccuracies in domain indices during training. For synthetic datasets, no approximation was needed due to the availability of ground truth domain indices. For real-world datasets, we determined reasonable intervals based on the task at hand, without analyzing the data itself. This process is documented on a per-dataset basis in Section A.11 of the appendix. Thus, the approximation of domain indices can be seen as a downstream domain knowledge task specific to the data the model is applied to.
>
> > Test Baselines trained on small parts of the OOD domains
>
> While this is an interesting suggestion, it is currently beyond the scope of our study as it would violate the domain generalization setup and align more closely with a domain adaptation task.
>
> > Considerations on ID capabilities when adopting Drift-Resilient TabPFN compared to original TabPFN. Is DSTabPFN guessing a distribution shift when there is none?
>
> This is a great question. One consideration is that our prior also generates datasets without any shifts during pre-training. Therefore, if no shifts are present in the training portion of a dataset, our model is unlikely to extrapolate a non-existent shift. This is evidenced by the strong in-distribution (ID) performance of our model, which is only slightly less than the TabPFN base.
>
> However, your question has motivated an interesting experiment that we have already started setting up. In this experiment, we evaluate the performance of our model on a strict i.i.d. dataset, where we (1) report all instances as belonging to the same domain, and (2) report different instances as belonging to different domains, even though they are i.i.d. We will share these results as soon as possible during the author discussion period.
>
> *Due to the extensive rebuttal and feedback we wish to provide, we have split our response into two parts.*

---

> ### Author Response · Authors · 2024-08-07
> **Rebuttal by Authors (contd.)**
>
> > What if there are some outliers in the inference phase despite the underlying distribution will continue longer term?
>
> As for the inference phase, an outlier in the predictions would only affect the prediction of that specific outlier, as each prediction is made independently. If we consider a scenario where the outlier is later added to the training set with its ground truth, our model could indeed be influenced by it. However, it is important to note that outliers can also occur during pre-training, so we expect our model to have some capacity to handle them. This is a great suggestion that has sparked discussions among us as coauthors, and it would certainly be interesting to investigate this further!
>
> > What is the rationale for the compute budget imposed in line 262?
>
> The rationale behind this compute budget is that it is sufficient for the baselines to reach saturation. To demonstrate this, we conducted additional runs of the GBDT methods with a computational budget three times higher than the original, increasing it from 20 minutes to 1 hour. The results in Table 1 of the PDF in the global response to all reviewers show that the OOD performance remains approximately the same as before, indicating that the baselines do not benefit from additional computation time beyond this budget.
>
> > It would be great to understand how the approach scales with the amount of samples from the SCM.
>
> First, both the baseline TabPFN and Drift-Resilient TabPFN were pre-trained on the same number of datasets generated by their respective priors. However, due to the increased complexity of temporally shifted datasets, our model improves performance more slowly than the base TabPFN. During model development, we observed that longer pre-training phases are required to accurately capture the dynamics in our validation datasets, likely due to (1) the difficulty of the task and (2) the need to cover the extensive sample space of our prior fairly.
>
> The minor drop in ID performance is primarily due to our focus on OOD generalization during pre-training, where we only train on the ID task for a small percentage of the generated datasets. This is a tradeoff; we could improve ID performance by dedicating more training to it, at the expense of some OOD generalization.
>
> Regarding the overfit regime, there are two points: First, we can’t overfit during pre-training since we only train on synthetic data and the data used for inferenceis never seen. Second, the longer we train, the more our model adapts to the dynamics generated by our prior. In this sense, one could say we overfit to these dynamics.
>
> > Should users rely on the model to distinguish between one shift to another or only turn to DSTabPFN when a distribution shift is noticed.
>
> As of now, we recommend applying Drift-Resilient TabPFN when a temporal distribution shift is expected in the data.
> However, the question raises an interesting point about the model's applicability. While Drift-Resilient TabPFN is primarily designed for scenarios where temporal shifts are expected, it can potentially offer benefits even in cases where shifts are not explicitly anticipated. As described in Section 5 of our paper, we envision a more relaxed version of our approach that could extend the base TabPFN to employ a prior modeling sparse temporal shifts without requiring explicit domain indices. This extension could enhance robustness in standard classification tasks where temporal shifts may be present but not explicitly identified. Additionally, Drift-Resilient TabPFN could be integrated into stacking ensembles to leverage its shift-awareness alongside other models, and time series cross-validation could be employed to provide more robust performance estimates in temporal settings.
>
> > A related analysis that could have been nice to have is to see how many “shots” in the shifted distribution are needed for DSTabPFN to adjust appropriately.
>
> We appreciate this insightful question. During model development, we observed that adding instances from just one additional domain to the training set significantly improved predictions. In addition, our method demonstrated the ability to already extrapolate shifts far into the future with only a few domains included in the training dataset.
>
> To provide experimental insights on this, we have set up an experiment where we fix the target domain (e.g., the last domain in the dataset) and incrementally add one domain at a time to the training set, starting from two training domains. We will then plot our model's performance and calibration for the last domain based on the current training set. We share the results of this experiment as they are available.
>
> We hope to have addressed your questions adequately and welcome any further discussions you might have. If we have addressed your concerns, we would very much appreciate it if you could consider raising your score. Thank you again for your thoughtful review, which has sparked ideas for follow-up analyses.

---

> > ### Comment · Reviewer_84bT · 2024-08-11
> >
> > Thank you for the detailed response. I believe most of my initial questions have been resolved.
> > I don't want to create more work for the authors in a short period of time, the work as is seemed impactful to me and the future directions seem interesting.
> > After deliberation the work seems clearer to me (and I feel that some questions about related works have also been resolved in the other rebuttals), so I would be willing to raise the score.

---

> ### Author Response · Authors · 2024-08-13
> **Additional Experiments and Follow-Up to Reviewer Response**
>
> Dear Reviewer 84bT,
>
> We are glad that our previous clarifications addressed most of your questions, and appreciate that you have raised your score. As promised, we are now providing the results of the two outstanding experiments we mentioned.
> > Considerations on ID capabilities when adopting Drift-Resilient TabPFN compared to original TabPFN. Is DSTabPFN guessing a distribution shift when there is none?
>
> For this, we conducted an analysis on strict synthetic i.i.d. tabular classification datasets. We first instructed our model that all instances in the train and test split belong to the same domain, which aligns with the ground truth. Then, we informed the model that each group of 10 instances belongs to a different domain in increasing order. Therefore, the test instances are pretended to belong to different domains than the training instances. Our preliminary experiments across multiple datasets show that incorrectly reporting domain indices does not significantly impact our model's performance. Below is a table, showing the mean scores across three model initializations for one example dataset:
>
> | Metric | Same Domain (Mean ± 95% CI) | Different Domains (Mean ± 95% CI) |
> |--------|-----------------------------|----------------------------------|
> | ROC AUC | 0.9031 (0.0043) | 0.9030 (0.0045) |
> | ACC | 0.8017 (0.0259) | 0.7983 (0.0072) |
> | ECE | 0.0512 (0.0278) | 0.0531 (0.0389) |
> | F1 | 0.8006 (0.0323) | 0.7966 (0.0147) |
>
> These results indicate that our model does not overly rely on domain indices to infer a shift when there is none across the training data. We appreciate your suggestion to conduct this experiment. We will continue to analyze this further and include these findings in the appendix of the camera-ready submission.
>
> > A related analysis that could have been nice to have is to see how many “shots” in the shifted distribution are needed for DSTabPFN to adjust appropriately.
>
> For this analysis, we evaluated the performance of both our model and the baseline TabPFN by fixing the prediction to the last domain of a dataset and gradually increasing the number of domains seen during training. Our results confirm observations made during model development: our method requires significantly less training domains to accurately extrapolate shifts to the distant future, whereas the baseline TabPFN only has acceptable decision boundaries when given data close to the testing domain. Below is the results table for this experiment on the Intersecting Blobs dataset, which was discussed in the qualitative evaluation section of our paper. There, we fixed the predictions to domain $\mathcal{C}^{\text{test}} = \\{9\\}$ and increased the train set from training on domains $\mathcal{C}^{\text{train}} = \\{0, 1\\}$ to $\mathcal{C}^{\text{train}} = \\{0, 1, …, 8\\}$ gradually. The columns in the table represent $\mathcal{C}^{\text{train}}$ that the model was fitted on for the respective scores.
>
> | Metric | $\\{0,1\\}$ | $\\{0,…,2\\}$ | $\\{0,...,3\\}$ | $\\{0,…,4\\}$ | $\\{0,…,5\\}$ | $\\{0,…,6\\}$ | $\\{0,…,7\\}$ | $\\{0,…,8\\}$ |
> |--------|-------|-------|-------|-------|-------|-------|-------|-------|
> | ROC AUC (TabPFN$_{\mathrm{dist}}$) | 0.7975 | 0.9273 | 0.9994 | 0.9997 | 0.9999 | 1.0000 | 1.0000 | 1.0000 |
> | ROC AUC (TabPFN$_{\mathrm{base}}$) | 0.4551 | 0.4855 | 0.3495 | 0.4281 | 0.4581 | 0.7856 | 0.9996 | 1.0000 |
> | Acc. (TabPFN$_{\mathrm{dist}}$) | 0.4972 | 0.6556 | 0.9083 | 0.9722 | 0.9917 | 0.9944 | 1.0000 | 0.9917 |
> | Acc. (TabPFN$_{\mathrm{base}}$) | 0.5028 | 0.4194 | 0.3028 | 0.2667 | 0.2417 | 0.4833 | 0.9806 | 0.9944 |
> | ECE (TabPFN$_{\mathrm{dist}}$) | 0.2994 | 0.2782 | 0.2551 | 0.2649 | 0.1554 | 0.0555 | 0.0218 | 0.0105 |
> | ECE (TabPFN$_{\mathrm{base}}$) | 0.2840 | 0.2658 | 0.3404 | 0.4109 | 0.5202 | 0.2539 | 0.1865 | 0.0211 |
> | F1 (TabPFN$_{\mathrm{dist}}$) | 0.4972 | 0.6556 | 0.9083 | 0.9722 | 0.9917 | 0.9944 | 1.0000 | 0.9917 |
> | F1 (TabPFN$_{\mathrm{base}}$) | 0.5028 | 0.4194 | 0.3028 | 0.2667 | 0.2417 | 0.4833 | 0.9806 | 0.9944 |
>
> Thank you again for suggesting this valuable analysis. We will include the quantitative results for this experiment in the form of line plots, along with decision boundaries, in the appendix of the camera-ready version.

---

> > ### Comment · Reviewer_84bT · 2024-08-13
> >
> > The continued commitment by the authors is appreciated. The robustness of the model despite ablation of the domain indices is good to know. The shots experiment I think is also great to show, as it gives a practical sense of how responsive the model could be if used in real world settings.

---

### Official Review · Reviewer_LELq · 2024-07-11

**Soundness:** 3
**Presentation:** 3
**Contribution:** 3
**Rating:** 6
**Confidence:** 4

**Summary:**

The paper presents a modification of TabPFN that incorporates a novel prior to incorporate temporal shifts. In particular, the proposed method introduces an additional SCM that, through a temporal representation (Time2Vec), learns to modify the model parameters in response to shifts. The authors train a version of the proposed model and compare it to TabPFN, GBDT baselines, and standard ERM and SWA baselines.

Overall, the paper's presentation is good (but could be improved), and this appears to be a logical addition to the TabPFN method. A method capable of zero-shot transfer to new distributions seems especially well-motivated, given that no labeled data from the target domain is available in many real-world scenarios. However, some revisions and additions to the paper are badly needed. For example, the literature review is inadequate and needs a complete rewrite; there are no domain generalization or robustness methods included in the experimental results (besides the proposed method), and the ablation study is too limited (in fact, the current results suggest that there is no advantage to the Time2Vec addition, which raises questions about the overall approach). I think that with extensiv revisions this paper could be brought up to acceptance level and am open to raising my score, but the scope of additions and revisions feels on the borderline of too much for a simple camera-ready phase.

**Strengths:**

Full review here:

# Major comments

* The paper is framed in a way that I find somewhat misleading: the title, abstract, and main text repeatedly frame the current work as an approach to "distribution shift". However, the approach is limited to strictly *temporal* shifts, with no discussion or demonstration of its extensibility to non-temporal shifts. This introduces some unnecessary confusion into the paper and its potential applications -- please either clarify this in the title/abstract/text (preferred), or add a thorough discussion why this should be considered a general method for distribution shift (less preffered unless the authors can already demonstrate this capacity, in which case the current framing does indeed fit).

* I like that the paper unifies all shifts (covariate, label, concept) under a single framework. This is a limitation of some (but not all) existing approaches to domain generalization, and it seems to be a clear advantage of this approach which the authors rightly highlight. However, it would also be useful to see a set of controlled experiments which separately vary these different components, to understand how the model's performance varies as different forms of shift are present; these impacts likely depend on the model's pretraining distribution in subtle ways).

* The "related work" section neither surveys relevant related work at a level even close to comprehensive, nor does it actually discuss related works relevant to understanding the current paper. In my opinion, this section should be completely rewritten. For example:  (1) the paper does not discuss related benchmarking or empirical studies such as Shifts [1] and Shift 2.0, Wild-Tab [2], TableShift [3] (already cited but not mentioned in related work despite applicable findings), and WhyShift [4]. All of these should be discussed and related to the current work. (Concurrent work TabRed [8] seems also relevant, but of course was published after this submission.) (2) the paper does not mention at all the many previously-proposed methods for robust learning and domain generalization, including those covered in the previous work. The only methods mentioned, strangely, are two methods that the authors state are *excluded* from the current study. (3) Many works have recently proposed changes to or conducted  empirical studies of TabPFN e,g, [5, 6, 7]; it would be useful to position the current work in relation to these studies.

* *Ablation study*: The ablation study should be expanded, and discussed in the main text. In particular: (1) the current results show no benefit from the Time2Vec addition, which seems to suggest that it shouldn't be included in the model; (2) the authors seem to have only conducted a single run, of the ablated model despite this evidence -- a number of trials equal to the other experiment seems in order. (3) please discuss the results in the main text, even if only in a sentence or two that state the conclusion from the experiment.

* *Resources for baselines*: The authors restrict all baselines to 1200 seconds on 8CPU/1GPU, but the proposed method is trained for 30 epochs on 8 GPUs with a random search over 300 configurations. This setup seems to starve the baselines of resources. Please either provide evidence that the baselines have in fact saturated (i.e. they would not be improved from further tuning) or allocate similar resources to the baselines as to the main model -- otherwise it is impossible to distinguish whether the proposed method simply benefits from more compute + hyperparameter tuning vs. whether it is in fact an improvement.

* *Lack of relevant baselines:* I was quite surprised to see that the authors do not include any domain generalization or robustness methods in their study. Open-source implementations of many relevant methods are available, including (I believe) through Wild-Time and TableShift. Please comment on why these were not included (less preferred) or add them to the experimental results (preferred), also conducting fair hyperparameter tuning as described above.

# Minor comments

* More details on the data should be in the main text. What are the datasets, where are they drawn from, why do the authors use 8 synthetic and 10 real-world? None of these are given, and so it is hard to contextualize the results or assess the reliability of the empirical study.

* It is interesting that CatBoost performs quite strongly on the proposed methods, particularly ID on the synthetic data. This could perhaps be worth mentioning in the main text, as it is in line with the findings of other tabular studies (i.e. CatBoost tends to slightly outperform XGBoost and LightGBM).

# Typos etc

* Abstract: "with even smaller gains for tabular data" - this is unclear, please revise.

* L48-49: "Building on..." this sentence is a fragment, please revise.

* Table 1: I don't see the acronym "SWA" defined in the paper (ERM is not defined either, but this is more familiar to most readers).

* L303: there are 2 lines hanging onto Page 9; the figure/table positioning should be changed to avoid this.

# References

[1] Andrey Malinin, Neil Band, Yarin Gal, Mark Gales, Alexander Ganshin, German Chesnokov, Alexey Noskov, Andrey Ploskonosov, Liudmila Prokhorenkova, Ivan Provilkov, et al. Shifts: A dataset of real distributional shift across multiple large-scale tasks. In Thirty-fifth Conference on Neural Information Processing Systems Datasets and Benchmarks Track (Round 2), 2022.

[2] Kolesnikov, Sergey. "Wild-Tab: A Benchmark For Out-Of-Distribution Generalization In Tabular Regression." arXiv preprint arXiv:2312.01792 (2023).

[3] Gardner, Josh, Zoran Popovic, and Ludwig Schmidt. "Benchmarking distribution shift in tabular data with tableshift." Advances in Neural Information Processing Systems 36 (2024).

[4] Liu, Jiashuo, et al. "On the need for a language describing distribution shifts: Illustrations on tabular datasets." Advances in Neural Information Processing Systems 36 (2024).

[5] Breejen, Felix den, et al. "Why In-Context Learning Transformers are Tabular Data Classifiers." arXiv preprint arXiv:2405.13396 (2024).

[6] Feuer, Benjamin, et al. "TuneTables: Context Optimization for Scalable Prior-Data Fitted Networks." arXiv preprint arXiv:2402.11137 (2024).

[7] Ma, Junwei, et al. "In-Context Data Distillation with TabPFN." arXiv preprint arXiv:2402.06971 (2024).

[8] Rubachev, Ivan, et al. "TabReD: A Benchmark of Tabular Machine Learning in-the-Wild." arXiv preprint arXiv:2406.19380 (2024).

**Weaknesses:**

See above.

**Questions:**

See above.

**Limitations:**

Yes.

---

> ### Author Rebuttal · Authors · 2024-08-07
>
> Dear Reviewer LELq,
>
> Thank you for your thorough review and constructive feedback. We appreciate your recognition of our work's potential and your openness to raising your score. We have worked very hard during the rebuttal to answer your concerns and made major progress in rewriting the “related work” section. We have carefully considered your comments and have addressed them as follows:
>
> ### 1. Framing of the Paper
> > The paper is framed in a way that I find somewhat misleading: the title, abstract, and main text repeatedly frame the current work as an approach to "distribution shift". However, the approach is limited to strictly temporal shifts, with no discussion or demonstration of its extensibility to non-temporal shifts. This introduces some unnecessary confusion into the paper and its potential applications -- please either clarify this in the title/abstract/text (preferred), or add a thorough discussion why this should be considered a general method for distribution shift (less preffered unless the authors can already demonstrate this capacity, in which case the current framing does indeed fit).
>
> We fully agree that the focus on temporal distribution shifts should be clear from the start. Therefore, we have revised both the title and the abstract to explicitly highlight this focus. The new title is "Drift-Resilient TabPFN: In-Context Learning *Temporal Distribution Shifts* on Tabular Data".
>
> ### 2. Controlled Experiments on Different Types of Shifts
>
> > I like that the paper unifies all shifts (covariate, label, concept) under a single framework. This is a limitation of some (but not all) existing approaches to domain generalization, and it seems to be a clear advantage of this approach which the authors rightly highlight. However, it would also be useful to see a set of controlled experiments which separately vary these different components [...]
>
> We appreciate your positive feedback on the unified approach to handling various types of shifts. In fact, we were also quite interested in the decision boundaries of our models on the different types of shifts, which motivated some of the synthetically generated datasets. In the provided Colab notebook, we give readers easy access to view the decision boundaries on these datasets. For reference, we can categorize the synthetic datasets (based on their id) according to their respective types or combinations of shifts:
>
> - **Prior Probability Shift:** test: [1]
> - **Covariate Shift:** valid: [3]
> - **Concept Shift:** test: [5, 6, 7]
> - **Concept Shift + Covariate Shift:** valid: [1,2], test: [0, 2]
> - **All Types of Shifts:** valid: [0]
>
> In addition, we now provide three exemplary decision boundaries for each type of shift individually in Figure 1 of the PDF in the global response to all reviewers. However, since we have only few datasets per shift type, quantitative results per individual shift would not allow for significant conclusions.
>
> Regarding the strength of the shifts, we conducted a detailed analysis in Section A.10.6 of the appendix. There, we assessed the impact of combined shifts across both synthetic and real-world datasets. Our findings indicate that as the strength of the shifts increases, Drift-Resilient TabPFN's performance remains more robust and keeps higher scores compared to the baselines.
>
> ### 3. Related Work Section
> > The "related work" section neither surveys relevant related work at a level even close to comprehensive, nor does it actually discuss related works relevant to understanding the current paper. [...] We appreciate your feedback on the related work section as well as the pointers to more related work.
>
> We only included a rather short section on related work as the space constraints for our initial submission were very tight. For the camera-ready version we have more space, though, and thus have already started rewriting the related work section from the ground up and will provide it here as a comment in the discussion period as soon as it is ready.
>
> In our rewrite of the related work section we are integrating work on domain generalization outside of temporal domain shifts, as you suggest, as well as robustness methods, including the benchmarking studies and empirical works you mentioned (Shifts, Shift 2.0, Wild-Tab, TableShift, WhyShift, and the recent TabRed). In addition, we position our work in relation to recent empirical studies of TabPFN.
>
> *Due to the extensive rebuttal and feedback we wish to provide, we have split our response into two parts.*

---

> ### Author Response · Authors · 2024-08-07
> **Rebuttal by Authors (Part 2)**
>
> ### 4. Expanded Ablation Study
>
> > Ablation study: The ablation study should be expanded, and discussed in the main text. In particular: (1) the current results show no benefit from the Time2Vec addition [..](2) the authors seem to have only conducted a single run, of the ablated model [..] (3) please discuss the results in the main text [..]
>
> You are right that our ablation does not show a significant improvement due to the Time2Vec encoding. (1) We chose to include Time2Vec based on a large HPO search, as it performed best there, even though the improvement is not significant. (2) We acknowledge the limitations of our initial single-run ablation study. To ensure statistical significance, we are conducting two additional runs, scheduled for completion between August 11-12, 2024. (3) We will share these results in the rebuttal once available and will include them in the main paper. Additionally, we will add the following sentence in L307 of the Experiment section, summarizing the main insights from our ablation study:
>
> “[...] The ablation reveals that while Time2Vec provides slight improvements, the substantial performance gains are to be attributed to the prior construction used during the model's pre-training phase. [...]”
>
> Beyond the Time2Vec ablation, we are open to suggestions for other parts to ablate. The time-dependent data-generating SCM and the hypernet for weight shifts were primarily tuned through hyperparameter optimization on validation datasets. Given the involvement of about 50 hyperparameters and the intensive computational resources required for pre-training, it is challenging to find a meaningful ablation that would provide valuable insights into our model's performance.
>
> ### 5. Resources for Baselines and Drift-Resilient TabPFN
>
> > Resources for baselines: The authors restrict all baselines to 1200 seconds on 8CPU/1GPU, but the proposed method is trained for 30 epochs on 8 GPUs with a random search over 300 configurations. This setup seems to starve the baselines of resources. Please either provide evidence that the baselines have in fact saturated (i.e. they would not be improved from further tuning) or allocate similar resources to the baselines as to the main model -- [...]
>
> We would like to clarify the resource allocation for Drift-Resilient TabPFN. As mentioned in lines 277-281, the pre-training of our model, which involved 30 epochs on 8 GPUs and preprocessing optimization over 300 configurations, was only done once in advance. This pre-training was conducted solely on synthetic data generated by our prior, and the preprocessing optimization was performed on our validation datasets. All these steps can be considered as part of the algorithm development process and have to be done only once. The same model was applied to each test dataset and can be used for new, unseen datasets.
>
> The actual training of each real test dataset as well as the inference was done on 8 CPUs and 1 GPU, identical to the baselines. For training and inference on the test datasets, our model was on average 110 times faster compared to the baselines.
>
> To further address the issue of baseline saturation, we trained the three best-performing baselines (CatBoost, XGBoost, and LightGBM) for 3600 seconds and observed nearly identical out-of-distribution performance. The table for these extended baseline runs can be found in Table 1 included in the PDF of the global response to all reviewers.
>
> Regarding neural network approaches: Previous works, such as TableShift and WildTime, have shown that no other DG method consistently outperforms GBDTs on tabular data.

---

> ### Author Response · Authors · 2024-08-07
> **Rebuttal by Authors (Part 3)**
>
> ### 6. Inclusion of relevant baselines
> > Lack of relevant baselines: I was quite surprised to see that the authors do not include any domain generalization or robustness methods in their study. [...]
>
> We had already included the strongest among these methods in our main results (Table 6) and fully in the appendix (Section A.10.7 and Section A.10.8), referenced from our main text. We will more prominently feature these methods in our camera ready. Due to their weak performance (as previously found by the authors of the TableShift and WildTime benchmark) we did not discuss these results in great detail. As you mentioned, Wild-Time provides open-source implementations, and we have spent considerable time adapting these implementations to work within our evaluation framework. We evaluated all Wild-Time methods applicable to tabular data and included the best-performing methods (classical ERM and SWA) in the results table of the main paper.
>
> However, as noted previously, due to the small size of tabular datasets and the resulting limited training instances, these methods do not perform nearly as well as GBDTs or TabPFN. This finding is consistent with previous results from Wild-Time and TableShift, which indicate that none of the DG methods consistently outperform GBDTs. We chose to focus on Wild-Time rather than TableShift because, while the evaluated methods largely overlap, Wild-Time specifically focuses on temporal DG, aligning more closely with the aims of our study.
>
> ### Minor details:
> We appreciate your feedback on providing more details about the datasets in the main text. We have now included the following paragraph in Section 4 - Experiments in line 251 to address this:
>
> “[...] While some of these datasets have been analyzed in previous work, there has been no comprehensive benchmark focusing on small tabular datasets undergoing distribution shifts. To address this gap, we carefully selected or generated a diverse range of datasets that exhibit temporal distribution shifts. The non-generated datasets were selected from open dataset platforms or previous work in DG. [...]”
>
> Additionally, we would like to point out that Section A.11 of the appendix provides an in-depth discussion of each dataset. This includes the nature of each dataset, the types of shifts they contain, the subsampling performed, as well as the approximation of domain indices.
>
> We found CatBoost's strong ID performance interesting as well. However, due to our focus on out-of-distribution performance, we did not highlight it further in our evaluation.
>
> Thank you for pointing out the typos and minor mistakes; we have gladly corrected them.
>
> Overall, we believe your in-depth review and the corresponding changes have significantly improved our paper, and we would be grateful if you could raise our score. Thank you again for your valuable feedback!

---

> > ### Comment · Reviewer_LELq · 2024-08-12
> > **Follow up to author response**
> >
> > Thank you to the authors for the detailed response. I have reviewed both the authors' overall response, and the authors' individualized response to my review. The authors seem to have addressed some of my concerns around title and framing, along with the resources dedicated to baselines (their findings wrt saturation are in line with other tabular studies).
> >
> > * **Framing**: It is hard to assess the degree of the authors' success in "rewriting the related work section from the ground up" as the new section does not appear to have been shared. I unfortunately cannot give much consideration to a promise to rewrite this section which is currently very lacking for reasons I outline in my initial review - but the related work section is perhaps not the most significant issue raised in the review (although it is a major one). I have similar reservations about the authors' promised revisions regarding dataset details in the main text.
> >
> > * **Ablation study aind Time2Vec (T2V)**: Similarly, I do not see updated ablation study results. The authors acknowledge that their submitted result "does not show a significant improvement due to the Time2Vec encoding". Indeed, T2V is worse according to ECE, and only improves AUC by 0.001 (with an error of +/- 0.006) according to Table 2. The other results (accuracy, F1) in this table are on the edge of statistical significance. Again, there are also no error estimates for the no-T2V variant because the authors only perform one iteration. This to me is a major weakness of the ablation study, as this is the purpose of an ablation study (to identify which components of a new method do and do not contribute to its performance). I remain concerned about adding the T2V component to this model when it appears to do little, if nothing at all, and the limited/partial experimental results do not help to assess the extent of this issue.
> >
> > * **Missing domain generalization (DG) baselines**: The author response says that "Previous works, such as TableShift and WildTime, have shown that no other DG method consistently outperforms GBDTs on tabular data." This is not mentioned in the submitted version of the paper to my reading, which is a part of why the paper's comparison to related work seemed to lacking. Additionally, while I agree that these works provide a strong prior, these other studies are not a substitute for performing these comparisons on the authors' data -- particularly when this is a domain shift method, and there are no domain shift or domain generalization baseline methods in the empirical experiments at all (the authors only compare to vanilla supervised methods like XGBoost and ERM which are not designed for domain shift).
> >
> > I remain open to revising my score upward, despite these outstanding issues. I would like to discuss with the reviewers during the dialogue window.

---

> ### Author Response · Authors · 2024-08-12
> **Follow up to reviewer response (Part 1)**
>
> Dear Reviewer LELq,
>
> Thank you for your response and for your continued engagement with our work.
>
> ### Framing
> > It is hard to assess the degree of the authors' success in "rewriting the related work section from the ground up" as the new section does not appear to have been shared. I unfortunately cannot give much consideration to a promise to rewrite this section which is currently very lacking for reasons I outline in my initial review - but the related work section is perhaps not the most significant issue raised in the review (although it is a major one). I have similar reservations about the authors' promised revisions regarding dataset details in the main text.
>
> We appreciate your comments regarding the related work section and dataset details. To address your concerns, we have completely rewritten the related work section and would now like to share these updates with you. We’ve added a global comment for all reviewers in the top, showcasing the new related work section. It now includes discussions on related benchmarks, DG methods, and recent studies on TabPFN. We would greatly value any feedback you could provide to ensure we’ve effectively addressed the issues you previously highlighted.
>
> We did not include the works by Breejen et al. [5] and Rubachev et al. [8], as they were published after our submission. However, if you think it would be helpful to include these references, we are more than willing to reconsider and incorporate them.
>
> > [5] Breejen, Felix den, et al. "Why In-Context Learning Transformers are Tabular Data Classifiers." arXiv preprint arXiv:2405.13396 (2024).
>
> > [8] Rubachev, Ivan, et al. "TabReD: A Benchmark of Tabular Machine Learning in-the-Wild." arXiv preprint arXiv:2406.19380 (2024).
>
> Regarding the revisions to the dataset details, we have incorporated the changes, which were already detailed in our rebuttal. Specifically, in Section 4 - Experiments (line 251), we added the following paragraph:
>
> “[...] While some of these datasets have been analyzed in previous work, there has been no comprehensive benchmark focusing on small tabular datasets undergoing distribution shifts. To address this gap, we carefully selected and generated a diverse range of datasets that exhibit temporal distribution shifts. The datasets were selected from open dataset platforms or previous work in DG. [...]”
>
> Additionally, as mentioned in our rebuttal, all dataset details are thoroughly documented in Section A.11 of the appendix. We hope these revisions meet your expectations, but we would be glad to hear if there’s anything more you would like to see addressed.
>
> ### Ablation study and Time2Vec (T2V):
> > Similarly, I do not see updated ablation study results. The authors acknowledge that their submitted result "does not show a significant improvement due to the Time2Vec encoding". Indeed, T2V is worse according to ECE, and only improves AUC by 0.001 (with an error of +/- 0.006) according to Table 2. The other results (accuracy, F1) in this table are on the edge of statistical significance. Again, there are also no error estimates for the no-T2V variant because the authors only perform one iteration. This to me is a major weakness of the ablation study, as this is the purpose of an ablation study (to identify which components of a new method do and do not contribute to its performance). I remain concerned about adding the T2V component to this model when it appears to do little, if nothing at all, and the limited/partial experimental results do not help to assess the extent of this issue.
>
> We fully understand your concerns regarding the missing runs for the no-T2V variant, and agree that these should be included. As promised, we have been running the two outstanding experiments, and while they have not yet completed due to cluster issues, the results should be available by tomorrow morning. We will share these results with you as soon as they are ready.

---

> > ### Author Response · Authors · 2024-08-12
> > **Follow up to reviewer response (Part 2)**
> >
> > ### Missing domain generalization (DG) baselines:
> > > The author response says that "Previous works, such as TableShift and WildTime, have shown that no other DG method consistently outperforms GBDTs on tabular data." This is not mentioned in the submitted version of the paper to my reading, which is a part of why the paper's comparison to related work seemed to lacking.
> >
> > We agree that the comparison to related work and its findings should have been more thoroughly discussed in the initial submission. To address this, we now include a discussion of these studies in the rewritten “Related Work” section.
> >
> > > Additionally, while I agree that these works provide a strong prior, these other studies are not a substitute for performing these comparisons on the authors' data -- particularly when this is a domain shift method, and there are no domain shift or domain generalization baseline methods in the empirical experiments at all (the authors only compare to vanilla supervised methods like XGBoost and ERM which are not designed for domain shift).
> >
> > We completely agree that prior work has to be re-evaluated by us and we cannot rely just on their reported findings. As we said in the initial rebuttal, we have rerun all of the baselines in the Wild-Time benchmark on our data to gather results for Table 6 of the appendix. Moreover, we included the two best-performing methods from the Wild-Time benchmark in the main results in Table 1.
> >
> > Our findings indicate that, due to the specific context of small-tabular datasets subject to temporal distribution shifts, neural network approaches struggle to generalize effectively because of the limited amount of training instances. Standard ERM and SWA performed best among the Wild-Time methods, which is consistent with the findings of previous work. For a more detailed argument, please refer to point 6, "Inclusion of relevant baselines," in our initial rebuttal response.
> >
> > We look forward to your feedback on these updates and any further suggestions you might have.

---

> > > ### Comment · Reviewer_LELq · 2024-08-12
> > >
> > > Thank you to the authors for the detailed follow-up.
> > >
> > > The rewritten related work section is a huge improvement and addresses my concerns with that part of the paper. In particular, it properly surveys related work (wrt tabular benchmarking, domain generalization methods, and TabPFN variants) and I think is also quite useful in more properly positioning the work.
> > >
> > > I understand that it is difficult to add so many baselines and to add more iterates of the ablation study during the revision window. However, I also feel that these are now the main missing piece of the paper, as they are critical to empirically assessing the proposed method. I will increase my score to 6.

---

> > > > ### Author Response · Authors · 2024-08-13
> > > > **Clarification and Follow-Up on Baseline Evaluation in Response to Reviewer Feedback**
> > > >
> > > > Dear Reviewer LELq,
> > > >
> > > > Thank you very much for your feedback on the related work, and we are happy that you like it. Regarding the baselines, we believe that there might be a misunderstanding that we would like to resolve. We *have evaluated all applicable Wild-Time methods* (ERM, FT, EWC, SI, A-GEM, CORAL-T, GroupDRO-T, IRM-T, Mixup, LISA, SWA) on our Test-Datasets, including performing hyperparameter optimization as described in our initial rebuttal response.
> > > >
> > > > The results of this evaluation are in Section A.10.7 *”Quantitative Analysis: Comparison Against Wild-Time Methods”* of the appendix and we also did a comparison with DRAIN and GI on one dataset in Section A.10.8. of the appendix.
> > > > We would like to understand what you exactly believe is missing? Is there a particular baseline that is not represented in our evaluation, or would you like this section to be inside the main paper? In our initial submission, we decided against this due to the size constraints of the paper and the results being uncompetitive.
> > > >
> > > > We hope that we can clear the misunderstanding and look forward to your response.

---

> > > > > ### Comment · Reviewer_LELq · 2024-08-13
> > > > >
> > > > > Thanks to the authors for the clarification.
> > > > >
> > > > > I indeed did not see these results (they are on page 25 of the paper deep in the appendix, and if they are mentioned in the main text, I perhaps missed it). I was also a bit confused by the author response (it was unclear that these were *already* in the paper, my interpretation was that the authors had *added* them to the paper in response to the discussion). In any case, these are important results and should absolutely be in the main text - these are the most appropriate baselines for the current method. I don't see why they should be buried in the appendix just because they are negative results (and indeed, it makes the proposed method look much stronger). The fact that this reviewer completely missed these results even on a careful reading of the paper indicates that it would probably be helpful to also revise the main text, and perhaps even abstract/intro, to clearly signpost that these baselines are included.
> > > > >
> > > > > I will further raise my score.
> > > > >
> > > > > One final note: the authors say in the response above that
> > > > >
> > > > > > Our findings indicate that, due to the specific context of small-tabular datasets subject to temporal distribution shifts, neural network approaches struggle to generalize effectively because of the limited amount of training instances
> > > > >
> > > > > I am not sure whether the experiments support this conclusion; it assumes that the cause is the sample complexity of the DG methods (which there are no experiments designed to assess, on my reading) and also that this failure is somehow due to their neural network architecture (also not demonstrated with experiments, this would require conducting a controlled experiment with different architectures in DG models, which is probably far beyond the scope of this paper). Are the ERM and SWA methods also neural networks, or are they strictly linear models? I would suggest that the authors offer this as a hypothesis, but be clear that this is speculative and not suggest it is somehow "indicated" by their results.

---

> > > > > > ### Author Response · Authors · 2024-08-13
> > > > > > **Follow-up on Reviewer's Response + Extended T2V Ablation (Part 1)**
> > > > > >
> > > > > > Dear Reviewer LELq,
> > > > > >
> > > > > > We are pleased that this discussion has clarified the misunderstanding and that you recognize how these baselines further strengthen our proposed method.
> > > > > >
> > > > > > We have indeed referenced these results in the main paper within Section 4 — Baseline Setup (L260 - L261), stating:
> > > > > >
> > > > > > “[...] Methods from the  Wild-Time benchmark are examined separately and detailed in Section A.10.7 of the Appendix. [...]”
> > > > > >
> > > > > > However, we agree that this could be extended, and the results are located quite deep in the Appendix. Unfortunately, due to space constraints, we are not able to include the whole results table of Wild-Time into the main part of our work. Nevertheless, (1) we add a table of contents at the beginning of the Appendix and (2) reposition the Wild-Time results so they are more prominently featured. Additionally, (3) we add the best DG method alongside ERM and SWA in Table 1. Furthermore, (4) we add a short discussion of our Wild-Time findings in Section 4 - Quantitative Evaluation, and (5) we explicitly list Wild-Time as a baseline in the abstract.
> > > > > >
> > > > > > We believe that the revisions make these important baselines more accessible to readers and further highlight the robustness of our approach.
> > > > > >
> > > > > > > I am not sure whether the experiments support this conclusion; it assumes that the cause is the sample complexity of the DG methods (which there are no experiments designed to assess, on my reading) and also that this failure is somehow due to their neural network architecture (also not demonstrated with experiments, this would require conducting a controlled experiment with different architectures in DG models, which is probably far beyond the scope of this paper).
> > > > > >
> > > > > > We fully agree that this statement in our response should have included a qualifier like “possibly” or “likely”. Furthermore, we are aware that this is mostly speculation and should only be used as a possible explanation that needs to be validated before stating it as a fact. For this reason, we included a more cautious statement when writing our paper (L736 - L740), where we describe this difference as "likely".
> > > > > >
> > > > > > > Are the ERM and SWA methods also neural networks, or are they strictly linear models?
> > > > > >
> > > > > > Yes, both the ERM and SWA methods, along with the other methods in the Wild-Time Benchmark, were applied to an MLP. The hyperparameters for these models were optimized following the same process we applied to the other baselines.

---

> ### Author Response · Authors · 2024-08-13
> **Follow-up on Reviewer's Response + Extended T2V Ablation (Part 2)**
>
> In addition, we are pleased to now provide the results from the additional iterations of the no-T2V ablation, which are presented in the table below. This table is an extension of Table 2 from our initial submission, now including the mean and confidence intervals across three initializations for the no-T2V variant.
>
> | **Model**                | **Variant**           | **Acc. ↑ (OOD)**           | **Acc. ↑ (ID)**            | **F1 ↑ (OOD)**             | **F1 ↑ (ID)**              | **ROC ↑ (OOD)**            | **ROC ↑ (ID)**             | **ECE ↓ (OOD)**            | **ECE ↓ (ID)**             |
> |--------------------------|-----------------------|----------------------------|----------------------------|----------------------------|----------------------------|----------------------------|----------------------------|----------------------------|----------------------------|
> | **TabPFN-dist** | all dom. w. ind.      | **0.744** (.018)           | 0.879 (.012)               | **0.689** (.028)           | 0.837 (.022)               | **0.832** (.018)           | 0.932 (.002)               | **0.091** (.006)           | 0.074 (.014)               |
> | **No T2V**               | all dom. w. ind.      | 0.742 (.004)               | 0.877 (.007)               | 0.685 (.002)               | 0.834 (.014)               | **0.832** (.004)           | 0.931 (.009)               | 0.093 (.009)               | 0.071 (.005)               |
> | **TabPFN-base** | all dom. w. ind.      | 0.688 (.010)               | **0.885** (.010)           | 0.620 (.012)               | **0.847** (.017)           | 0.786 (.007)               | **0.935** (.010)           | 0.119 (.006)               | **0.067** (.005)           |
> |                          | all dom. wo. ind.     | 0.645 (.011)               | 0.852 (.016)               | 0.579 (.014)               | 0.801 (.020)               | 0.736 (.001)               | 0.914 (.007)               | 0.202 (.011)               | 0.076 (.007)               |
> |                          | last dom. wo. ind.    | 0.670 (.005)               | 0.867 (.004)               | 0.609 (.004)               | 0.823 (.011)               | 0.760 (.003)               | 0.915 (.019)               | 0.181 (.003)               | 0.128 (.007)               |
>
> Upon reviewing these results, we acknowledge that the impact of the T2V component appears minimal and statistically insignificant. However, it is worth noting that, aside from the ROC metric, which remains on par, each OOD performance metric shows a slight improvement in mean values when T2V is included. Moreover, we try to be reproducible in using the best configuration our HPO found, which included T2V.
>
> We believe that our final response has addressed all outstanding concerns. We noticed that the mentioned second score increase has not yet been reflected in the review, and with the discussion period ending shortly, we wanted to bring this to your attention. We completely understand if the score was meant to be adjusted only after this final response, and we are grateful for any score you deem appropriate. We simply wanted to ensure that nothing was overlooked as the discussion concludes.
>
> Thank you again for all your feedback, which has helped to improve our work considerably. If additional adjustments or clarifications are needed, we are more than happy to address and incorporate them.
>
> Best regards, The Authors

---

### Official Review · Reviewer_aHHC · 2024-07-12

**Soundness:** 2
**Presentation:** 2
**Contribution:** 2
**Rating:** 5
**Confidence:** 4

**Summary:**

This paper studies the setting of non-iid train / test distribution shift in the tabular machine learning. The authors extend the previous TabPFN sota tabular model to deal with domain shift cases. TabPFN uses SCM to model data prior, and in this paper, the core idea is to update the SCM prior graph by hypernetworks. The output of hypernetworks is the SCM update. As a result, TabPFN could deal with data shift. The whole idea is very simple and the authors demonstrate the superior effectiveness in various synthetic and real datasets

**Strengths:**

- The authors study the data distribution shift in the tabular domain, which is not commonly studied and provides practical value
- The authors present a simple method (hypernetwork) to update SCM prior
- The resulted method has been verified in numerical experiments with synthetic and real datasets

**Weaknesses:**

The main weakness lies in the experimentation. See the questions below. Technically, the core part is the integration of hypernetworks, which seems very simple. However, in the experiments, there is little discussion of this module. Besides, since many synthetic datasets are used, it is nature to study the SCM from a more explicit way and especially how hypernetworks work well in the setting.

**Questions:**

- If I understand correctly, the core part of the method lies in training a hypernetworks to update SCM. All the rest follows TabPFN?
- It is vague how hypernetworks is trained and processed. It is suggested to provide details on this
- The authors mentioned in Figure 4, various cases of distribution shifts, how are these shifts mitigated in the experiments? Any investigation on different cases?
- How good has the hypernetworks been trained? Since there is many synthetic datasets used, it is great to study cases where hypernetworks updated the SCM correctly
- How much data is required to train hypernetworks?
- How is the hypernetworks training related to number of features, number of classes, etc.  since TabPFN has limitations on the dataset requirement.
- In Table 1, it is recommended to separate real datasets and synthetic datasets. It is also necessary to briefly discuss the nature of the used datasets

---

> ### Author Rebuttal · Authors · 2024-08-07
>
> Dear Reviewer aHHC,
>
> Thank you for your insightful review. We appreciate the time and effort you've invested. Your comments have given us insights to clarify and strengthen several aspects of our paper.
>
> We are especially encouraged by your recognition of the strong practical value of our approach and that temporal distribution shifts in tabular data are understudied. Your acknowledgment reinforces our belief that this work contributes meaningfully to the field.
> Upon careful consideration of your feedback, we realize that there have been fundamental misunderstandings regarding our work. We believe that using the term "hypernetwork" was potentially have thus updated this term to “$2^{\text{nd}}\text{-order SCM}$”  throughout the work. In our revised manuscript, we have made a concerted effort to address these areas, providing additional explanations and context to ensure our methodology and findings are more accurately presented.
>
> In the following responses, we address your questions and recommendations point by point. We have taken the liberty of reordering them for clarity.
>
> > The authors mentioned in Figure 4, various cases of distribution shifts, how are these shifts mitigated in the experiments? Any investigation on different cases?
>
> Great question! One key advantage of our method is that each type of shift, as well as any combination of them, is handled by our model implicitly, as long as our prior generated those shifts during the pre-training phase. During model development, we extensively analyzed the generated datasets to ensure that each type of shift and their combinations would indeed appear.
>
> Regarding the experiments, while these types of shifts are theoretically interesting on their own, in real-world data, they rarely occur in isolation. Nevertheless, we were also quite interested in investigating how our model deals with the different kinds of shifts, which can be seen nicely by examining the decision boundaries, which can be explored through the demo code in our Colab notebook. To simplify this process, we categorized the 2D-synthetic datasets according to their respective types or combinations of shifts (based on their ID):
>
> - **Prior Probability Shift:** test: [1]
> - **Covariate Shift:** valid: [3]
> - **Concept Shift:** test: [5, 6, 7]
> - **Concept Shift + Covariate Shift:** valid: [1,2], test: [0, 2]
> - **All Types of Shifts:** valid: [0]
>
> In addition, we now provide three exemplary decision boundaries for each type of shift individually in Figure 1 of the PDF in the global response to all reviewers. When investigating the decision boundaries, we did not notice a significant difference in performance between the different types of distribution shifts, as our model adjusts well to each of them. For combinations of covariate and concept shifts, we observed that our model handles linear shifts particularly well, while (high-dimensional) rotations in synthetic data remain a challenging task. Nevertheless, we perform significantly better than our baselines in these scenarios.
>
> > In Table 1, it is recommended to separate real datasets and synthetic datasets. It is also necessary to briefly discuss the nature of the used datasets
>
> We agree with your recommendation. In the camera-ready version, we will separate synthetic and real-world datasets in the main text. The initial combination of both types into one table in the main text, with separate tables in the appendix, was primarily due to space restrictions. Now that we have an additional page for the rebuttal, we are including both tables separately in the main text and have added additional discussion of these results to be consistent with the added tables. Additionally, we will provide a brief discussion on the nature of the datasets by adding the following paragraph to L251 in Section 4 - Datasets:
>
> “[...] While some of these datasets have been analyzed in previous work, there has been no comprehensive benchmark focusing on small tabular datasets undergoing distribution shifts. To address this gap, we carefully selected or generated a diverse range of datasets that exhibit temporal distribution shifts. The non-generated datasets were selected from open dataset platforms or previous work in DG. [...]”
>
> Please note, however, that we had provided an in-depth discussion of each dataset used in our evaluation in Section A.11 of the appendix.
>
> *Due to the extensive rebuttal and feedback we wish to provide, we have split our response into two parts.*

---

> ### Author Response · Authors · 2024-08-07
> **Rebuttal by Authors (contd.)**
>
> > If I understand correctly, the core part of the method lies in training a hypernetworks to update SCM. All the rest follows TabPFN?
>
> Not quite. As described in Section 3 of our work, while we do employ a "hypernetwork" to update each data generating SCM, this approach is not merely a simple extension of the TabPFN framework. Instead, it introduces substantial changes in three key areas:
> 1. Temporal Dependence: Unlike TabPFN, which generates each instance within a tabular dataset from the same randomly-sampled SCM during pre-training, our model introduces temporal dependencies within the SCM. We dynamically shift causal relationships as instances are generated, simulating real-world scenarios where underlying models evolve over time.
> 2. Employing a "Hypernetwork": Our "hypernetwork", which is itself a secondary SCM, samples the weight shift for each shifted causal relationship given a certain time index as input. This leads to the generation of correlated weight shifts that adhere to underlying causal principles.
> 3. Temporal Encoding: In processing datasets in the transformer, either generated during pre-training or provided during training and inference, we use Time2Vec encoding for the temporal domain indices. This learned representation attempts to effectively capture temporal aspects of data, such as seasonality, enabling the transformer to better utilize time-indexed information.
>
> To address potential confusion arising from our terminology, it’s important to clarify that the "hypernetwork" we refer to is not the typical learned "hypernetwork" commonly used in machine learning. Instead, it is randomly generated and used solely for sampling. To further enhance clarity, we propose renaming it to “$2^{\text{nd}}\text{-order SCM}$” in our work. Additionally, we will add the following clarification after the end of line 202 in Section 3.2:
>
> “[...] Note that although we perform a forward pass, there is no backward pass associated with it. Each $2^{\text{nd}}\text{-order SCM}$ is randomly generated and used solely for sampling the weight shifts. [...]”
>
> > How good has the hypernetworks been trained? Since there is many synthetic datasets used, it is great to study cases where hypernetworks updated the SCM correctly
>
> We apologize for any confusion caused by our use of the term "hypernetwork". To clarify: Our "hypernetwork" is not a traditional trained neural network, but rather a secondary Structural Causal Model (SCM) used for sampling temporal shifts. This secondary SCM is randomly generated, not trained, and serves to produce correlated weight shifts that adhere to underlying causal principles. This modified prior generates datasets with temporal distribution shifts that closely resemble real-world scenarios. During model development, we conducted extensive analyses on the datasets generated from our prior, and observed the three main types of shifts as well as their combination over time (see Figure 4). To calibrate the strengths and types of these shifts, we employed a random search of hyperparameters on our validation datasets.
>
> > It is vague how hypernetworks is trained and processed. It is suggested to provide details on this
>
> As outlined in the previous response, there seems to be confusion when we refer to our secondary SCM, which is used solely for sampling, as the "hypernet". We are glad that this was caught in the review and have updated this term, see above.
>
> > How much data is required to train hypernetworks?
>
> Since our "hypernetworks" are randomly sampled SCMs, there is no training involved. However, modeling the sample space of SCMs requires careful consideration. For this, we followed the insights of the TabPFN paper as well as empirical evidence by performing a random search of hyperparameters on our validation datasets.
>
> > How is the hypernetworks training related to number of features, number of classes, etc. since TabPFN has limitations on the dataset requirement.
>
> Thank you for this important question. To clarify: Our "hypernetwork" is not trained in the traditional sense, see our comments above. The limitations of TabPFN regarding the number of features and classes remain the same for our model, since the hypernet is only involved in the data generation process. The actual transformer training is therefore nearly as efficient as TabPFNs base implementation. Ongoing improvements made to the TabPFN architecture and in-context learning will translate to our line of work as well.
>
> Thank you again for your feedback. We hope our clarifications and the forthcoming revisions will address your concerns adequately. If this is indeed the case, we would very much appreciate it if you considered raising your score.

---

> ### Comment · Reviewer_aHHC · 2024-08-12
>
> Thank very much the authors for clarifying the terminology. Indeed, hypernetworks are commonly referred as another methodology.
>
> I still feel the improvement wrt TabPFN is bit incremental. But this is very subjective. At any rate, I don't feel a strong novelty in the second order SCM used here, as how SCM is used in Domain Generalization / Invariance Learning literatures.
>
> I am happy to discuss with other reviewers later on it and raise the score if needed.

---

> > ### Comment · Reviewer_aHHC · 2024-08-13
> >
> > Thank the authors for further development of related works. I raised my score.

---

> > > ### Author Response · Authors · 2024-08-13
> > > **Follow up to reviewer response**
> > >
> > > Dear Reviewer aHHC,
> > >
> > > Thank you for your response and willingness to discuss the paper further with the reviewers. We also greatly appreciate your acknowledgement of our efforts to improve the related work section.
> > >
> > > The novelty as mentioned in your response is indeed a subjective topic. However, we believe that our approach of using a second-order SCM within the TabPFN framework, to successfully address temporal distribution shifts in tabular data is indeed very relevant. Although the concept is quite intuitive, the actual implementation involved considerable complexity. Designing the second-order SCM, along with specifying and optimizing all relevant hyperparameters and constructing the necessary sampling spaces, were complex tasks that demanded substantial effort and validation to achieve practical performance. Moreover, the validation of this approach further demonstrates its potential as an effective modeling strategy for future methods outside TabPFN.
> > >
> > > Our empirical results highlight that our modifications to TabPFN can lead to substantial improvements in handling real-world and synthetic datasets with temporal distribution shifts. The performance gains of our method, especially in the synthetic case, are unmatched by existing approaches in our setting and are in our opinion a significant step forward.
> > >
> > > Best regards,
> > > The Authors

---

### Author Rebuttal · Authors · 2024-08-07

We sincerely appreciate all reviewers for their constructive feedback and insightful comments. We have addressed all the key criticisms raised in the reviews and made corresponding adjustments to our paper. We are delighted that the reviewers recognize our approach of modeling temporal distribution shifts in TabPFN via a secondary SCM (previously referred to as “hypernet”) as a “logical addition to the TabPFN method" (LELq) and that our “proposed method intuitively makes sense” (m2Lp). We value the acknowledgment that our paper is “excellently” (84bT) presented and “well structured and well written” (84bT). Furthermore, it is encouraging that the setting is recognized as “not commonly studied” (aHHc) and our method as “providing practical value” (aHHc) with a “real-world impact in areas where TabPFN-like methods are a good fit” (84bT). We also appreciate the recognition that the implicit, unified handling of shifts present in the data “seems to be a clear advantage of this approach” (LELq). We believe our contribution adds significant value to the NeurIPS community by demonstrating a novel approach for handling temporal distribution shifts in tabular data.

Regarding the concerns raised by reviewers LELq and 84bT about the computational budget of our baselines, we have conducted additional experiments to address these questions. The new results, included in Table 1 of the attached PDF, compare the performance reported in our work with 20 minutes of hyperparameter optimization (HPO) to the performance after 1 hour for each dataset split. These results demonstrate that, although the HPO had three times more computational budget, the out-of-distribution performance already saturated within the first 20 minutes.

In response to the feedback from all reviewers, we have made several improvements to our paper:
1. **Framing of Our Work:** We have renamed the paper to "Drift-Resilient TabPFN: In-Context Learning *Temporal Distribution Shifts* on Tabular Data" and have updated the abstract to better reflect the scope of our work.
2. **Terminology Update:** We have renamed “hypernet” to “$2^{\text{nd}}\text{-order SCM}$” to avoid confusion with the commonly used term "hypernetworks" in the literature, as our approach differs significantly.
3. **Dataset Discussion:** We have added a brief discussion about the nature of our datasets in the main paper, complementing the in-depth discussion already provided in the appendix.
4. **Quantitative Results:** We have further improved the clarity of our results by splitting the main table of quantitative results into two separate tables that were previously in the Appendix - one for synthetic datasets and one for real-world datasets.
5. **Improved Visualizations:** We have enhanced the visualizations of the decision boundary to now show the probability of the most likely class at each point. The updated version of Figure 5 is included as Figure 1a in the attached PDF.

We thank the reviewers for their valuable input, which has significantly helped to refine our work. We have addressed all the key points raised during the review and made corresponding adjustments to our paper. We look forward to insightful discussions during the discussion period and are happy to answer any questions or address any misunderstandings that might arise from our responses. We look forward to the opportunity to contribute to NeurIPS and believe our work advances the application of neural networks in domains affected by temporal distribution shifts.

---

> ### Author Response · Authors · 2024-08-12
> **New Version of the Section "Related work"**
>
> Dear Reviewers,
>
> We agree with Reviewer LELq's observation that our original "Related Work" section was insufficiently detailed, primarily due to space limitations. With the additional page provided for the rebuttal, we have significantly expanded and reworked this section. The revised version now offers a more comprehensive overview of related benchmarks, general DG methods, and recent studies on TabPFN. We also detail relevant findings and clarify how they relate to our work. Given the importance of this section, we are sharing the updated “Related Work” section with all of you in the comment below.
>
> We believe these revisions address the primary concerns regarding this section, and we would appreciate any further feedback you may have to ensure clarity and completeness.

---

> ### Author Response · Authors · 2024-08-12
> **Related Work (Part 1)**
>
> ### Related Work
> While DG has drawn increasing attention in the research community [28–38], its temporal variant remains under-explored. In this section, we review existing DG benchmarks on tabular data, DG methods, the temporal DG benchmark Wild-Time [11], specialized temporal DG methods, as well as relevant studies of TabPFN.
>
> **Tabular Distribution Shift Benchmarks.** Several benchmarks have been introduced to assess methods for tabular data under distribution shifts. (1) **Shifts** and **Shifts 2.0** [39, 40] are uncertainty focused benchmarks. **Shifts 2.0** includes five tasks, two of which involve tabular data subject to temporal and spatio-temporal shifts. (2) **WhyShift** [41], focusing on spatio-temporal shifts, offers five real-world tabular datasets, including the ACS dataset [42], which we also use in a subsampled form. Evaluating 22 methods like Gradient Boosted Decision Trees (GBDTs), MLPs, and robustness techniques, WhyShift finds that robustness methods do not consistently enhance out-of-distribution (OOD) performance. They also find that while GBDTs perform better, the gap between in-distribution (ID) and OOD performance persists, likely due to GBDTs being better fitted to the ID distribution. (3) **TableShift** [14] includes 15 tabular binary classification tasks, with 10 relevant to DG. Their work evaluates 19 model types, including several DG methods, finding that methods tailored for distribution shifts do not consistently outperform GBDTs. While generalization gaps can be slightly reduced, each robustness method comes at the cost of ID performance. (4) **Wild-Tab** [43] focuses on domain generalization within tabular regression using three large datasets. Their study compares 10 generalization techniques against standard Empirical Risk Minimization (ERM) applied to MLPs. Similar to previous work [44], they find that ERM was not consistently outperformed by specialized DG methods, and notably, no advantage of GBDTs over ERM on MLPs was observed in their datasets.
>
> Unlike these benchmarks, which focus on large-scale datasets, our work addresses the overlooked challenges of small-scale temporal distribution shift datasets. However, their insights on generalization and robustness methods closely align with our findings, providing valuable context for our approach.
>
> **DG Methods.** Several techniques have been proposed to improve robustness to distribution shifts in DG [45]. Key approaches include domain-invariant learning methods, such as **Deep CORAL** [35], **IRM** [31], and **DANN** [36], which aim to learn representations that generalize across domains. Data augmentation strategies, like **Mixup** [33] and **LISA** [34], contribute to generalization by generating synthetic data variations. Additionally, robust optimization techniques, including **VRex** [37], **GroupDRO** [32], **EQRM** [38], and **SWA** [46], aim to improve performance under distributional shifts by optimizing for worst-case scenarios or incorporating model uncertainty.
>
> In relation to our work, data augmentation strategies are conceptually similar, as we also teach the model DG rather than adding invariances to the architecture. In contrast to augmentation strategies, though, we learn to generalize using completely artificial data on the meta level rather than through manipulation of the target dataset. The other methods focus on designing models specifically for DG, while our approach completely relies on the model to learn to handle distribution shifts.
>
> **Wild-Time Benchmark.** Wild-Time [11] is a benchmark of five datasets designed to study the real-world effects of temporal distribution shifts - an area largely overlooked by previous benchmarks. While Wild-Time primarily uses non-tabular data, it evaluates a wide array of techniques on its tabular dataset, including classical supervised learning (ERM), fine-tuning, and several previously mentioned general DG methods adapted to temporal distribution shifts. Despite the diversity of methods evaluated, Wild-Time reveals a significant performance gap between ID and OOD data, with none of the 13 tested methods consistently outperforming the standard ERM approach.
>
> In our evaluation, we employ their evaluation strategy Eval-Fix and also benchmark our approach against the methods they considered. A comprehensive overview of these methods is provided in Appendix A.8.

---

> ### Author Response · Authors · 2024-08-12
> **Related Work (Part 2)**
>
> **Recent Temporal DG methods.** Recent specialized temporal DG methods include: (1) DRAIN [12], which employs a Bayesian framework alongside a recurrent neural network for predicting the dynamics of the model parameters across temporal domains. (2) GI [13], which explicitly incorporates the temporal domain as a feature, using a specialized Gradient Interpolation loss function, a time-sensitive activation, and enhanced domain reasoning via Time2Vec preprocessing [22]. However, both DRAIN and GI are limited in their ability to extrapolate beyond the near future, leading to their exclusion from our main evaluation. Notably, even on the Rotated Two Moons Dataset - where DRAIN and GI have demonstrated their capabilities - our approach Drift-Resilient TabPFN, outperforms both. See Appendix A.10.8 for details.
>
> **Recent TabPFN Studies.** To overcome the limitations of TabPFN in handling large samples and feature sets, typically found in DG benchmarks, several improvements have been proposed. Ma et al. [47] introduced a data distillation approach, where a distilled dataset serves as the model’s context, optimized to maximize the training-data likelihood. Similarly, TuneTables [48], uses prompt-tuning to compress large datasets into a smaller context. Either of these methods could potentially be combined with Drift-Resilient TabPFN, as our modifications focus primarily on the pre-training phase, which remains unchanged in their approaches.
>
> Another TabPFN variation, ForecastPFN [49], also introduces time dependence, but it does not consider any features. It only models simple time series data. Unlike our approach, which builds on TabPFN’s SCM architecture for pre-training to handle a large set of features, ForecastPFN models synthetic time series using a single handcrafted function with sampled hyperparameters, simplifying the architecture while trading off diversity of the synthetic datasets.
>
> ### References:
> - [...]
> - [11]  Huaxiu Yao, Caroline Choi, Bochuan Cao, Yoonho Lee, Pang Wei Koh, and Chelsea Finn. Wild-time: A benchmark of in-the-wild distribution shift over time. In Oh et al. [52]. URL https://openreview.net/forum?id=F9ENmZABB0.
> - [12]  Guangji Bai, Chen Ling, and Liang Zhao. Temporal domain generalization with drift-aware dynamic neural networks. In Proceedings of the International Conference on Learning Representations (ICLR’23) icl [53]. URL https://openreview.net/forum?- id=sWOsRj4nT1n. Published online: iclr.cc.
> - [13]  Anshul Nasery, Soumyadeep Thakur, Vihari Piratla, Abir De, and Sunita Sarawagi. Training for the future: A simple gradient interpolation loss to generalize along time. In Ranzato et al. [54]. URL https://openreview.net/forum?id=U7SBcmRf65.
> - [14]  Josh Gardner, Zoran Popovic, and Ludwig Schmidt. Benchmarking distribution shift in tabular data with tableshift. Advances in Neural Information Processing Systems, 36, 2024.
> - [...]
> - [22]  Seyed Mehran Kazemi, Rishab Goel, Sepehr Eghbali, Janahan Ramanan, Jaspreet Sahota, Sanjay Thakur, Stella Wu, Cathal Smyth, Pascal Poupart, and Marcus Brubaker. Time2vec: Learning a vector representation of time, 2020. URL https://openreview.net/forum? - id=rklklCVYvB.
> - [...]
> - [29]  Yogesh Balaji, Swami Sankaranarayanan, and Rama Chellappa. Metareg: Towards domain generalization using meta-regularization. In Bengio et al. [55].
> - [30]  Saeid Motiian, Marco Piccirilli, Donald A Adjeroh, and Gianfranco Doretto. Unified deep supervised domain adaptation and generalization. In Proceedings of the IEEE international conference on computer vision, pages 5715–5725, 2017.
> - [31]  Martin Arjovsky, Léon Bottou, Ishaan Gulrajani, and David Lopez-Paz. Invariant risk mini- mization, 2020.
> - [32]  Shiori Sagawa, Pang Wei Koh, Tatsunori B. Hashimoto, and Percy Liang. Distributionally robust neural networks for group shifts: On the importance of regularization for worst-case generalization. In Proceedings of the International Conference on Learning - Representations (ICLR’20), 2020. Published online: iclr.cc.
> - [33]  Hongyi Zhang, Moustapha Cisse, Yann N. Dauphin, and David Lopez-Paz. mixup: Beyond empirical risk minimization. In Proceedings of the International Conference on Learning Representations (ICLR’18), 2018. Published online: iclr.cc.
> - [34]  Huaxiu Yao, Yu Wang, Sai Li, Linjun Zhang, Weixin Liang, James Zou, and Chelsea Finn. Improving out-of-distribution robustness via selective augmentation, 2022.
> - [35]  Baochen Sun and Kate Saenko. Deep coral: Correlation alignment for deep domain adaptation. In Gang Hua and Hervé Jégou, editors, Computer Vision – ECCV 2016 Workshops, pages 443–450, Cham, 2016. Springer International Publishing. doi:10.1007/- 978-3-319-49409-8_35.
> - [36]  Yaroslav Ganin, Evgeniya Ustinova, Hana Ajakan, Pascal Germain, Hugo Larochelle, François Laviolette, Mario Marchand, and Victor Lempitsky. Domain-adversarial training of neural networks. Journal of Machine Learning Research, 17(1):2096–2030, jan 2016. - ISSN 1532- 4435.

---

> ### Author Response · Authors · 2024-08-12
> **Related Work (Part 3)**
>
> - [37]  David Krueger, Ethan Caballero, Joern-Henrik Jacobsen, Amy Zhang, Jonathan Binas, Dinghuai Zhang, Remi Le Priol, and Aaron Courville. Out-of-distribution generalization via risk extrapolation (rex). In M. Meila and T. Zhang, editors, Proceedings of the 38th - International Conference on Machine Learning (ICML’21), volume 139 of Proceedings of Machine Learning Research. PMLR, 2021. URL https://proceedings.mlr.press/v139/krueger21a.html.
> - [38]  Cian Eastwood, Alexander Robey, Shashank Singh, Julius von Kügelgen, Hamed Hassani, George J. Pappas, and Bernhard Schölkopf. Probable domain generalization via quantile risk minimization. In Oh et al. [52]. URL https://proceedings.neurips.cc/paper_files/ - paper/2022/file/6f11132f6ecbbcafafdf6decfc98f7be-Paper-Conference.pdf.
> - [39]  Andrey Malinin, Neil Band, Alexander Ganshin, German Chesnokov, Yarin Gal, Mark J. F. Gales, Alexey Noskov, Andrey Ploskonosov, Liudmila Prokhorenkova, Ivan Provilkov, Vatsal Raina, Vyas Raina, Denis Roginskiy, Mariya Shmatova, Panos Tigar, and Boris - Yangel. Shifts: A dataset of real distributional shift across multiple large-scale tasks. arXiv preprint arXiv:2107.07455, 2021.
> - [40]  Andrey Malinin, Andreas Athanasopoulos, Muhamed Barakovic, Meritxell Bach Cuadra, Mark J. F. Gales, Cristina Granziera, Mara Graziani, Nikolay Kartashev, Konstantinos Kyriakopoulos, Po-Jui Lu, Nataliia Molchanova, Antonis Nikitakis, Vatsal Raina, Francesco - La Rosa, Eli Sivena, Vasileios Tsarsitalidis, Efi Tsompopoulou, and Elena Volf. Shifts 2.0: Extending the dataset of real distributional shifts, 2022. URL https://arxiv.org/abs/2206.15407.
> - [41]  Jiashuo Liu, Tianyu Wang, Peng Cui, and Hongseok Namkoong. On the need for a language describing distribution shifts: Illustrations on tabular datasets. In Thirty-seventh Conference on Neural Information Processing Systems Datasets and Benchmarks Track, - 2023.
> - [42]  Frances Ding, Moritz Hardt, John Miller, and Ludwig Schmidt. Retiring adult: New datasets for fair machine learning. In Ranzato et al. [54].
> - [43]  Sergey Kolesnikov. Wild-tab: A benchmark for out-of-distribution generalization in tabular regression, 2023. URL https://arxiv.org/abs/2312.01792.
> - [44]  Ishaan Gulrajani and David Lopez-Paz. In search of lost domain generalization. In Proceedings of the International Conference on Learning Representations (ICLR’21), 2021. URL https: //openreview.net/forum?id=lQdXeXDoWtI. Published online: iclr.cc.
> - [45]  Jindong Wang, Cuiling Lan, Chang Liu, Yidong Ouyang, Tao Qin, Wang Lu, Yiqiang Chen, Wenjun Zeng, and Philip S. Yu. Generalizing to unseen domains: A survey on domain generalization. IEEE Transactions on Knowledge and Data Engineering, 35(8):8052–8072, - 2023. doi:10.1109/TKDE.2022.3178128.
> - [46]  Pavel Izmailov, Dmitrii Podoprikhin, Timur Garipov, Dmitry Vetrov, and Andrew Gordon Wilson. Averaging weights leads to wider optima and better generalization. In A. Globerson and R. Silva, editors, Proceedings of The 34th Uncertainty in Artificial - Intelligence Conference (UAI’18). AUAI Press, 2018.
> - [47]  Junwei Ma, Valentin Thomas, Guangwei Yu, and Anthony Caterini. In-context data distillation with tabpfn, 2024. URL https://arxiv.org/abs/2402.06971.
> - [48]  Benjamin Feuer, Robin Tibor Schirrmeister, Valeriia Cherepanova, Chinmay Hegde, Frank Hutter, Micah Goldblum, Niv Cohen, and Colin White. Tunetables: Context optimization for scalable prior-data fitted networks. arXiv preprint arXiv:2402.11137, 2024.
> - [49]  Samuel Dooley, Gurnoor Singh Khurana, Chirag Mohapatra, Siddartha V Naidu, and Colin White. Forecastpfn: Synthetically-trained zero-shot forecasting. In Advances in Neural Information Processing Systems, 2023.
> - [...]

---

### Decision · Program_Chairs · 2024-09-25

**Decision:**

Accept (poster)

**Comment:**

This paper extends Prior-Data Fitted Networks (PFNs) for tabular data (TabPFNs) to handle temporal distribution shifts by augmenting the base model with a structural causal model (SCM) as a prior.  The core idea is to use a hypernetwork (which the authors later call a 2nd-order SCM) to perform amortized computation to adjust the SCM to the drift.  The reviewers were unanimously in favor of acceptance albeit with varying degrees of enthusiasm (1 x borderline accept, 2 x weak accept, 1 x accept).  Reviewer aHHC had the most negative opinion of the work, with the primary criticisms being the lack of clarity in the description of the training process and the work being incremental in nature.  The former issue was cleared up the authors in the rebuttal.  The latter issue still remained, but in my judgement, the paper is a clear extension of TabPFNs, and while the novelty lies in including a standard technique from the SCM literature, the exact methodology does not yet exist and demonstrates empirical merits.

Reviewer LELq took issue with the paper's framing and presentation of related work (which was echoed in other reviews).  In my own reading, indeed the current draft suffers from some over generalizations (e.g. scoping all shifts vs a focus on temporal, "Until now, no method has significantly outperformed classical supervised learning...") and a lack of organization in its description of related work.  The related work, while there is a dedicated section, is still dispersed throughout the introduction and even in Section 3.  The authors have promised to address these issues in the camera-ready version.  While the authors should certainly make these changes, I feel that these are mostly cosmetic issues that do not jeopardize the core contribution of the work.